# AN EFFECTIVE THEORY OF BIAS AMPLIFICATION

**Arjun Subramonian**[1,][*] **Samuel J. Bell**[2]**, Levent Sagun**[2]**, Elvis Dohmatob**[2,3,4]
[1]UCLA    [2]Meta FAIR    [3]Concordia University    [4]Mila

## ABSTRACT

Machine learning models can capture and amplify biases present in data, leading to disparate test performance across social groups. To better understand, evaluate, and mitigate these biases, a deeper theoretical understanding of how model design choices and data distribution properties contribute to bias is needed. In this work, we contribute a precise analytical theory in the context of ridge regression, both with and without random projections, where the former models feedforward neural networks in a simplified regime. Our theory offers a unified and rigorous explanation of machine learning bias, providing insights into phenomena such as bias amplification and minority-group bias in various feature and parameter regimes. For example, we observe that there may be an optimal regularization penalty or training time to avoid bias amplification, and there can be differences in test error between groups that are not alleviated with increased parameterization. Importantly, our theoretical predictions align with empirical observations reported in the literature on machine learning bias. We extensively empirically validate our theory on synthetic and semi-synthetic datasets.

## 1 INTRODUCTION

Machine learning (ML) datasets can encode a plethora of biases which, when said data is used to train models, can result in systems that can cause practical harm. Datasets that encode correlations that only hold for a subset of the data may cause disparate performance when models are used more broadly, such as an X-ray pneumonia classifier that only functions on images from certain hospitals (Zech et al., 2018). This issue is magnified when coupled with under-representation, whereby a dataset fails to adequately reflect parts of the underlying data distribution, often further marginalizing certain groups. Lack of representation results in systems that might work well on average, but fail for minoritized groups, including facial recognition systems that fail for darker-skinned women (Buolamwini & Gebru, 2018), large language models that consistently misgender transgender and nonbinary people (Ovalle et al., 2023), or image classification technology that only works in Western contexts (de Vries et al., 2019; Richards et al., 2024).

Unfortunately, contemporary models may exhibit *bias amplification*, whereby dataset biases are not only replicated, but exacerbated (Zhao et al., 2017; Hendricks et al., 2018; Wang & Russakovsky, 2021). While previous research has shown that amplification is a function of both dataset properties and how we choose to construct our models (Hall et al., 2022; Sagawa et al., 2020; Bell & Sagun, 2023), it is not fully clear how bias amplification occurs mechanistically, nor do we precisely understand which settings lead to its emergence. Thus, in this work, we propose a novel theoretical framework that explains how model design choices (e.g., number of parameters, regularization penalty) and data distributional properties (e.g., number of features, group imbalance, label noises) interact to amplify bias. Moreover, our framework provides an account of prior work on bias amplification (Bell & Sagun, 2023) and minority-group bias (Sagawa et al., 2020).

A theory of bias amplification is important for several reasons. First, as empirical research necessarily yields only sparse data points—often focused on only the most common regimes—theory allows us to interpolate between past findings, and reason about how bias emerges in under-explored settings. Second, a precise theory gives us the depth of understanding needed in order to intervene, potentially supporting the development of both novel evaluations and mitigations. Finally, beyond explaining

---

[*]Work done while interning at Meta. Corresponding author: Arjun Subramonian (arjunsub@cs.ucla.edu).

already-known phenomena, our theory makes new predictions, suggesting new avenues for future research.

## 1.1 MAIN CONTRIBUTIONS

In this work, we develop a unifying and rigorous theory of ML bias in the settings of ridge regression with and without random projections. In particular, we precisely analyze test error disparities between groups (e.g., demographic groups or protected categories) with different data distributions when training on a mixture of data from these groups. We characterize these disparities in high dimensions using operator-valued free probability theory (OVFPT), thereby avoiding possibly loose bounds on critical quantities. Our theory encompasses different parameterization regimes, group sizes, label noises, and data covariance structures. Moreover, our theory has applications to important problems in ML bias that have recently been empirically investigated:

- **Bias amplification.** Even in the absence of group imbalance and spurious correlations, a single model that is trained on a combination of data from different groups can amplify bias beyond separate models that are trained on data from each group (Bell & Sagun, 2023). With our theory, we reproduce and analyze the bias amplification findings of Bell & Sagun (2023) in controlled settings. We further observe how stopping model training early or tuning the regularization hyperparameter can alleviate bias amplification.
- **Minority-group bias.** Overparamaterization can hurt test performance on minority groups due to spurious features (Sagawa et al., 2020; Khani & Liang, 2021). We theoretically analyze how model size and extraneous features affect minority-group bias.

We extensively empirically validate our theory in controlled and semi-synthetic settings. Specifically, we show that our theory aligns with practice in the cases of: (1) bias amplification with synthetic data generated from isotropic covariance matrices and the semi-synthetic dataset Colored MNIST (Arjovsky et al., 2019), and (2) minority-group bias under different model sizes with synthetic data generated from diatomic covariance matrices. In these applications, we expose new, interesting phenomena in various regimes. For example, a larger number of features than samples can amplify bias under overparameterization, there may be an optimal regularization penalty or training time to avoid bias amplification, and there can be differences in test error between groups that are not alleviated with increased parameterization. Our observations of phenomena in Sections 4 and 5 are largely empirical but are supported by their agreement with our theory. Our theory of ML bias can inform strategies to evaluate and mitigate unfairness in ML, or be used to caution against the usage of ML in certain applications.

## 1.2 RELATED WORK

**Bias amplification.** A long line of research has explored how ML exacerbates biases in data. For example, a single model that is trained on a combination of data from different groups can amplify bias (Zhao et al., 2017; Wang & Russakovsky, 2021), even beyond what would be expected when separate models are trained on data from each group (Bell & Sagun, 2023). Hall et al. (2022) conduct a systematic empirical study of bias amplification in the context of image classification, finding that amplification can vary greatly as a function of model size, training set size, and training time. Furthermore, overparameterization, despite reducing a model's overall test error, can disproportionately hurt test performance for minority groups (Sagawa et al., 2020; Khani & Liang, 2021). Models can also overestimate the importance of poorly-predictive, low-signal features for minority groups, thereby hurting performance on these groups (Leino et al., 2019). In this paper, we distill a holistic theory of how model design choices and data distributional properties affect disparate test performance across groups, which can encompass seemingly disparate bias phenomena.

**High-dimensional analysis of ML.** A suite of works have analyzed the expected dynamics of ML in appropriate asymptotic scaling limits, e.g., the rate of features $d$ to samples $n$ converges to a finite values as $d$ and $n$ respectively scale towards infinity (Adlam & Pennington, 2020b; Tripuraneni et al., 2021; Lee et al., 2023). Notably, Bach (2023) theoretically analyzes the double descent phenomenon (Spigler et al., 2019; Belkin et al., 2019) in ridge regression with random projections by computing deterministic equivalents for relevant random matrix quantities in a proportionate scaling limit. Like Adlam & Pennington (2020b); Tripuraneni et al. (2021); Lee et al. (2023), we leverage the tools of

OVFPT (Mingo & Speicher, 2017), which is at the intersection of random matrix theory (RMT) and functional analysis. However, Adlam & Pennington (2020b) focus on training and testing a random features model on data from the same Gaussian distribution. Furthermore, Tripuraneni et al. (2021); Lee et al. (2023) focus on training a random features model on data from one Gaussian distribution and testing the model on a different Gaussian. In contrast, we study the random features model in the setting of training on a mixture of Gaussian distributions and testing on each component. Because a mixture is more expressive than a single Gaussian, our theoretical results cannot be derived as a special case of these other works. Furthermore, our theory non-trivially generalizes Bach (2023), which we recover in Corollary J.1 as a special case, and requires more powerful analytical techniques.

Certain prior theoretical work precisely analyzes the bias of models trained on a mixture of data from different groups in a high-dimensional setting (Mannelli et al., 2024; Jain et al., 2024). Like Mannelli et al. (2024); Jain et al. (2024), we study linear models and consider bias as the disparity in test performance of a model between groups. We further consider some similar factors that give rise to bias amplification (e.g., group imbalance, group data variance, inter-group similarity, dataset size). We also share some theoretical conclusions, such as bias can occur even when the groups have the same ground-truth weights (see Section 5) and are balanced (Section 4.1). Additionally, we both discuss the paradigms of training a single model for both groups vs. separate models for each group. However, the *main distinction* between our work and Mannelli et al. (2024); Jain et al. (2024) is that we precisely characterize how models amplify bias in different *parameterization regimes*, that is, we examine the impact of model size on bias. This enables us to expose new, richer insights into the impact of over- and underparameterization on bias amplification (see Figure 1, Section 4, and Section 5). See Appendix C for further comparison of our work to Mannelli et al. (2024); Jain et al. (2024).

## 2 PRELIMINARIES

### 2.1 DATA DISTRIBUTIONS

We consider a ridge regression problem on a dataset from the following multivariate Gaussian mixture with two groups $s = 1$ and $s = 2$. These groups could represent different demographic groups or protected categories.

$$\text{(Group ID) } \text{Law}(s) = \text{Bernoulli}(p), \tag{1}$$

$$\text{(Features) } \text{Law}(x \mid s) = \mathcal{N}(0, \Sigma_s), \tag{2}$$

$$\text{(Ground-truth weights) } \text{Law}(w_1^*) = \mathcal{N}(0, \Theta/d), \quad \text{Law}(w_2^* - w_1^*) = \mathcal{N}(0, \Delta/d), \tag{3}$$

$$\text{(Labels) } \text{Law}(y \mid s, x) = \mathcal{N}(f_s^\star(x), \sigma_s^2), \text{ with } f_s^\star(x) := x^\top w_s^*. \tag{4}$$

The scalar $p \in (0, 1)$ controls for the relative size of the two groups (e.g., $p = 1/2$ in the balanced setting). For simplicity of notation, we define $p_1 = p$ and $p_2 = 1 - p$. The $d \times d$ positive-definite matrices $\Sigma_1$ and $\Sigma_2$ are the covariance matrices for the different groups. The $d$-dimensional vectors $w_1^*$ and $w_2^*$ are the ground-truth weights vectors for each group. $w_1^*$ and $w_2^* - w_1^*$ are independently sampled from zero-mean Gaussian distributions with covariances $\Theta/d$ and $\Delta/d$, respectively. In particular, setting $\Delta = 0$ corresponds to the case that both groups have identical ground-truth weights. We define $\Theta_1 = \Theta, \Theta_2 = \Theta + \Delta$. Finally, $\sigma_s^2$ corresponds to the label noise for each group $s$. While we consider the case of two groups only for conciseness, our theoretical methods readily extend to any finite number of groups.

### 2.2 MODELS AND METRICS

**Learning.** A learner is given an IID sample $\mathcal{D} = \{(x_1, y_1), \ldots, (x_n, y_n)\} \equiv (X \in \mathbb{R}^{n \times d}, Y \in \mathbb{R}^n)$ of data from the above distribution and it learns a model for predicting the label $y$ from the feature vector $x$. Thus, $X$ is the total design matrix with $i$th row $x_i$, and $y$ the total response vector with $i$th component $y_i$. Let $\mathcal{D}^s = (X \in \mathbb{R}^{n_s \times d}, Y \in \mathbb{R}^{n_s})$ be the data pertaining only to group $s$, so that $\mathcal{D} = \mathcal{D}^1 \cup \mathcal{D}^2$ is a partitioning of the entire dataset. Two choices are available to the learner: (1) learn a model a $\widehat{f}_s \in \mathcal{F}$ on each dataset $\mathcal{D}^s$, or (2) learn a single model $\widehat{f} \in \mathcal{F}$ on the entire dataset $\mathcal{D}$. In practice, a choice is made based on scaling vs. personalization considerations.

We consider two solvable settings for linear models: classical ridge regression in the ambient input space, and ridge regression in a feature space given by random projections. The latter allows us to

study the role of model size in ML bias, by varying the output dimension of the random projection mapping. This output dimension $m$ controls the size of a feedforward neural network in a simplified regime (Maloney et al., 2022; Bach, 2023).

**Classical Ridge Regression.** We will first consider the function class $\mathcal{F} \subseteq \{\mathbb{R}^d \rightarrow \mathbb{R}\}$ of linear ridge regression models without random projections. For any vector $w \in \mathbb{R}^d$, the model $f$ with parameters $w$ is defined by $f(x) = x^\top w$, for all $x \in \mathbb{R}^d$, and is learned with $\ell_2$-regularization. We define the generalization error or risk of any model $f$ with respect to group $s$ as:

$$R_s(f) = \mathbb{E}\left[(f(x) - f_s^\star(x))^2 \mid s\right]. \tag{5}$$

We consider ridge regression because in addition to its analytical tractability, it can be viewed as the asymptotic limit of many learning problems (Dobriban & Wager, 2018; Richards et al., 2021; Hastie et al., 2022). We now formally define some metrics related to bias amplification.

**Definition 2.1** (Bias Amplification). *We isolate the contribution of the model to bias when learning from data with different groups. This intuitive conceptualization of bias amplification allows us to quantify the phenomenon. Grounded in the literature (Bell & Sagun, 2023), we define the Expected Difficulty Disparity (EDD) as:*

$$EDD = |\mathbb{E}\, R_2(\widehat{f}_2) - \mathbb{E}\, R_1(\widehat{f}_1)|, \tag{6}$$

*where the expectations are w.r.t. randomness in the training data and any other sources of randomness in the models. The EDD captures the difference in test risk between models trained and evaluated on each group separately. In contrast, we define the Observed Difficulty Disparity (ODD) as:*

$$ODD = |\mathbb{E}\, R_2(\widehat{f}) - \mathbb{E}\, R_1(\widehat{f})|. \tag{7}$$

*The ODD captures the bias (i.e., difference in test risk between groups) of a model trained on both groups. Finally, we define the Amplification of Difficulty Disparity (ADD) as $ADD = \frac{ODD}{EDD}$. We say that bias amplification occurs when $ADD > 1$. See Appendix C for further motivation of ADD.*

**Ridge Regression with Random Projections.** We consider feedforward neural networks in a simplified regime which can be approximated via random projections, i.e., a one-hidden-layer neural network $f(x) = v^\top Sx$ with a *linear* activation function. In particular, we extend classical ridge regression by transforming our learned weights as $\widehat{w} = S\widehat{\eta} \in \mathbb{R}^d$, where $S \in \mathbb{R}^{d \times m}$ is a random projection with entries that are IID sampled from $\mathcal{N}(0, 1/d)$. Ridge regression with random projections offers analytical tractability while exposing bias amplification phenomena related to model size; such phenomena are not exposed by classical ridge regression (see Figure 1). Moreover, it has been shown that in high dimensions, training a one-hidden-layer neural network with gradient descent effectively learns a linear predictor over random features (Yehudai & Shamir, 2019). Furthermore, Adlam & Pennington (2020a); Bach (2023); inter alia are able to reproduce interesting phenomena like double descent using the random features model. Nevertheless, Yehudai & Shamir (2019) have shown that the model often cannot learn even a ReLU neuron, suggesting that some mechanisms of bias amplification could be different in nonlinear networks.

## 3 THEORETICAL ANALYSIS

**Assumptions.** Some of our theorems will require standard technical assumptions that we detail here and in Appendix B. Assumption 3.1 describes the proportionate scaling limits, standard in RMT, in which we will work. These limits enable us to derive deterministic analytical formulae for the expected test risk of models. Our experiments (see Sections 4 and 5) validate our theory.

**Assumption 3.1.** *In the case of ridge regression with random projections, we will work in the following proportionate scaling limit:*

$$n, n_1, n_2, d \rightarrow \infty, \quad n_1/n \rightarrow p_1, n_2/n \rightarrow p_2, d/n \rightarrow \phi, m/n \rightarrow \psi, m/d \rightarrow \gamma, \tag{8}$$

$$d/n_1 \rightarrow \phi_1, m/n_1 \rightarrow \psi_1, \quad d/n_2 \rightarrow \phi_2, m/n_2 \rightarrow \psi_2, \tag{9}$$

*for some constants $\phi_1, \phi_2, \phi, \psi_1, \psi_2, \psi \in (0, \infty)$. The scalar $\phi$ captures the rate of features to samples. Observe that $\phi = p_1\phi_1$ and $\phi = p_2\phi_2$. We note that $\phi\gamma = \psi$ and $\phi_s\gamma = \psi_s$. The scalar $\psi$ captures the rate of parameters to samples. The setting $\psi > 1$ (resp. $\psi < 1$) corresponds to the overparameterized (resp. underparameterized) regime.*

### 3.1 Main Result: Ridge Regression with Random Projections

To provide a mechanistic understanding of how ML models may amplify bias, our theory elucidates differences in the test error between groups when a single model is trained on a combination of data from both groups vs. when separate models are trained on data from each group. We first consider the classical ridge regression model in Appendix D before studying ridge regression with random projections below, which is a more realistic but still analytically solvable setup.

**Single Random Projections Model Learned for Both Groups.** We first consider the ridge regression model $\widehat{f}$ with random projections, which is learned using empirical risk minimization and $\ell_2$-regularization with penalty $\lambda$. The parameter $\widehat{w}$ of the linear model $\widehat{f}$ is given by the following optimization problem:

$$\widehat{w} = S\widehat{\eta} \in \mathbb{R}^d, \text{ with } \widehat{w} = \arg\min_{\eta \in \mathbb{R}^m} L(\eta) = \sum_{s=1}^{2} n^{-1}\|X_s S\eta - Y_s\|_2^2 + \lambda\|\eta\|_2^2. \tag{10}$$

Explicitly, one can write $\widehat{w} = S(Z^\top Z + n\lambda I_m)^{-1}Z^\top Y$, where $Z := XS$. Before presenting our result for the random projections model, we provide some relevant definitions.

**Definition 3.1.** *Let $\bar{\mathrm{tr}}\, A := (1/d)\,\mathrm{tr}\, A$ be the normalized trace operator and $(e_1, e_2, \tau, u_1, u_2, \rho)$ be the unique positive solution to the following fixed-point equations:*

$$1/\tau = 1 + \bar{\mathrm{tr}}\, LK^{-1}, \quad 1/e_s = 1 + \psi\tau\bar{\mathrm{tr}}\, \Sigma_s K^{-1}, \text{ for } s \in \{1, 2\}, \tag{11}$$

$$\rho = \tau^2\bar{\mathrm{tr}}\,(\gamma\rho L^2 + \lambda^2 D)K^{-2}, \quad u_s = \psi e_s^2\bar{\mathrm{tr}}\, \Sigma_s(\gamma\tau^2 D + \rho I_d)K^{-2}, \text{ for } s \in \{1, 2\}, \tag{12}$$

$$\text{where: } L = p_1 e_1 \Sigma_1 + p_2 e_2 \Sigma_2, \ K = \gamma\tau L + \lambda I_d, \ D = p_1 u_1 \Sigma_1 + p_2 u_2 \Sigma_2 + B. \tag{13}$$

*For deterministic $d \times d$ PSD matrices $A$ and $B$, we define the following auxiliary quantities:*

$$h_j^{(1)}(A) := p_j \gamma e_j \tau \bar{\mathrm{tr}}\, A\Sigma_j K^{-1}, \tag{14}$$

$$h_j^{(2)}(A, B) := p_j \gamma \bar{\mathrm{tr}}\, A\Sigma_j(\gamma e_j \tau^2 B + p_{j'}\gamma\tau^2\Sigma_{j'}(e_j u_{j'} - e_{j'}u_j) + e_j\rho I_d - \lambda u_j \tau I_d)K^{-2}, \tag{15}$$

$$\begin{aligned} h_j^{(3)}(A, B) := {}& p_j\bar{\mathrm{tr}}\, A\Sigma_j(\gamma e_j^2 p_j \Sigma_j(p_{j'}\gamma\tau^2 u_{j'}\Sigma_{j'} + \gamma\tau^2 B + \rho I_d) \\ & + u_j(p_{j'}\gamma e_{j'}\tau\Sigma_{j'} + \lambda I_d)^2)K^{-2}, \end{aligned} \tag{16}$$

$$\begin{aligned} h_j^{(4)}(A, B) := {}& p_j\gamma p_{j'}\bar{\mathrm{tr}}\, \Sigma_j\Sigma_{j'}A(\gamma\tau^2(e_j e_{j'}B - p_j e_j^2 u_{j'}\Sigma_j - p_{j'}\Sigma_{j'}e_{j'}^2 u_j) \\ & - \lambda\tau(e_j u_{j'} + e_{j'}u_j)I_d + e_j e_{j'}\rho I_d)K^{-2}. \end{aligned} \tag{17}$$

In Appendix H, we intuitively interpret the scalars $e_s, \tau, u_s, \rho$ in the setting where a separate model is learned for each group. In essence, our theory extends the scalars to the more general setting where a single model is trained on a mixture of data from different groups. Furthermore, each of the terms $h_j^{(1)}, \ldots, h_j^{(4)}$ capture the limiting values of different sources of covariance between the sample covariance matrices for the groups, the resolvent matrix, and the random projections matrix $S$. These sources of covariance are written explicitly in Appendix G, and naturally arise from expanding the solution to the ridge regression problem with random projections.

We now present Theorem 3.1, which is our *main contribution*. Theorem 3.1 presents a novel bias-variance decomposition for the test error $R_s(\widehat{f})$ for each group $s \in \{1, 2\}$ in the context of ridge regression with random projections. It is a non-trivial generalization of theories in high-dimensional ML which requires the powerful machinery of OVFPT (see proof in Appendix G).

**Theorem 3.1.** *Under Assumptions B.2 and 3.1, it holds that $R_s(\widehat{f}) \simeq B_s(\widehat{f}) + V_s(\widehat{f})$, with*

$$V_s(\widehat{f}) = \sum_{j=1}^{2} \sigma_j^2 \phi h_j^{(2)}(I_d, \Sigma_s), \tag{18}$$

$$B_s(\widehat{f}) = \bar{\mathrm{tr}}\, \Theta_s\Sigma_s + h_1^{(3)}(\Theta_s, \Sigma_s) + h_2^{(3)}(\Theta_s, \Sigma_s) + 2h_1^{(4)}(\Theta_s, \Sigma_s) \tag{19}$$

$$- 2h_1^{(1)}(\Theta_s\Sigma_s) - 2h_2^{(1)}(\Theta_s\Sigma_s) + h_{s'}^{(3)}(\Delta, \Sigma_s) \tag{20}$$

$$- 2\begin{cases} 0, & s = 1, \\ h_1^{(3)}(\Delta, \Sigma_2) + h_2^{(4)}(\Delta, \Sigma_2) - h_1^{(1)}(\Delta\Sigma_2), & s = 2. \end{cases} \tag{21}$$

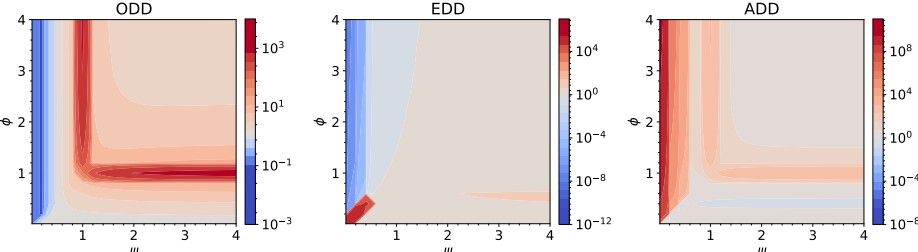

Figure 1: $ODD$, $EDD$, **and** $ADD$ **phase diagrams for ridge regression with random projections.** We plot the bias amplification phase diagrams with respect to $\phi$ (rate of features to samples) and $\psi$ (rate of parameters to samples), as predicted by our theory for ridge regression with random projections (Theorems 3.1, 3.2). Red regions indicate theoretical predictions greater than 1 (i.e., bias amplification in the rightmost plot), while blue regions indicate theoretical predictions less than 1 (i.e., bias deamplification in the rightmost plot). Darkness indicates intensity. We consider isotropic covariance matrices: $\Sigma_1 = 2I_d, \Sigma_2 = I_d, \Theta = 2I_d, \Delta = I_d$. Additionally, $n = 1 \times 10^4, \sigma_1^2 = \sigma_2^2 = 1$. We further choose $\lambda = \lambda_1 = \lambda_2 = 1 \times 10^{-6}$ to approximate the minimum-norm interpolator. We show that bias amplification can occur even in the balanced data setting, i.e., when $p_1 = p_2 = 1/2$.

The unregularized limit corresponds to the minimum-norm interpolator, and alternatively may be viewed as training a neural network until convergence (Ali et al., 2019). We discuss methods for, and the complexity of, solving the above fixed-point equations in Appendix I. Furthermore, in Appendix J, we directly express the bias and variance of the test risk of an unregularized model trained on just group $s$ in terms of the second and first-order degrees of freedom of $\Sigma_s$ and the parameterization rate $\psi_s$. Moreover, in Appendix M, we derive the approximate bias amplification profile of an unregularized model with respect to the ratio $c = \sigma_2^2/\sigma_1^2$ of label noises, in the setting where the eigenspectra of the covariance matrices have power-law decay.

**Separate Random Projections Model Learned for Each Group.** We now consider the ridge regression models $\widehat{f}_1$ and $\widehat{f}_2$ with random projections, which are learned using empirical risk minimization and $\ell_2$-regularization with penalties $\lambda_1$ and $\lambda_2$, respectively. In particular, we have the following optimization problem for each group $s$: $\arg\min_{\eta \in \mathbb{R}^m} L(w) = n_s^{-1}\|X_s S\eta - Y_s\|_2^2 + \lambda_s\|\eta\|_2^2$. Alternatively, the reader can think of each $\widehat{f}_s$ as the limit of $\widehat{f}$ when $p_s \to 1$. In this setting, we deduce Theorem 3.2, which follows from Theorem 3.1.

**Theorem 3.2.** *Under Assumptions B.2 and 3.1, it holds that* $R_s(\widehat{f}_s) \simeq B_s(\widehat{f}_s) + V_s(\widehat{f}_s)$, *where* $V_s(\widehat{f}_s) = \lim_{p_s \to 1} V_s(\widehat{f})$ *and* $B_s(\widehat{f}_s) = \lim_{p_s \to 1} B_s(\widehat{f})$ *(see Appendix H for explicit formulae).*

**Phase Diagram.** The phase diagram for the random projections model (Figure 1) offers rich insights into how the rate of parameters to samples ($\psi$), in interaction with the rate of features to samples ($\phi$), affects bias amplification. In the $ODD$ and $EDD$ profiles, we observe phase transitions at $\phi = \psi$ (when $\psi < 0.5$) and $\psi = 0.5$ (i.e., $\psi_1 = \psi_2 = 1$), where these metrics begin decreasing significantly. $\psi_s = 1$ is a known interpolation threshold for random features models (Adlam & Pennington, 2020a; D'Ascoli et al., 2020). In contrast, at $\psi = 1$ and $\phi = 1$, the $ODD$ drastically increases. Furthermore, at $\phi = \psi$ (when $\psi < 0.5$) and $\phi = 0.5$ (for $\psi > 0.5$), the $EDD$ greatly increases. Accordingly, in the $ADD$ profile, we observe phase transitions at $\phi = \psi$ (when $\psi < 0.5$), $\psi = 0.5$, $\psi = 1$, and $\phi = 1$, where bias amplification begins occurring (i.e., $ADD > 1$). However, bias seems to be deamplified (i.e., $ADD < 1$) at $\phi = \psi$ (when $\psi < 0.5$) and $\phi = 0.5$ (when $\psi > 0.5$). Some observations are less visible due the granularity of the color thresholding in Figure 1.

## 4 BIAS AMPLIFICATION

We empirically show how ridge regression models with random projections may amplify bias when a single model is trained on a combination of data from different groups vs. when separate models are trained on data from each group (Bell & Sagun, 2023). We further show how our theory: (1) predicts bias amplification, and (2) exposes new, interesting bias amplification phenomena in various regimes.

### 4.1 ISOTROPIC COVARIANCE

**Setup.** To mirror the setting of Bell & Sagun (2023), we consider balanced data ($p_1 = p_2 = 1/2$) without spurious correlations ($\Sigma_1 = a_1 I_d, \Sigma_2 = a_2 I_d$, for $a_1, a_2 > 0$). The groups have different ground-truth weights ($\Theta = 2I_d, \Delta = I_d$). Refer to App. K.1 for full details due to space limitations.

**Validation of Theory.** Figure 2 and the figures in Appendix L reveal that Theorems 3.1 and 3.2 closely predict the $ODD$, $EDD$, and $ADD$ of ridge regression models with random projections under diverse settings. Note that, as indicated by the error bars, some of our empirical estimates (especially those with larger magnitude) have higher variance and their variance is influenced by the choice of $\psi, \phi, a_1, a_2, \sigma_1^2, \sigma_2^2$. **Notably, our theory predicts the observation of Bell & Sagun (2023) that models can amplify bias even with balanced groups and without spurious correlations.** We present new phenomena predicted by our theory below.

**Effect of Label Noise.** In the $ODD$ profile, when the label noise ratio $c = \sigma_2^2/\sigma_1^2$ is larger, the right tail is higher for $\phi$ (rate of features to samples) closer to 1 than other $\phi$. This suggests that under overparameterization, a larger noise ratio and similar number of features and samples can increase disparities in test risk between groups when a single model is learned for both groups. We aim to explain this phenomenon analytically in Section M. Moreover, the $EDD$ curve is generally higher for larger $c$, suggesting that a larger noise ratio increases disparities in test risk when a separate model is learned for each group. This finding is supported by our experiment with real data (see Figure 11).

**Effect of Model Size.** We observe interesting divergent behavior as $\psi$ (rate of parameters to samples) increases for different $\phi$ (rate of features to samples). When $\phi > 1$, as $\psi$ increases, the $ODD$ increases and then decreases, peaking at the interpolation threshold at $\psi = 1$. Similarly, when $\phi > 0.5$ (i.e., $\phi_1 = \phi_2 > 1$), as $\psi$ increases, the $EDD$ increases and then decreases, peaking at the interpolation threshold at $\psi = 0.5$ (i.e., $\psi_1 = \psi_2 = 1$). Accordingly, when $\phi > 0.5$, bias is effectively deamplified ($ADD < 1$) at $\psi = 0.5$ and when $\phi > 1$, bias amplification peaks ($ADD > 1$) at $\psi = 1$. In contrast, when $\phi < 1$, the $ODD$ decreases as $\psi$ increases, plateauing at different finite values. Similarly, when $\phi < 0.5$, the $EDD$ generally decreases and plateaus as $\psi$ increases; in some cases, the $EDD$ dips and/or increases and plateaus. A notable exception to these trends occurs when $\phi \approx 1$, with the corresponding $ODD$ and $ADD$ curves increasing as $\psi$ increases, plateauing at a significantly larger value (i.e., $ADD \gg 1$) than the curves corresponding to other values of $\phi$. We observe a similar phenomenon for the $EDD$ curves when $\phi_1 = \phi_2 \approx 1$. Hence, overparameterization can greatly amplify bias when the number of features is close to the number of samples. Regardless of the regime of $\phi$, the left tail of the $ADD$ profile appears to plateau at 1. The right tail plateaus at different finite values, with the curves corresponding to $\phi > 1$ consistently plateauing above 1. This suggests that when there are more features than samples, overparameterization amplifies bias.

Some of the peaks and valleys in Figure 2 can be attributed to double descent. However, double descent in high dimensions has primarily been studied in the setting where data are drawn from a single Gaussian distribution; this corresponds to the $EDD$ setting, where a separate model is learned for each group. In Figure 1, we observe a double descent peak in the $EDD$ at $\psi_1 = \psi_2 = 1$ (Adlam & Pennington, 2020a; D'Ascoli et al., 2020). Our work extends the theoretical treatment of double descent to the setting of training a model on a mixture of Gaussians. However, our theory of bias amplification cannot be reduced to double descent. For example, we note other interpolation thresholds in Figure 1; our use of a linear activation does not have a confounding effect here, as interpolation thresholds have also been observed in random features models with nonlinear activations (Adlam & Pennington, 2020b). In addition, much of Sections 4 and 5, and Appendix M, are devoted to studying the tails or limiting behavior of bias amplification with respect to $\psi$ and $\phi$.

**Effect of Number of Features.** In the $ODD$ and $ADD$ profiles, when the rate of features to samples $\phi > 1$, the right tail generally plateaus at higher values (i.e., greater than 1) when $\phi$ is closer to 1. This suggests that with a similar number of features and samples, under overparameterization, bias amplification increases. In contrast, when $\phi < 1$, the right tail of the $ODD$ and $EDD$ curves seems to plateau at higher values when $\phi$ is larger. Regardless of the regime of the rate of features to samples $\phi$, the left tails of the $ODD$ and $EDD$ curves are generally higher for larger $\phi$.

### 4.2 REGULARIZATION AND TRAINING DYNAMICS

We now explore how regularization and training dynamics affect bias amplification.

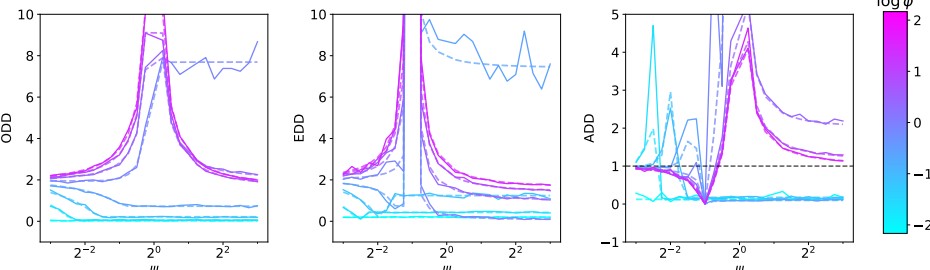

Figure 2: **Our theory predicts that models can amplify bias even with balanced groups and without spurious correlations.** We empirically validate our theory (Theorems 3.1 and 3.2) for $ODD$, $EDD$, and $ADD$ under the setup described in Section 4.1, with $a_1 = 0.5, a_2 = 1, \sigma_1^2 = 1$, and $\sigma_2^2 = 1 \times 10^{-5}$. The solid lines capture empirical values while the corresponding lower-opacity dashed lines represent what our theory predicts. We plot $ODD$ and $EDD$ on the same scale for easy comparison, and include a black dashed line at $ADD = 1$ to contrast bias amplification vs. deamplification. We include all the plots with error bars in Appendix L.

**Setup.** We revisit the experimental setup for Section 4.1. We modulate $a_1, a_2, \psi$ (rate of parameters to samples), as well as $\lambda$ (regularization penalty) to understand the effects of regularization and early stopping on bias amplification. We fix $\sigma_1^2 = \sigma_2^2 = 1$, and the rate of features to samples $\phi = 0.75$.

**Effect of Regularization and Training Time.** In simplistic settings, we can simulate model learning over training time $t$ by setting $\lambda = 1/t$ (Ali et al., 2019). In the figures in Appendix O, we observe that regardless of the regime of $\psi$, $ADD \approx 1$ (i.e., there is neither bias amplification nor deamplification) with high regularization or a short training time. When $\psi > 1$ (i.e., in the overparameterized regime), the $ADD$ is generally greater than 1 across values of $\lambda$ (i.e., bias is amplified), while when $\psi < 1$ (i.e., in the underparameterized regime), the $ADD$ is generally less than 1 (i.e., bias is deamplified). Moreover, when $\psi > 1$, as regularization decreases (or training time increases), bias amplification increases and plateaus. In contrast, when $\psi < 1$, as regularization decreases (or training time increases), bias deamplification increases and plateaus. A notable exception to this trend occurs when $\psi$ is close to 1, where bias is initially deamplified and then amplified as $\lambda$ decreases (or $t$ increases). **This suggests that there may be an optimal regularization penalty or training time to avoid bias amplification and increase bias deamplification.** Intuitively, as training progresses, overparameterized models may discover "shortcut" associations (Geirhos et al., 2020) that do not generalize equally well across groups, yielding bias amplification. In practice, an optimal $\lambda$ or $t$ can be selected by searching for values that strike a desired balance between overall validation error and empirical bias amplification. The search space can be reduced by using the above $ADD$ trends w.r.t. $\lambda$ and $t$ that our theory predicts for over- vs. underparameterized models (see Appendix R for more details). It is important for ML practitioners the consider the interplay between high vs. low feature-to-sample regimes and overparameterization in inducing bias amplification vs. deamplification when selecting optimal hyperparameters (see Figure 1).

In general, the calibration $\lambda = 1/t$ may not yield a theoretically tight picture of how bias evolves with $t$. The use of discrete gradient descent in practice rather than continuous-time gradient flows might yield further discrepancies. However, the calibration $\lambda = 1/t$ yields a ratio of gradient flow to ridge risk that is at most 1.69, with no assumptions on the features $X$ (Ali et al., 2019). Moreover, in the controlled settings considered by Ali et al. (2019), this ratio empirically appears to be quite close to 1, and thus may be sufficient for extrapolating our results. Like us, Jain et al. (2024) and Hall et al. (2022) find that bias and bias amplification can vary substantially during training; future work can establish stronger connections between our observations and the results of Jain et al. (2024), who analytically identify phases in the evolution of bias and a crossing phenomenon in the test error curves of groups during training. However, Jain et al. (2024) do not consider the effect of over- and underparameterization on bias evolution. While our analysis relies on the simplistic calibration $\lambda = 1/t$, it reveals divergent behavior in how bias evolves depending on model size.

**Corroboration on Real Data.** We further investigate the effect of training time on bias amplification on a more realistic dataset. We train a convolutional neural network (CNN) on Colored MNIST (see Appendix K.2 for more details). Colored MNIST is a semi-synthetic dataset derived from MNIST where digits are randomly re-colored to be red or green (Arjovsky et al., 2019). We treat the color of each digit as its group, and we manipulate the groups to have different levels of label noise. In

our experimental protocol: (1) the color of each digit (in both train and test) is chosen uniformly at random (i.e., with probability 0.5) and independently of the label; (2) by default, in the training set, the labels of red digits are flipped with probability 0.05 while the labels of green digits are flipped with probability 0.25; (3) labels are binarized (i.e., digits 0-4 correspond to 0 while digits 5-9 correspond to 1); and (4) each training step constitutes a step of gradient descent based on a batch of 250 instances. Although Colored MNIST is a classification task and we use a complex CNN architecture, **our theory correctly predicts that as the training time $t$ increases, the $ODD$ of the CNN is relatively low while the $EDD$ is much larger**, producing bias deamplification.

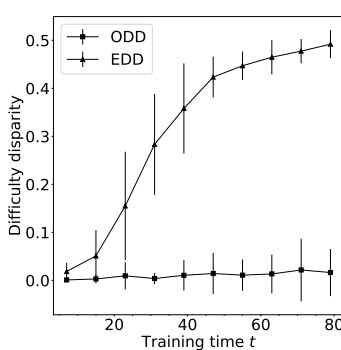

Taking $t \to \infty$ corresponds to the setting of $\lambda \to 0^+$ in our theory (Theorems 3.1, 3.2). Because we assign the colors at random, the only difference in image features between groups is color; therefore, we expect the covariance matrices $\Sigma_1$ and $\Sigma_2$ to roughly coincide and $\Delta = 0$ (i.e., $w_1^* = w_2^*$). Note that we do not make any assumptions about the structure of $\Sigma_1, \Sigma_2$. Furthermore, $p_1 = p_2 = 1/2$, and thus, $\phi_1 = \phi_2$ and $\psi_1 = \psi_2$. Additionally, we analogize the probability of label flipping to label noise in ridge regression. Hence, $e_1 = e_2, u_1 = u_2$. Accordingly, $\lim_{\lambda \to 0^+} B_1(\widehat{f}) = \lim_{\lambda \to 0^+} B_1(\widehat{f}_1) \approx \lim_{\lambda \to 0^+} B_2(\widehat{f}) = \lim_{\lambda \to 0^+} B_2(\widehat{f}_2)$. Simultaneously, $\lim_{\lambda \to 0^+} V_1(\widehat{f}) \approx \lim_{\lambda \to 0^+} V_2(\widehat{f})$. However, $\lim_{\lambda \to 0^+} V_1(\widehat{f}_1) \approx \sigma_1^2/2 \cdot V = 0.05/2 \cdot V = 0.025V$ (where $V = \phi_1 h_1^{(2)}(I_d, \Sigma)$), while $\lim_{\lambda \to 0^+} V_2(\widehat{f}_2) \approx \sigma_2^2/2 \cdot V = 0.25/2 \cdot V = 0.125V$. This results in $ODD \approx 0$ while $EDD \approx 0.1|V|$, which explains the divergence of $ODD$ and $EDD$ in Figure 3. Intuitively, the high label noise for the green digits prohibits the separate model $\widehat{f}_2$ from achieving a low test risk

Figure 3: **Our theory predicts that disparate label noise between groups deamplifies bias on Colored MNIST.** We plot the $ODD$ and $EDD$ of a CNN over training time $t$ for Colored MNIST. As $t$ increases, the $ODD$ is relatively low while the $EDD$ is noticeably higher. The error bars capture the standard deviation computed over 10 random seeds.

compared to $\widehat{f}_1$; the single model $\widehat{f}$ achieves a comparable test risk on both groups, effectively deamplifying bias, because of the better learning signal from the red digits. This phenomenon is similar to *positive transfer*, wherein the $EDD$ of a model generally tends to be higher than the $ODD$ when the labeling rules of imbalanced groups are sufficiently similar (Mannelli et al., 2024). However, Mannelli et al. (2024) do not explore the impact of model size on positive transfer. We show that the $ODD$ can be less than the $EDD$ depending on $\psi$ in Figure 2, where $\Delta = I_d$ (i.e., the groups have different labeling rules). Future work can study the $ADD = \frac{ODD}{EDD}$ profile when $\Delta = 0$. Refer to Appendix P for additional Colored MNIST experiments.

## 5 MINORITY-GROUP BIAS

Recent work has revealed that overparameterization may hurt test performance on minority groups due to spurious features (Sagawa et al., 2020; Khani & Liang, 2021). Our theory provides new insights into how model size and extraneous features affect minority-group bias.

**Setup.** To mirror the settings of Sagawa et al. (2020); Khani & Liang (2021), we consider diatomic covariance matrices of *core* and *extraneous* features. We define $A \oplus B = \begin{pmatrix} A & 0 \\ 0 & B \end{pmatrix}$, and choose $\Sigma_1 = a_1 I_{\pi d} \oplus 0 I_{(1-\pi)d}, \Sigma_2 = a_2 I_{\pi d} \oplus b_2 I_{(1-\pi)d}$, for $\pi \in (0,1)$, $a_1, b_2 > 0$, $a_1 = a_2$. Refer to App. K.1 for full details and a discussion of extraneous vs. spurious features (due to space limitations).

**Interpolation Thresholds.** The together $R_2$ (i.e., the test risk for the minority group in the single model setting) has different interpolation thresholds as $\psi$ (rate of parameters to samples) increases, depending on $\phi$ (rate of features to samples) and $\pi$ (fraction of core features). Notably, as $\phi$ increases, the interpolation thresholds occur at larger model sizes, culminating at $\psi = 1$. This suggests that for a higher rate of features to samples, a larger model size can greatly increase the together test risk of the minority group. Furthermore, the interpolation thresholds all occur closer to $\psi = 1$ for larger $\pi$, collapsing to a single threshold at $\psi = 1$ when $\pi \to 1$ (as in Appendix L). Therefore, a lower fraction

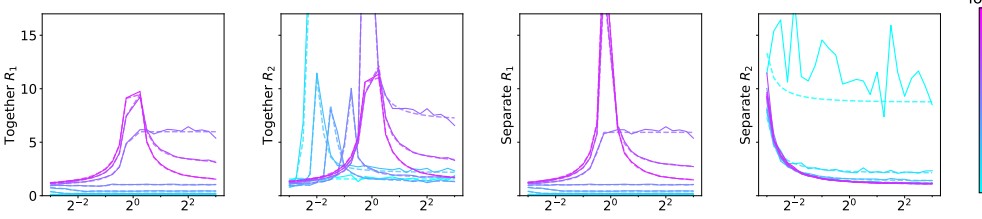

Figure 4: **Minority-group test risk can peak with different model sizes depending on the rate of features to samples.** We empirically demonstrate that minority-group bias is affected by extraneous features. We validate our theory (Theorems 3.1 and 3.2) for together $R_1, R_2$ (i.e., single model learned for both groups) and separate $R_1, R_2$ (i.e., separate model learned per group) under the setup described in Section 4.2, with $a_1 = 2, b_2 = 0.2$, and $\pi = 0.5$. The solid lines capture empirical values while the corresponding lower-opacity dashed lines represent what our theory predicts. We include a black dashed line at $ADD = 1$ to contrast bias amplification vs. deamplification. All y-axes are on the same scale for easy comparison. All the plots with error bars are in Appendix Q.

of core features can yield more possible model sizes that increase the test risk of the minority group. In addition, the together $R_2$ exhibits a steeper rate of growth around the interpolation thresholds for larger $b_2$, suggesting that a higher variance in the extraneous features can also increase the test risk of the minority group in the single model setting. The phenomenon of different interpolation thresholds is not visible for $R_2$ when a separate model is trained per group; however, we do observe the expected double descent peaks in the separate $R_1$ and $R_2$ curves at $\psi_1 = 1$ and $\psi_2 = 1$, respectively.

**Overparameterization.** The right tails of the together $R_2$ curves plateau at different finite values depending on $\phi$. In particular, for $\phi$ closer to 1, the together $R_2$ curves generally plateau at a higher value, suggesting that a similar number of features and samples can exacerbate minority-group bias under overparameterization. Furthermore, for smaller $\pi$ and certain values of $\phi < 1$, the right tail of the together $R_1$ curve plateaus at a lower value than the together $R_2$ curve. This suggests that there can be differences in test error between groups that are not alleviated even with increased model size. This phenomenon diminishes in magnitude as the fraction of core features increases. This phenomenon supports the finding of Sagawa et al. (2020) that **overparameterization with spurious features can increase test risk disparities between groups**. We identify that the magnitude of this phenomenon may **depend on both the rate of features to samples and fraction of core features**.

## 6    CONCLUSION

We present a unifying, rigorous, and effective theory of ML bias in the settings of ridge regression with and without random projections. Our theory predicts interesting insights into bias amplification and minority-group bias in different feature and parameter regimes. These findings can inform strategies to evaluate and mitigate unfairness in ML (see Appendix R for more details). However, there remain practical challenges to assessing whether a model is prone to bias amplification. These include robustly estimating the feature covariance matrices (Bickel & Levina, 2008) and label noises (Frénay & Kabán, 2014) for groups from sample data, especially for minority groups which have limited data. Even so, practitioners can use our theory and empirical observations to form intuition about when *disparities* in the variability of features and labels across groups can amplify bias.

Our theoretical methods are easily extendable to the case of more than two groups and can accommodate label noise sampled from other distributions. However, our theory is not directly extendable to different proportionate scaling limits (e.g., $d^2/n$ has a finite limit instead of $d/n$). Additionally, our theory requires approximately normally-distributed data and thus does not currently account for missing features, which are common in the real world (Feng et al., 2024). Furthermore, our theory implicitly assumes that group information is known, which is not always true (Coston et al., 2019); however, because we work in an asymptotic scaling limit, having access to group information with $o(\min(n_1, n_2))$ noise is sufficient. As future work, we can leverage "Gaussian equivalents" (Goldt et al., 2022) to extend our theory to wide, fully-trained networks in the NTK (Jacot et al., 2018) and lazy (Chizat et al., 2019) regimes; this will enable us to understand how, apart from model size, other design choices like nonlinear activation functions and learning rate may affect bias amplification.

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

# Appendix

## Table of Contents

## A WARM-UP: DERIVING MARCHENKO-PASTUR LAW VIA OPERATOR-VALUED FREE PROBABILITY THEORY

We provide a detailed example of how to apply linear pencils and operator-valued free probability theory (OVFPT) to derive the classical Marchenko-Pastur (MP) law (Marčenko & Pastur, 1967). Let $S = (1/n)X^\top X \in \mathbb{R}^{d \times d}$ be the empirical covariance matrix for an $n \times d$ random matrix $X$ with IID entries from $\mathcal{N}(0, 1)$. If $n$ tends to infinity while $d$ is held fixed, then $S$ converges to the population covariance matrix, here $\Sigma = I_d$. If $d$ also tends to infinity, then the limit seizes to exist. It turns out that one can still make sense of the limiting distribution of eigenvalues of $S$ in the case $d/n$ stays constant, i.e.,

$$n, d \to \infty, \ d/n \to \gamma \in (0, \infty). \tag{22}$$

In particular, we seek to understand the behavior of the random histogram:

$$\widehat{\mu}_n = \frac{1}{d} \sum_{i=1}^{n} \widehat{\lambda}_i, \tag{23}$$

where $\widehat{\lambda}_1, \ldots, \widehat{\lambda}_d$ are the eigenvalues of $S$. In the aforementioned limit, $\widehat{\mu}_n$ converges to a deterministic law $\mu_{\mathrm{MP}}$ on $\mathbb{R}$ called the MP law. This is central to the field of random matrix theory (RMT), a primary tool in probability theory, statistical analysis of neural networks, finance, etc. We are interested in an even more powerful tool – free probability theory (FPT) – which is powerful enough to give a precise picture of deep learning in certain linearized regimes (e.g., random features, NTK) and interesting phenomena (e.g., triple descent) via analytic calculation.

### A.1 STEP 1: CONSTRUCTING A LINEAR PENCIL

For any positive $\lambda$, consider the $2 \times 2$ block matrix $Q$ defined by:

$$Q = \begin{pmatrix} I_n & -\frac{X}{\sqrt{n\lambda}} \\ \frac{X^\top}{\sqrt{n\lambda}} & I_d \end{pmatrix}. \tag{24}$$

Let $\bar{\mathrm{tr}}$ be the normalized trace operator on square matrices and set $\varphi = \mathbb{E} \circ \bar{\mathrm{tr}}$. This gives random $(n + d) \times (n + d)$ matrices the structure of a von Neumann algebra $\mathcal{A}$. Define a $2 \times 2$ matrix $G = G(Q)$ by:

$$G = (I_2 \otimes \varphi)Q^{-1}, \text{ i.e } g_{i,j} = \varphi([Q^{-1}]_{i,j}) = [\varphi(Q^{-1})]_{i,j} \text{ for all } i, j \in \{1, 2\}. \tag{25}$$

> Thus, the operator $(I_2 \otimes \varphi)Q^{-1}$ extracts the expectation of the normalized trace of the blocks of the inverse of the a $2 \times 2$ block matrix $Q$.

Observe that:

$$\mathbb{E} \bar{\mathrm{tr}} (S + \lambda I_d)^{-1} = \frac{g_{2,2}}{\lambda}. \tag{26}$$

This is a direct consequence of inverting a $2 \times 2$ block matrix (namely Schur's complement). The mechanical advantage of Equation 26 is that the *resolvent* $(S + \lambda I_d)^{-1}$ depends quadratically on $X$ while $g_{2,2}$ is defined via $Q$, which is linear in $X$. For this reason, $Q$ is called a *linear pencil* for $(S + \lambda I_d)^{-1}$. The construction of appropriate linear pencils for rational functions of random matrices is a crucial step in leveraging FPT.

### A.2 STEP 2: CONSTRUCTING THE FUNDAMENTAL EQUATION VIA FREENESS

For any $B \in M_b(\mathbb{C})^+$, define a block matrix $B \otimes 1_\mathcal{A}$ by:

$$[B \otimes 1_\mathcal{A}]_{ij} = \begin{cases} b_{ij} I_{d_i}, & \text{if } d_i = d_j \\ 0, & \text{else} \end{cases}. \tag{27}$$

Here, $b \times b$ is the number of blocks in the linear pencil $Q_X$, that is, $b = 2$. Now, observe that we can write $Q = F - Q_X$, where:

$$F = \begin{pmatrix} I_d & 0 \\ 0 & I_n \end{pmatrix} = I_2 \otimes 1_{\mathcal{A}} \text{ and } Q_X = \begin{pmatrix} 0 & \frac{X}{\sqrt{n\lambda}} \\ -\frac{X^\top}{\sqrt{n\lambda}} & 0 \end{pmatrix}. \tag{28}$$

One can then express $G = (I_b \otimes \varphi)Q^{-1} = (I_b \otimes \varphi)(F - Q_X)^{-1}$. From operator-valued FPT, we know that in the proportionate scaling limit given by Equation 22, the following fixed-point equation (due to the asymptotic freeness of $Q_X$ and $F$) is satisfied by $G$:

$$G = (I_b \otimes \varphi)(F - R \otimes 1_{\mathcal{A}})^{-1}, \tag{29}$$

where $R = \mathcal{R}_{Q_X}(G)$, and $R_{Q_X}$ is the R-transform of $Q_X$ which maps $M_b(\mathbb{C})^+$ to itself like so:

$$\mathcal{R}_{Q_X}(B)_{ij} = \sum_{k,\ell} \sigma(i, k; \ell, j)\alpha_k b_{k\ell}. \tag{30}$$

Here, $\sigma(i, k; \ell, j)$ is the covariance between the entries of block $(i, k)$ and block $(\ell, j)$ of $Q_X$, while $\alpha_k$ is the dimension of the block $(k, \ell)$.

### A.3 STEP 3: THE FINAL CALCULATION

By the structure of $Q_X$, one can compute from Equation 30:

$$r_{1,1} = d \cdot \frac{-1}{n\lambda} g_{2,2} = -\frac{\gamma}{\lambda} g_{2,2}, \tag{31}$$

$$r_{1,2} = 0, \tag{32}$$

$$r_{2,1} = 0, \tag{33}$$

$$r_{2,2} = n \cdot \frac{-1}{n\lambda} g_{1,1} = -\frac{1}{\lambda} g_{1,1}. \tag{34}$$

Combining this with Equation 29, one has:

$$G = (I_2 \otimes \varphi)(Z - R \otimes 1_{\mathcal{A}})^{-1} = (I_2 - R)^{-1} = \begin{pmatrix} 1 + (\gamma/\lambda)g_{2,2} & 0 \\ 0 & 1 + g_{2,2}/\lambda \end{pmatrix}^{-1}$$
$$= \begin{pmatrix} \lambda/(\lambda + \gamma g_{2,2}) & 0 \\ 0 & \lambda/(\lambda + g_{1,1}) \end{pmatrix}. \tag{35}$$

Comparing the matrix entries, this translates to the following scalar equations:

$$g_{1,1} = \frac{\lambda}{\lambda + \gamma g_{2,2}}, \tag{36}$$

$$g_{2,2} = \frac{\lambda}{\lambda + g_{1,1}}, \tag{37}$$

$$g_{2,1} = g_{1,2} = 0. \tag{38}$$

Plugging the second equation into the first (to eliminate $g_{1,1}$) gives:

$$g_{2,2} = \frac{\lambda}{\lambda + \lambda/(\lambda + \gamma g_{2,2})}.$$

Setting $m = g_{2,2}/\lambda$ then gives $m = (\lambda + 1/(1 + \gamma m))^{-1}$, i.e.,

$$\frac{1}{m} = \lambda + \frac{1}{1 + \gamma m}, \tag{39}$$

which is precisely the functional equation characterizing the Stieltjes transform (evaluated at $\lambda = -z$) of the MP law with shape parameter $\gamma$. By treating $\lambda$ as a complex number and applying the Cauchy-inversion formula, we can recover $\mu_{\text{MP}}$.

# B  TECHNICAL ASSUMPTIONS

**Assumption B.1.** *In the case of classical ridge regression, we will work in the following proportionate scaling limit:*

$$n, n_1, n_2, d \to \infty, \quad n_1/n \to p_1, \, n_2/n \to p_2, \quad d/n_1 \to \phi_1, \, d/n_2 \to \phi_2, \, d/n \to \phi, \quad (40)$$

*for some constants $\phi_1, \phi_2, \phi \in (0, \infty)$. The scalar $\phi$ captures the rate of features to samples. Observe that $\phi = p_1 \phi_1$ and $\phi = p_2 \phi_2$.*

**Assumption B.2.** *The per-group covariance matrices $\Sigma_1$ and $\Sigma_2$ and ground-truth weight covariance matrices $\Theta$ and $\Delta$ are all simultaneously diagonalizable; hence, all these matrices commute.*

While Assumption B.2 may appear reductive, our goal is to analyze the bias amplification phenomenon in a sufficient setting that does not introduce complexities due to non-commutativity. Notably, our main theoretical result does not assume isotropic covariance. For example, our theory accommodates diatomic covariance (see Section 5) and power-law covariance (see Appendix M).

**Assumption B.3.** *In Corollary M.1, we assume the following spectral densities exist when $d \to \infty$:*

- $\nu \in \mathcal{P}(\mathbb{R}_+)$ *is the limiting spectral density of $\Sigma_2 \Sigma_1^{-1}$, of the ratios $\lambda_j^{(2)}/\lambda_j^{(1)}$ of the eigenvalues of the respective covariance matrices,*

- $\mu \in \mathcal{P}(\mathbb{R}_+, \mathbb{R}_+)$ *is the joint limiting density of the spectra of $\Sigma_2 \Sigma_1^{-1}$ and $\Sigma_1$,*

- $\pi \in \mathcal{P}(\mathbb{R}_+)$ *is the limiting density of the spectrum of $\Delta$.*

## C    RELATED WORK (CONTINUED)

**High-dimensional analysis of bias.**    Mannelli et al. (2024) employ the replica method, which is non-rigorous, while we use OVFPT, which is entirely rigorous. Moreover, Mannelli et al. (2024); Jain et al. (2024) study the application of linear classification to Gaussian data with isotropic covariance; in contrast, we study the application of regression with random projections (a simplified model of feedforward neural networks) to Gaussian data with more general covariance structure (i.e., covariance matrices that are simultaneously diagonalizable) and noisy labels. This allows us to analyze the effects of these additional factors on bias. We make additional connections between our work and Mannelli et al. (2024); Jain et al. (2024) in Section 4.2.

**Bias amplification metrics.**    Our definition of $ADD$ is consistent with the conceptualization of bias of Bell & Sagun (2023). At a high level, our definition quantifies how many times worse model bias would be if a ML practitioner opted to train a single model on a mixture of data from two groups (i.e., the setting in which bias is observed in practice) vs. separate models for the data from each group (i.e., the setting which corresponds to the bias in the data alone, and thus the a priori amount of bias we would expect in the case of a single model). In sum, we seek to isolate the contribution of the *model* to bias when learning from data with different groups.

# D WARM-UP: CLASSICAL LINEAR MODEL

**Technical Difficulty.** The analysis of the test errors (e.g., $R_s(\widehat{f})$) amounts to the analysis of the trace of rational functions of sums of random matrices. Although the limiting spectral density of sums of random matrices is a classical computation using subordination techniques (Marčenko & Pastur, 1967; Kargin, 2015), a more involved analysis is required in our case. This difficulty is even greater in the setting of random projections (see Section 3.1). Thus, we employ OVFPT to compute the exact high-dimensional limits of such quantities. We derive Theorems D.2 and D.1 using OVFPT (in Appendices F and E). Theorem D.1 is a non-trivial generalization of Proposition 3 from (Bach, 2023), which can be recovered by taking $p_s \to 1$ (i.e., $p_{s'} \to 0$).

## D.1 SINGLE MODEL LEARNED FOR BOTH GROUPS

We first consider the classical ridge regression model $\widehat{f}$, which is learned using empirical risk minimization and $\ell_2$-regularization with penalty $\lambda$. The parameter vector $\widehat{w} \in \mathbb{R}^d$ of the linear model $\widehat{f}$ is given by the following problem:

$$\widehat{w} = \arg\min_{w \in \mathbb{R}^d} L(w) = \sum_{s=1}^{2} n^{-1} \|X_s w - Y_s\|_2^2 + \lambda \|w\|_2^2. \tag{41}$$

The unregularized limit $\lambda \to 0^+$ corresponds to ordinary least-squares (OLS). We provide in Theorem D.1 a novel bias-variance decomposition for the test error $R_s(\widehat{f})$ for each group $s \in \{1, 2\}$. We first present some relevant definitions.

**Definition D.1.** *For any group index $s \in \{1, 2\}$, we define $(e_1, e_2, u_1^{(s)}, u_2^{(s)})$ to be the unique positive solution to the following system of fixed-point equations:*

$$1/e_s = 1 + \phi \bar{\mathrm{tr}}\, \Sigma_s K^{-1}, \quad u_k^{(s)} = \phi e_k^2 \bar{\mathrm{tr}}\, \Sigma_k (p_1 u_1^{(s)} \Sigma_1 + p_2 u_2^{(s)} \Sigma_2 + \Sigma_s) K^{-2}, \ k \in \{1, 2\}, \tag{42}$$

*where $K = p_1 e_1 \Sigma_1 + p_2 e_2 \Sigma_2 + \lambda I_d$ and $\bar{\mathrm{tr}}\, A := (1/d)\,\mathrm{tr}\, A$ is the normalized trace operator.*

The fixed-point equations for $e_s$ are non-linear and often not analytically solvable for general $\Sigma_1, \Sigma_2$. This is typical in RMT.

**Theorem D.1.** *Under Assumptions B.2 and B.1, it holds that: $R_s(\widehat{f}) \simeq B_s(\widehat{f}) + V_s(\widehat{f})$, with*

$$V_s(\widehat{f}) = V_s^{(1)}(\widehat{f}) + V_s^{(2)}(\widehat{f}), \tag{43}$$

$$V_s^{(k)}(\widehat{f}) = p_k \sigma_k^2 \phi \bar{\mathrm{tr}}\, \Sigma_k \big(e_k \Sigma_s - \lambda u_k^{(s)} I_d + p_{k'} \Sigma_{k'} (e_k u_{k'}^{(s)} - e_{k'} u_k^{(s)})\big) K^{-2}, \tag{44}$$

$$B_s(\widehat{f}) = B_s^{(1)}(\widehat{f}) + B_s^{(3)}(\widehat{f}) + \begin{cases} 0, & s = 1, \\ 2B_2^{(2)}(\widehat{f}), & s = 2, \end{cases} \tag{45}$$

$$B_s^{(1)}(\widehat{f}) = p_{s'} \bar{\mathrm{tr}}\, \Delta \Sigma_{s'} (p_{s'}(1 + p_s u_s^{(s)}) e_{s'}^2 \Sigma_{s'} \Sigma_s + u_{s'}^{(s)} (p_s e_s \Sigma_s + \lambda I_d)^2) K^{-2}, \tag{46}$$

$$B_2^{(2)}(\widehat{f}) = p_1 \lambda \bar{\mathrm{tr}}\, \Sigma_1 ((1 + p_2 u_2^{(2)}) e_1 \Sigma_2 - u_1^{(2)} (p_2 e_2 \Sigma_2 + \lambda I_d)) K^{-2}, \tag{47}$$

$$B_s^{(3)}(\widehat{f}) = \lambda^2 \bar{\mathrm{tr}}\, \Theta_s (p_1 u_1^{(s)} \Sigma_1 + p_2 u_2^{(s)} \Sigma_2 + \Sigma_s) K^{-2}, \tag{48}$$

*where $1' = 2$ and $2' = 1$.*

## D.2 SEPARATE MODEL LEARNED PER GROUP

We now treat the case of fitting a separate model $\widehat{f}_s$ per group. Suppose that the classical ridge regression models $\widehat{f}_1$ and $\widehat{f}_2$ are learned using empirical risk minimization and $\ell_2$-regularization with penalties $\lambda_1$ and $\lambda_2$, respectively. In particular, we have the following optimization problem for each group $s$:

$$\arg\min_{w \in \mathbb{R}^d} L(w) = \frac{1}{n_s} \sum_{(x_i, y_i) \in \mathcal{D}^s} (x_i^\top w - y_i)^2 + \lambda_s \|w\|_2^2 = \frac{\|X_s w - Y_s\|_2^2}{n_s} + \lambda_s \|w\|_2^2. \tag{49}$$

We first present some relevant definitions.

**Definition D.2.** *Let* $\bar{\mathrm{df}}_m^{(s)}(t) = \bar{\mathrm{tr}}\,\Sigma_s^m \left(\Sigma_s + tI_d\right)^{-m}$, *and $\kappa_s$ be the unique positive solution to the equation $\kappa_s - \lambda_s = \kappa_s \phi_s \bar{\mathrm{df}}_1^{(s)}(\kappa_s)$.*

In this setting, we deduce Theorem D.2.

**Theorem D.2.** *Under Assumptions B.2 and B.1, it holds that:*

$$R_s(\widehat{f}_s) \simeq B_s(\widehat{f}_s) + V_s(\widehat{f}_s), \ with \tag{50}$$

$$V_s(\widehat{f}_s) = \frac{\sigma_s^2 \phi_s \bar{\mathrm{df}}_2^{(s)}(\kappa_s)}{1 - \phi_s \bar{\mathrm{df}}_2^{(s)}(\kappa_s)}, \ B_s(\widehat{f}_s) = \frac{\kappa_s^2 \bar{\mathrm{tr}}\,\Theta_s \Sigma_s \left(\Sigma_s + \kappa_s I_d\right)^{-2}}{1 - \phi_s \bar{\mathrm{df}}_2^{(s)}(\kappa_s)}. \tag{51}$$

### D.3    PHASE DIAGRAM

We present the bias amplification phase diagram (Figure 5) predicted by Theorems D.1 and D.2 for the classical ridge regression model. The phase diagram offers insights into how $\phi$ (rate of features to samples) affects bias amplification. To obtain the precise phase diagram, we solve the scalar equations numerically. In the $ODD$ profile, we observe an interpolation threshold at $\phi = 1$. To the right of the threshold, we observe a tail that descends towards 1. To the left of the threshold, the $ODD$ descends below 1 with a local minimum at $\phi \approx 0.25$ before increasing. In contrast, we observe that the $EDD$ continually grows as $\phi$ increases, ascending from a small value, exhibiting an inflection point at $\phi = 0.5$, and plateauing after $\phi = 1$. Accordingly, the $ADD$ increases significantly as $\phi$ decreases (with an intermediate inflection point at $\phi = 0.5$), peaks at $\phi = 1$, and descends towards 1 as $\phi$ increases (i.e., bias remains amplified in this phase). In sum, bias is most amplified when the rate of features to samples $\phi \ll 1$ and $\phi = 1$. Interestingly, bias amplification consistently occurs (i.e., $ADD > 1$) across all observed values of $\phi$.

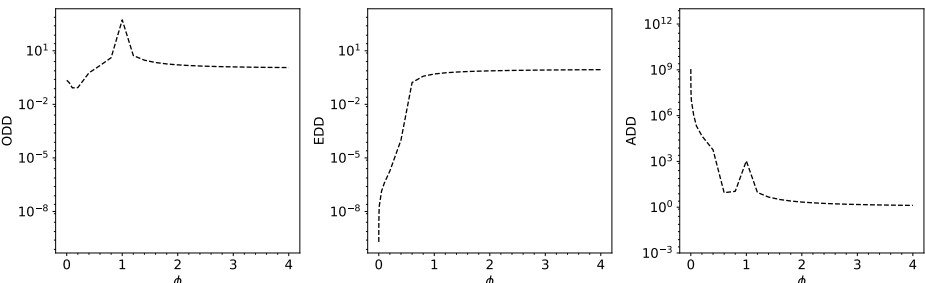

Figure 5: $ODD$**,** $EDD$**, and** $ADD$ **phase diagrams for classical ridge regression.** We plot the bias amplification phase diagrams with respect to $\phi$ (rate of features to samples), as predicted by our theory for ridge regression without random projections (Theorems D.1, D.2). Dashed black lines indicate theoretical predictions. We consider isotropic covariance matrices: $\Sigma_1 = 2I_d, \Sigma_2 = I_d, \Theta = 2I_d$, $\Delta = I_d$. Additionally, $n = 1 \times 10^4, \sigma_1^2 = \sigma_2^2 = 1$. We further choose $\lambda = \lambda_1 = \lambda_2 = 1 \times 10^{-6}$ to approximate the minimum-norm interpolator. We observe that bias amplification can occur even in the balanced data setting, i.e., when $p_1 = p_2 = 1/2$, without spurious correlations.

# E  PROOF OF THEOREM D.2

*Proof.* We define $M_s = X_s^\top X_s$ and $E_s = Y_s - X_s w_s^*$. Note that $\widehat{w}_s = (X_s^\top X_s + n_s \lambda_s I_d)^{-1} X_s^\top (X_s w_s^* + E_s) = (M_s + n_s \lambda_s I_d)^{-1} M_s w_s^* + (M_s + n_s \lambda_s I_d)^{-1} X_s^\top E_s$. We deduce that $R_s(\widehat{f}_s) = B_s(\widehat{f}_s) + V_s(\widehat{f}_s)$, where:

$$B_s(\widehat{f}_s) = \mathbb{E} \, \|(M_s + n_s \lambda_s I_d)^{-1} M_s w_s^* - w_s^*\|_{\Sigma_s}^2, \tag{52}$$

$$V_s(\widehat{f}_s) = \mathbb{E} \, \|(M_s + n_s \lambda_s I_d)^{-1} X_s^\top E_s\|_{\Sigma_s}^2. \tag{53}$$

## E.1  VARIANCE TERM

Note that the variance term $V_s(\widehat{f})$ of the test error of $\widehat{f}_s$ evaluated on group $s$ is given by:

$$V_s(\widehat{f}_s) = \sigma_s^2 \mathbb{E} \, \text{tr} \, X_s (M_s + n_s \lambda_s I_d)^{-1} \Sigma_s (M_s + n_s \lambda_s I_d)^{-1} X_s^\top \tag{54}$$

$$= \sigma_s^2 \mathbb{E} \, \text{tr} \, (M_s + n_s \lambda_s I_d)^{-1} M_s (M_s + n_s \lambda_s I_d)^{-1} \Sigma_s. \tag{55}$$

We can re-express this as:

$$n_s V_s(\widehat{f}_s) = \sigma_s^2 \mathbb{E} \, \text{tr} \, (H_s + \lambda_s I_d)^{-1} H_s (H_s + \lambda_s I_d)^{-1} \Sigma_s \tag{56}$$

$$= \frac{\sigma_s^2}{\lambda_s} \mathbb{E} \, \text{tr} \, (H_s/\lambda_s + I_d)^{-1} (H_s/\lambda_s)(H_s/\lambda_s + I_d)^{-1} \Sigma_s, \tag{57}$$

where $H_s = X_s^\top X_s / n_s$ and $X_s = Z_s \Sigma_s^{1/2}$, with $Z_1 \in \mathbb{R}^{n_1 \times d}$ and $Z_2 \in \mathbb{R}^{n_2 \times d}$ being independent random matrices with IID entries from $\mathcal{N}(0, 1)$. Thus, the variance term is proportional to:

$$\bar{\text{tr}} \, (H_s + \lambda_s I_d)^{-1} H_s (H_s + \lambda_s I_d)^{-1} \Sigma_s. \tag{58}$$

WLOG, we consider the case where $s = 1$. The matrix of interest has a linear pencil representation given by (with zero-based indexing):

$$(H_1/\lambda_1 + I_d)^{-1}(H_1/\lambda_1)(H_1/\lambda_1 + I_d)^{-1}\Sigma_1 = Q_{0,8}^{-1}, \tag{59}$$

where the linear pencil $Q$ is defined as follows:

$$Q = \begin{pmatrix}
I_d & \Sigma_1^{\frac{1}{2}} & 0 & 0 & -\Sigma_1^{\frac{1}{2}} & 0 & 0 & 0 & 0 \\
0 & I_d & -\frac{1}{\sqrt{\lambda_1}\sqrt{n_1}}Z_1^\top & 0 & 0 & 0 & 0 & 0 & 0 \\
0 & 0 & I_{n_1} & -\frac{1}{\sqrt{\lambda_1}\sqrt{n_1}}Z_1 & 0 & 0 & 0 & 0 & 0 \\
-\Sigma_1^{\frac{1}{2}} & 0 & 0 & I_d & 0 & 0 & 0 & 0 & 0 \\
0 & 0 & 0 & 0 & I_d & -\frac{1}{\sqrt{\lambda_1}\sqrt{n_1}}Z_1^\top & 0 & 0 & 0 \\
0 & 0 & 0 & 0 & 0 & I_{n_1} & -\frac{1}{\sqrt{\lambda_1}\sqrt{n_1}}Z_1 & 0 & 0 \\
0 & 0 & 0 & 0 & 0 & 0 & I_d & -\Sigma_1^{\frac{1}{2}} & 0 \\
0 & 0 & 0 & 0 & \Sigma_1^{\frac{1}{2}} & 0 & 0 & I_d & -\Sigma_1 \\
0 & 0 & 0 & 0 & 0 & 0 & 0 & 0 & I_d
\end{pmatrix}. \tag{60}$$

We compute $Q$ using the `NCMinimalDescriptorRealization` function of the NCAlgebra library[1]. We further symmetrize $Q$ by constructing the self-adjoint matrix $\overline{Q}$:

$$\overline{Q} = \begin{pmatrix} 0 & Q^\top \\ Q & 0 \end{pmatrix}. \tag{61}$$

This enables us to apply known formulae for the $R$-transform of Gaussian block matrices (Far et al., 2006). We note that $\overline{Q}_{0,17}^{-1} = Q_{0,8}^{-1}$. Taking similar steps as Lee et al. (2023), we use OVFPT on $\overline{Q}$. Let $G = (I_{18} \otimes \mathbb{E} \bar{\text{tr}}) \overline{Q}^{-1} \in \mathbb{R}^{18 \times 18}$ be the matrix whose entries are normalized traces of blocks[2] of $\overline{Q}^{-1}$. We provide a detailed example of how to apply OVFPT to derive the MP law in Appendix A. One can arrive at that, in the asymptotic limit given by Equation 40, the following holds:

$$\mathbb{E}\bar{\text{tr}} \, (H_1 + \lambda_1 I_d)^{-1} H_1 (H_1 + \lambda_1 I_d)^{-1} \Sigma_1 = \frac{G_{0,17}}{\lambda_1},$$

$$\text{with } \frac{G_{0,17}}{\lambda_1} = (G_{5,14} - G_{2,14})\bar{\text{tr}} \, (\Sigma_1 G_{2,11} + \lambda_1 I_d)^{-1} \Sigma_1 \, (\Sigma_1 G_{5,14} + \lambda_1 I_d)^{-1} \Sigma_1. \tag{62}$$

---

[1] https://github.com/NCAlgebra/NC
[2] By convention, the trace of a non-square block is zero.

We will now obtain the fixed-point equations satisfied by $G_{2,11}$ and $G_{5,14}$. We observe that:

$$G_{2,11} = -\frac{\lambda_1}{-\lambda_1 + \phi_1 G_{3,10}}, \quad G_{3,10} = -\lambda_1 \bar{\text{tr}}\, \Sigma_1 (\Sigma_1 G_{2,11} + \lambda_1 I_d)^{-1} \tag{63}$$

$$\implies G_{2,11} = \frac{1}{1 + \phi_1 \bar{\text{tr}}\, \Sigma_1 (\Sigma_1 G_{2,11} + \lambda_1 I_d)^{-1}}, \tag{64}$$

$$G_{5,14} = -\frac{\lambda_1}{-\lambda_1 + \phi_1 G_{6,13}}, \quad G_{6,13} = -\lambda_1 \bar{\text{tr}}\, \Sigma_1 (\Sigma_1 G_{5,14} + \lambda_1 I_d)^{-1} \tag{65}$$

$$\implies G_{5,14} = \frac{1}{1 + \phi_1 \bar{\text{tr}}\, \Sigma_1 (\Sigma_1 G_{5,14} + \lambda_1 I_d)^{-1}}. \tag{66}$$

We recognize that we must have the identification $e_1 = G_{2,11} = G_{5,14}$, where $e_1 \geq 0$. Therefore:

$$e_1 = \frac{e_1}{e_1 + \phi_1 \bar{\text{df}}_1^{(1)}(\lambda_1/e_1)} \tag{67}$$

$$\text{i.e., } 1 = e_1 + \phi_1 \bar{\text{df}}_1^{(1)}(\lambda_1/e_1) = \lambda_1/\kappa_1 + \phi_1 \bar{\text{df}}_1^{(1)}(\kappa_1) \tag{68}$$

$$\kappa_1 = \lambda_1 + \kappa_1 \phi_1 \bar{\text{df}}_1^{(1)}(\kappa_1), \tag{69}$$

where $\bar{\text{df}}_m^{(s)}(t) = \bar{\text{tr}}\, \Sigma_s^m (\Sigma_s + tI_d)^{-m}$ and $\kappa_1 = \lambda_1/e_1$. Additionally:

$$G_{2,14} = \frac{\lambda_1 \phi_1 G_{3,13}}{(-\lambda_1 + \phi_1 G_{3,10})(-\lambda_1 + \phi_1 G_{6,13})} = \phi_1 e_1^2 \frac{G_{3,13}}{\lambda_1}, \tag{70}$$

$$\frac{G_{3,13}}{\lambda_1} = \bar{\text{tr}}\, (\Sigma_1 G_{2,11} + \lambda_1 I_d)^{-2}(\Sigma_1 G_{2,14} + \lambda_1 I_d)\Sigma_1 \tag{71}$$

$$= \frac{G_{2,14}}{e_1^2} \bar{\text{df}}_2^{(1)}(\kappa_1) + \lambda_1 \bar{\text{tr}}\, (\Sigma_1 e_1 + \lambda_1 I_d)^{-2}\Sigma_1, \tag{72}$$

$$\frac{G_{3,10}}{\lambda_1} = -\bar{\text{tr}}\, (\Sigma_1 e_1 + \lambda_1 I_d)^{-1}\Sigma_1. \tag{73}$$

Then:

$$G_{5,14} - G_{2,14} = e_1^2 \left(1 - \phi_1 \frac{G_{3,10} + G_{3,13}}{\lambda_1}\right), \tag{74}$$

$$\frac{G_{3,10} + G_{3,13}}{\lambda_1} = \frac{G_{2,14}}{e_1^2} \bar{\text{df}}_2^{(1)}(\kappa_1) + \lambda_1 \bar{\text{tr}}\, (\Sigma_1 e_1 + \lambda_1 I_d)^{-2}\Sigma_1 \tag{75}$$

$$- \bar{\text{tr}}\, (\Sigma_1 e_1 + \lambda_1 I_d)^{-2}(\Sigma_1 e_1 + \lambda_1 I_d)\Sigma_1 \tag{76}$$

$$= \frac{G_{2,14}}{e_1^2} \bar{\text{df}}_2^{(1)}(\kappa_1) - \frac{e_1}{e_1^2} \bar{\text{df}}_2^{(1)}(\kappa_1) \tag{77}$$

$$= -\frac{G_{5,14} - G_{2,14}}{e_1^2} \bar{\text{df}}_2^{(1)}(\kappa_1). \tag{78}$$

We define:

$$c_1 \geq 1, c_1 = \frac{G_{5,14} - G_{2,14}}{e_1^2} = 1 + \phi_1 c_1 \bar{\text{df}}_2^{(1)}(\kappa_1), \tag{79}$$

$$\text{i.e., } c_1 = \frac{1}{1 - \phi_1 \bar{\text{df}}_2^{(1)}(\kappa_1)}. \tag{80}$$

Hence:

$$\frac{G_{0,17}}{\lambda_1} = c_1 \bar{\text{df}}_2^{(1)}(\kappa_1) = \frac{\bar{\text{df}}_2^{(1)}(\kappa_1)}{1 - \phi_1 \bar{\text{df}}_2^{(1)}(\kappa_1)}. \tag{81}$$

In conclusion:

$$\kappa_1 = \lambda_1 + \kappa_1 \phi_1 \bar{\text{df}}_1^{(1)}(\kappa_1), \tag{82}$$

$$V_1(\hat{f}_1) = \frac{\sigma_1^2 \phi_1 \bar{\text{df}}_2^{(1)}(\kappa_1)}{1 - \phi_1 \bar{\text{df}}_2^{(1)}(\kappa_1)}. \tag{83}$$

Following similar steps for $V_2(\widehat{f}_2)$, we get:

$$\kappa_2 = \lambda_2 + \kappa_2\phi_2\bar{\mathrm{df}}_1^{(2)}(\kappa_2), \tag{84}$$

$$V_2(\widehat{f}_2) = \frac{\sigma_2^2\phi_2\bar{\mathrm{df}}_2^{(2)}(\kappa_2)}{1 - \phi_2\bar{\mathrm{df}}_2^{(2)}(\kappa_2)}. \tag{85}$$

To further substantiate our result, let us consider the unregularized case where $\lambda_s = 0$ and $\phi_s < 1$:

$$\kappa_s = 0, V_s(\widehat{f}_s) = \frac{\sigma_s^2\phi_s}{1 - \phi_s}. \tag{86}$$

From an alternate angle, we know that:

$$R_s(\widehat{f}_s) = \mathbb{E}\,\|\widehat{w}_s - w_s^*\|_{\Sigma_s}^2 = \mathbb{E}\,\|(X_s^\top X_s)^{-1}X_s^\top E_s\|_{\Sigma_s}^2 \tag{87}$$

$$= \sigma_s^2\mathbb{E}\,\mathrm{tr}\,X_s(X_s^\top X_s)^{-1}\Sigma_s(X_s^\top X_s)^{-1}X_s^\top \tag{88}$$

$$= \sigma_s^2\mathbb{E}\,\mathrm{tr}\,(X_s^\top X_s)^{-1}\Sigma_s = \frac{\sigma_s^2}{n_s - d - 1}\,\mathrm{tr}\,I_d = \sigma_s^2\frac{d}{n_s - d - 1} \simeq \frac{\sigma_s^2\phi_s}{1 - \phi_s}, \tag{89}$$

where we have used Lemma E.1 below.

**Lemma E.1.** *Let $n$ and $d$ be positive integers with $n \geq d + 2$. If $Z$ is an $n \times d$ random matrix with IID rows from $\mathcal{N}(0, \Sigma)$, then:*

$$\mathbb{E}(Z^\top Z)^{-1} = \frac{1}{n - d - 1}\Sigma^{-1}. \tag{90}$$

## E.2 BIAS TERM

We can compute the bias term $B_s(\widehat{f}_s)$ of the test error of $\widehat{f}_s$ evaluated on group $s$ as:

$$B_s(\widehat{f}_s) = \mathbb{E}\,\|(M_s + n_s\lambda_s I_d)^{-1}M_s w_s^* - w_s^*\|_{\Sigma_s}^2 \tag{91}$$

$$= \mathbb{E}\,\|(M_s + n_s\lambda_s I_d)^{-1}M_s w_s^* - (M_s + n_s\lambda_s I_d)^{-1}(M_s + n_s\lambda_s I_d)w_s^*\|_{\Sigma_s}^2 \tag{92}$$

$$= \mathbb{E}\,\|(M_s + n_s\lambda_s I_d)^{-1}n_s\lambda_s w_s^*\|_{\Sigma_s}^2 \tag{93}$$

$$= n_s^2\lambda_s^2\mathbb{E}\,\mathrm{tr}\,(M_s + n_s\lambda_s I_d)^{-1}w_s^*(w_s^*)^\top(M_s + n_s\lambda_s I_d)^{-1}\Sigma_s. \tag{94}$$

We can re-express this as:

$$\frac{1}{\lambda_s^2}B_s(\widehat{f}_s) = \mathbb{E}\,\bar{\mathrm{tr}}\,(H_s + \lambda_s I_d)^{-1}\Theta_s(H_s + \lambda_s I_d)^{-1}\Sigma_s \tag{95}$$

$$B_s(\widehat{f}_s) = \mathbb{E}\,\bar{\mathrm{tr}}\,(H_s/\lambda_s + I_d)^{-1}\Theta_s(H_s/\lambda_s + I_d)^{-1}\Sigma_s, \tag{96}$$

where $\Theta_s = \begin{cases} \Theta, & s = 1 \\ \Theta + \Delta, & s = 2 \end{cases}$. WLOG, we consider the case where $s = 1$. The matrix of interest has a linear pencil representation given by (with zero-based indexing):

$$(H_1/\lambda_1 + I_d)^{-1}\Theta(H_1/\lambda_1 + I_d)^{-1}\Sigma_1 = Q_{0,8}^{-1}, \tag{97}$$

where the linear pencil $Q$ is defined as follows:

$$Q = \begin{pmatrix} I_d & \Sigma_1^{\frac{1}{2}} & 0 & 0 & -\Theta & 0 & 0 & 0 & 0 \\ 0 & I_d & -\frac{1}{\sqrt{\lambda}\sqrt{n}}Z_1^\top & 0 & 0 & 0 & 0 & 0 & 0 \\ 0 & 0 & I_{n_1} & -\frac{1}{\sqrt{\lambda}\sqrt{n}}Z_1 & 0 & 0 & 0 & 0 & 0 \\ -\Sigma_1^{\frac{1}{2}} & 0 & 0 & I_d & 0 & 0 & 0 & 0 & 0 \\ 0 & 0 & 0 & 0 & I_d & \Sigma_1^{\frac{1}{2}} & 0 & 0 & -\Sigma_1 \\ 0 & 0 & 0 & 0 & 0 & I_d & -\frac{1}{\sqrt{\lambda}\sqrt{n}}Z_1^\top & 0 & 0 \\ 0 & 0 & 0 & 0 & 0 & 0 & I_{n_1} & -\frac{1}{\sqrt{\lambda}\sqrt{n}}Z_1 & 0 \\ 0 & 0 & 0 & 0 & -\Sigma_1^{\frac{1}{2}} & 0 & 0 & I_d & 0 \\ 0 & 0 & 0 & 0 & 0 & 0 & 0 & 0 & I_d \end{pmatrix}. \tag{98}$$

We note that $\overline{Q}_{0,17}^{-1} = Q_{0,8}^{-1}$. Using OVFPT, we deduce that, in the limit given by Equation 40, the following holds:

$$\mathbb{E}\,\bar{\mathrm{tr}}\,(H_1/\lambda_1 + I_d)^{-1}\Theta(H_1/\lambda_1 + I_d)^{-1}\Sigma_1 = G_{0,17}, \tag{99}$$

$$\text{with } G_{0,17} = \lambda_1\bar{\mathrm{tr}}\,(\Sigma_1 G_{2,11} + \lambda_1 I_d)^{-1}(\lambda_1\Theta + \Sigma_1 G_{2,15})(\Sigma_1 G_{6,15} + \lambda_1 I_d)^{-1}\Sigma_1. \tag{100}$$

We will now obtain the fixed-point equations satisfied by $G_{2,11}$ and $G_{6,15}$. We observe that:

$$G_{2,11} = -\frac{\lambda_1}{-\lambda_1 + \phi_1 G_{3,10}}, \quad G_{3,10} = -\lambda_1\bar{\mathrm{tr}}\,\Sigma_1(\Sigma_1 G_{2,11} + \lambda_1 I_d)^{-1} \tag{101}$$

$$\implies G_{2,11} = \frac{1}{1 + \phi_1\bar{\mathrm{tr}}\,\Sigma_1(\Sigma_1 G_{2,11} + \lambda_1 I_d)^{-1}}, \tag{102}$$

$$G_{6,15} = -\frac{\lambda_1}{-\lambda_1 + \phi_1 G_{7,14}}, \quad G_{7,14} = -\lambda_1\bar{\mathrm{tr}}\,\Sigma_1\,(\Sigma_1 G_{6,15} + \lambda_1 I_d)^{-1} \tag{103}$$

$$\implies G_{6,15} = \frac{1}{1 + \phi_1\bar{\mathrm{tr}}\,\Sigma_1\,(\Sigma_1 G_{6,15} + \lambda_1 I_d)^{-1}}. \tag{104}$$

We recognize that we must have the identification $e_1 = G_{2,11} = G_{6,15}$, where $e_1 \geq 0$. Therefore:

$$e_1 = \frac{1}{1 + \phi_1\bar{\mathrm{tr}}\,\Sigma_1\,(\Sigma_1 e_1 + \lambda_1 I_d)^{-1}}, \tag{105}$$

$$\text{i.e., } \kappa_1 = \lambda_1 + \kappa_1\phi_1\bar{\mathrm{df}}_1^{(1)}(\kappa_1). \tag{106}$$

Additionally:

$$G_{2,15} = \frac{\lambda_1\phi_1 G_{3,14}}{(-\lambda_1 + \phi_1 G_{3,10})(-\lambda_1 + \phi_1 G_{7,14})} = \phi_1 e_1^2\frac{G_{3,14}}{\lambda_1}, \tag{107}$$

$$\frac{G_{3,14}}{\lambda_1} = \bar{\mathrm{tr}}\,(\Sigma_1 G_{2,11} + \lambda_1 I_d)^{-2}(\Sigma_1 G_{2,15} + \lambda_1\Theta)\Sigma_1 \tag{108}$$

$$= \frac{G_{2,15}}{e_1^2}\bar{\mathrm{df}}_2^{(1)}(\kappa_1) + \frac{\lambda_1}{e_1^2}\bar{\mathrm{tr}}\,(\Sigma_1 + \kappa_1 I_d)^{-2}\Theta\Sigma_1, \tag{109}$$

$$\implies G_{2,15} = \phi_1 G_{2,15}\bar{\mathrm{df}}_2^{(1)}(\kappa_1) + \lambda_1\phi_1\bar{\mathrm{tr}}\,(\Sigma_1 + \kappa_1 I_d)^{-2}\Theta\Sigma_1, \tag{110}$$

$$\text{i.e., } G_{2,15} = \frac{\lambda_1\phi_1}{1 - \phi_1\bar{\mathrm{df}}_2^{(1)}(\kappa_1)}\bar{\mathrm{tr}}\,(\Sigma_1 + \kappa_1 I_d)^{-2}\Theta\Sigma_1. \tag{111}$$

Hence:

$$G_{0,17} = \kappa_1^2\bar{\mathrm{tr}}\,(\Sigma_1 + \kappa_1 I_d)^{-2}\Theta\Sigma_1 + \kappa_1^2\bar{\mathrm{df}}_2^{(1)}(\kappa_1)\frac{G_{2,15}}{\lambda_1} \tag{112}$$

$$= \kappa_1^2\bar{\mathrm{tr}}\,(\Sigma_1 + \kappa_1 I_d)^{-2}\Theta\Sigma_1 + \kappa_1^2\frac{\phi_1\bar{\mathrm{df}}_2^{(1)}(\kappa_1)}{1 - \phi_1\bar{\mathrm{df}}_2^{(1)}(\kappa_1)}\bar{\mathrm{tr}}\,(\Sigma_1 + \kappa_1 I_d)^{-2}\Theta\Sigma_1 \tag{113}$$

$$= \left(1 + \frac{\phi_1\bar{\mathrm{df}}_2^{(1)}(\kappa_1)}{1 - \phi_1\bar{\mathrm{df}}_2^{(1)}(\kappa_1)}\right)\kappa_1^2\bar{\mathrm{tr}}\,(\Sigma_1 + \kappa_1 I_d)^{-2}\Theta\Sigma_1. \tag{114}$$

In conclusion:

$$B_1(\widehat{f}_1) = \frac{\kappa_1^2\bar{\mathrm{tr}}\,(\Sigma_1 + \kappa_1 I_d)^{-2}\Theta\Sigma_1}{1 - \phi_1\bar{\mathrm{df}}_2^{(1)}(\kappa_1)}. \tag{115}$$

Following similar steps for $B_2(\widehat{f}_2)$, we get:

$$B_2(\widehat{f}_2) = \frac{\kappa_2^2\bar{\mathrm{tr}}\,(\Sigma_2 + \kappa_2 I_d)^{-2}(\Theta + \Delta)\Sigma_2}{1 - \phi_2\bar{\mathrm{df}}_2^{(2)}(\kappa_2)}. \tag{116}$$

We observe that in the unregularized case (i.e., $\lambda_s = 0$), $\kappa_s = 0$. In this setting, $B_s(\widehat{f}_s) = 0$ as expected. $\qquad\square$

# F  PROOF OF THEOREM D.1

*Proof.* We define $M = X^\top X + n\lambda I_d$. Note that one has:

$$\widehat{w} = M^{-1}(M_1 w_1^* + X_1^\top E_1 + M_2 w_2^* + X_2^\top E_2). \tag{117}$$

We deduce that $R_s(\widehat{f}) = B_s(\widehat{f}) + V_s(\widehat{f})$, where:

$$B_s(\widehat{f}) = \mathbb{E} \left\| M^{-1} M_{s'} w_{s'}^* + M^{-1} M_s w_s^* - w_s^* \right\|_{\Sigma_s}^2, \tag{118}$$

$$V_s(\widehat{f}) = \mathbb{E} \left\| M^{-1}(X_1^\top E_1 + X_2^\top E_2) \right\|_{\Sigma_s}^2 \tag{119}$$

$$= \mathbb{E} \left\| M^{-1} X_1^\top E_1 \right\|_{\Sigma_s}^2 + \mathbb{E} \left\| M^{-1} X_2^\top E_2 \right\|_{\Sigma_s}^2, \tag{120}$$

with $s' = \begin{cases} 2, & s = 1 \\ 1, & s = 2 \end{cases}$.

## F.1  VARIANCE TERMS

Note that $V_s(\widehat{f})$ of the test error of $\widehat{f}$ evaluated on group $s$ is given by:

$$V_s(\widehat{f}) = \sigma_1^2 \mathbb{E} \ \text{tr} \ X_1 M^{-1} \Sigma_s M^{-1} X_1^\top + \sigma_2^2 \mathbb{E} \ \text{tr} \ X_2 M^{-1} \Sigma_s M^{-1} X_2^\top \tag{121}$$

$$= \sigma_1^2 \mathbb{E} \ \text{tr} \ M^{-1} M_1 M^{-1} \Sigma_s + \sigma_2^2 \mathbb{E} \ \text{tr} \ M^{-1} M_2 M^{-1} \Sigma_s. \tag{122}$$

We can re-express this as:

$$nV_s(\widehat{f}) = \sigma_1^2 \mathbb{E} \, \text{tr} \, (H + \lambda I_d)^{-1} H_1 (H + \lambda I_d)^{-1} \Sigma_s + \sigma_2^2 \mathbb{E} \, \text{tr} \, (H + \lambda I_d)^{-1} H_2 (H + \lambda I_d)^{-1} \Sigma_s, \tag{123}$$

where $H = H_1 + H_2$, $H_s = X_s^\top X_s / n$, and $X_s = Z_s \Sigma_s^{1/2}$ with $Z_1 \in \mathbb{R}^{n_1 \times d}$ and $Z_2 \in \mathbb{R}^{n_2 \times d}$ being independent random matrices with IID entries from $\mathcal{N}(0, 1)$.

WLOG, we focus on $\text{tr} \, (H + \lambda I_d)^{-1} H_2 (H + \lambda I_d)^{-1} \Sigma_s$. The matrix of interest has a linear pencil representation given by (with zero-based indexing):

$$(H_1/\lambda + H_2/\lambda + I_d)^{-1} (H_2/\lambda)(H_1/\lambda + H_2/\lambda + I_d)^{-1} \Sigma_s = Q_{1,8}^{-1}, \tag{124}$$

where the linear pencil $Q$ is defined as follows:

$$
Q = 
$$

$$(125)$$

Using OVFPT, we deduce that, in the limit given by Equation 40, the following holds:

$$\mathbb{E}\,\bar{\mathrm{tr}}\,(H_1 + H_2 + \lambda I_d)^{-1} H_2 (H_1 + H_2 + \lambda I_d)^{-1} \Sigma_s = \frac{G_{1,23}}{\lambda}, \tag{126}$$

with:

$$\frac{G_{1,23}}{\lambda} = \lambda^{-1}\bar{\mathrm{tr}}\,p_2 \Sigma_2 (\lambda \Sigma_s G_{0,15} + \lambda G_{0,27} I_d - p_1 \Sigma_1 G_{0,15} G_{5,24} + p_1 \Sigma_1 G_{0,27} G_{5,20}) \tag{127}$$

$$\cdot (p_1 \Sigma_1 G_{5,20} + p_2 \Sigma_2 G_{0,15} + \lambda I_d)^{-2}. \tag{128}$$

By identifying identical entries of $\overline{Q}^{-1}$, we must have that $\frac{G_{5,20}}{\lambda} = \frac{G_{6,21}}{\lambda} = \frac{G_{10,25}}{\lambda}, \frac{G_{0,15}}{\lambda} = \frac{G_{2,17}}{\lambda} = \frac{G_{13,28}}{\lambda}$. For $G_{6,21}$ and $G_{2,17}$, we observe that:

$$G_{6,21} = -\frac{\lambda}{-\lambda + \phi G_{7,20}}, \quad G_{7,20} = -\lambda \bar{\mathrm{tr}}\, \Sigma_1 \left(p_1 \Sigma_1 G_{6,21} + p_2 \Sigma_2 G_{2,17} + \lambda I_d\right)^{-1} \tag{129}$$

$$\implies G_{6,21} = \frac{1}{1 + \phi \bar{\mathrm{tr}}\, \Sigma_1 \left(p_1 \Sigma_1 G_{6,21} + p_2 \Sigma_2 G_{2,17} + \lambda I_d\right)^{-1}}, \tag{130}$$

$$G_{2,17} = -\frac{\lambda}{-\lambda + \phi G_{3,15}}, \quad G_{3,15} = -\lambda \bar{\mathrm{tr}}\, \Sigma_2 \left(p_1 \Sigma_1 G_{6,21} + p_2 \Sigma_2 G_{2,17} + \lambda I_d\right)^{-1} \tag{131}$$

$$\implies G_{2,17} = \frac{1}{1 + \phi \bar{\mathrm{tr}}\, \Sigma_2 \left(p_1 \Sigma_1 G_{6,21} + p_2 \Sigma_2 G_{2,17} + \lambda I_d\right)^{-1}}. \tag{132}$$

We define $\eta_1 = \frac{G_{6,21}}{\lambda}, \eta_2 = \frac{G_{2,17}}{\lambda}$, with $\eta_1 \geq 0, \eta_2 \geq 0$. Therefore:

$$\eta_s = \frac{1}{\lambda + \phi \bar{\mathrm{tr}}\, \Sigma_s K^{-1}}, \tag{133}$$

where $K = \eta_1 p_1 \Sigma_1 + \eta_2 p_2 \Sigma_2 + I_d$. Additionally, by identifying identical entries of $\overline{Q}^{-1}$, we must have that $G_{5,24} = G_{6,25}, G_{0,27} = G_{2,28}$. We observe that:

$$G_{10,25} = \frac{-\lambda}{-\lambda + \phi G_{11,24}}, \tag{134}$$

$$G_{6,25} = \frac{\lambda \phi G_{7,24}}{(-\lambda + \phi G_{7,20})(-\lambda + \phi G_{11,24})} = \phi \lambda^2 \eta_1^2 \frac{G_{7,24}}{\lambda}, \tag{135}$$

$$\frac{G_{7,24}}{\lambda} = \lambda^{-2} \bar{\mathrm{tr}}\, K^{-2}(p_1 \Sigma_1 G_{6,25} + p_2 \Sigma_2 G_{2,28} - \lambda \Sigma_s)\Sigma_1, \tag{136}$$

$$\implies G_{6,25} = \phi \eta_1^2 \bar{\mathrm{tr}}\, K^{-2}(p_1 \Sigma_1 G_{6,25} + p_2 \Sigma_2 G_{2,28} - \lambda \Sigma_s)\Sigma_1, \tag{137}$$

$$G_{13,28} = \frac{-\lambda}{-\lambda + \phi G_{14,27}}, \tag{138}$$

$$G_{2,28} = \frac{\lambda \phi G_{3,27}}{(-\lambda + \phi G_{3,15})(-\lambda + \phi G_{14,27})} = \phi \lambda^2 \eta_2^2 \frac{G_{3,27}}{\lambda}, \tag{139}$$

$$\frac{G_{3,27}}{\lambda} = \lambda^{-2} \bar{\mathrm{tr}}\, K^{-2}(p_1 \Sigma_1 G_{6,25} + p_2 \Sigma_2 G_{2,28} - \lambda \Sigma_s)\Sigma_2, \tag{140}$$

$$\implies G_{2,28} = \phi \eta_2^2 \bar{\mathrm{tr}}\, K^{-2}(p_1 \Sigma_1 G_{6,25} + p_2 \Sigma_2 G_{2,28} - \lambda \Sigma_s)\Sigma_2. \tag{141}$$

We now define $v_1^{(s)} = -G_{6,25}, v_2^{(s)} = -G_{2,28}$, with $v_1^{(s)} \geq 0, v_2^{(s)} \geq 0$. Therefore, $v_1^{(s)}, v_2^{(s)}$ obey the following system of equations:

$$v_k^{(s)} = \phi \eta_k^2 \bar{\mathrm{tr}}\, K^{-2}(v_1^{(s)} p_1 \Sigma_1 + v_2^{(s)} p_2 \Sigma_2 + \lambda \Sigma_s)\Sigma_k. \tag{142}$$

We further define $u_k^{(s)} = \frac{v_k^{(s)}}{\lambda}$. Putting all the pieces together:

$$\frac{G_{1,23}}{\lambda} = \lambda^{-1} \bar{\mathrm{tr}}\, p_2 \Sigma_2 \left(\eta_2 \Sigma_s - u_2^{(s)} I_d + p_1 \Sigma_1 (\eta_2 u_1^{(s)} - \eta_1 u_2^{(s)})\right) K^{-2}. \tag{143}$$

By symmetry, in conclusion:

$$V_s(\widehat{f}) = V_s^{(1)}(\widehat{f}) + V_s^{(2)}(\widehat{f}), \tag{144}$$

$$V_s^{(k)}(\widehat{f}) = \lambda^{-1} \phi \sigma_k^2 \bar{\mathrm{tr}}\, p_k \Sigma_k \left(\eta_k \Sigma_s - u_k^{(s)} I_d + p_{k'} \Sigma_{k'} (\eta_k u_{k'}^{(s)} - \eta_{k'} u_k^{(s)})\right) K^{-2}, \tag{145}$$

with $k' = \begin{cases} 2, & k = 1 \\ 1, & k = 2 \end{cases}$.

We now corroborate our result in the limit $p_2 \to 1$ (i.e., $p_1 \to 0$) and $s = 2$. We observe that:

$$\phi \to \phi_2, \lambda \to \lambda_2, \tag{146}$$

$$V_2^{(1)}(\widehat{f}) = 0, \tag{147}$$

$$\frac{V_2^{(2)}(\widehat{f})}{\lambda^{-1}\phi_2\sigma_2^2} = \bar{\mathrm{tr}}\,\Sigma_2(\eta_2\Sigma_2 - u_2^{(2)}I_d)K^{-2} \tag{148}$$

$$v_2^{(2)} = \phi_2\eta_2^2\bar{\mathrm{tr}}\,K^{-2}(v_2^{(2)}\Sigma_2 + \lambda_2\Sigma_2)\Sigma_2 \tag{149}$$

$$= \phi_2(v_2^{(2)} + \lambda_2)\bar{\mathrm{df}}_2^{(2)}(\kappa_2), \tag{150}$$

$$u_2^{(2)} = \frac{\phi_2\bar{\mathrm{df}}_2^{(2)}(\kappa_2)}{1 - \phi_2\bar{\mathrm{df}}_2^{(2)}(\kappa_2)}, \tag{151}$$

$$\frac{V_2^{(2)}(\widehat{f})}{\lambda^{-1}\phi_2\sigma_2^2} = \kappa_2\bar{\mathrm{df}}_2^{(2)}(\kappa_2) - u_2^{(2)}\bar{\mathrm{tr}}\,\Sigma_2(\eta_2\Sigma_2 + I_d)^{-2} \tag{152}$$

$$= \kappa_2\bar{\mathrm{df}}_2^{(2)}(\kappa_2) - \kappa_2^2 u_2^{(2)}\bar{\mathrm{tr}}\,\Sigma_2(\Sigma_2 + \kappa_2 I_d)^{-2} \tag{153}$$

$$= \kappa_2\bar{\mathrm{df}}_2^{(2)}(\kappa_2) - \kappa_2 u_2^{(2)}(\bar{\mathrm{df}}_1^{(2)}(\kappa_2) - \bar{\mathrm{df}}_2^{(2)}(\kappa_2)) \tag{154}$$

$$= \kappa_2(1 + u_2^{(2)})\bar{\mathrm{df}}_2^{(2)}(\kappa_2) - \kappa_2 u_2^{(2)}\bar{\mathrm{df}}_1^{(2)}(\kappa_2) \tag{155}$$

$$= \frac{\kappa_2 - \kappa_2\phi_2\bar{\mathrm{df}}_1^{(2)}(\kappa_2)}{1 - \phi_2\bar{\mathrm{df}}_2^{(2)}(\kappa_2)} \cdot \bar{\mathrm{df}}_2^{(2)}(\kappa_2) \tag{156}$$

$$= \frac{\lambda\bar{\mathrm{df}}_2^{(2)}(\kappa_2)}{1 - \phi_2\bar{\mathrm{df}}_2^{(2)}(\kappa_2)}, \tag{157}$$

$$V_2^{(2)}(\widehat{f}) = \frac{\sigma_2^2\phi_2\bar{\mathrm{df}}_2^{(2)}(\kappa_2)}{1 - \phi_2\bar{\mathrm{df}}_2^{(2)}(\kappa_2)}, \tag{158}$$

which exactly recovers the result for $V_2(\widehat{f}_2)$ as expected.

### F.2 BIAS TERMS

Recall that:

$$B_s(\widehat{f}) = \mathbb{E}\,\|M^{-1}M_{s'}w_{s'}^* + M^{-1}M_s w_s^* - w_s^*\|_{\Sigma_s}^2. \tag{159}$$

Now, observe that $M^{-1}M_1 w_1^* - w_1^* = M^{-1}M_1 w_1^* - M^{-1}M w_1^* = -M^{-1}M_2 w_1^* - n\lambda M^{-1}w_1^*$. Let $\delta = w_2^* - w_1^*$. Then:

$$B_s(\widehat{f}) = \mathbb{E}\|M^{-1}M_{s'}(-1)^{s-1}\delta - n\lambda M^{-1}w_s^*\|_{\Sigma_s}^2 \tag{160}$$

$$= \mathbb{E}\,\mathrm{tr}\,\delta^\top M_{s'} M^{-1}\Sigma_s M^{-1}M_{s'}\delta \tag{161}$$

$$\quad - 2(-1)^{s-1}n\lambda\mathbb{E}\,\mathrm{tr}\,\delta^\top M_{s'} M^{-1}\Sigma_s M^{-1}w_s^* \tag{162}$$

$$\quad + n^2\lambda^2\mathbb{E}\,\mathrm{tr}\,(w_s^*)^\top M^{-1}\Sigma_s M^{-1}w_s^* \tag{163}$$

$$= B_s^{(1)}(\widehat{f}) - 2(-1)^{s-1}B_s^{(2)}(\widehat{f}) + B_s^{(3)}(\widehat{f}), \tag{164}$$

where:

$$B_s^{(1)}(\widehat{f}) = \mathbb{E}\bar{\mathrm{tr}}\,(H_1/\lambda + H_2/\lambda + I_d)^{-1}(H_{s'}/\lambda)\Delta(H_{s'}/\lambda)(H_1/\lambda + H_2/\lambda + I_d)^{-1}\Sigma_s, \tag{165}$$

$$B_s^{(2)}(\widehat{f}) = \mathbb{E}\,\mathrm{tr}\,\delta^\top(H_{s'}/\lambda)(H_1/\lambda + H_2/\lambda + I_d)^{-1}\Sigma_s(H_1/\lambda + H_2/\lambda + I_d)^{-1}w_s^*, \tag{166}$$

$$B_s^{(3)}(\widehat{f}) = \mathbb{E}\bar{\mathrm{tr}}\,(H_1/\lambda + H_2/\lambda + I_d)^{-1}\Theta_s(H_1/\lambda + H_2/\lambda + I_d)^{-1}\Sigma_s. \tag{167}$$

Because $\delta$ and $w_1^*$ are independent and sampled from zero-centered distributions:

$$B_1^{(2)}(\widehat{f}) = 0, \tag{168}$$

$$B_2^{(2)}(\widehat{f}) = \mathbb{E}\bar{\mathrm{tr}}\,(H_1/\lambda + H_2/\lambda + I_d)^{-1}\Delta(H_1/\lambda)(H_1/\lambda + H_2/\lambda + I_d)^{-1}\Sigma_2. \tag{169}$$

WLOG, for $B_s^{(1)}$, we focus on the case $s = 1$. The matrix of interest has a linear pencil representation given by (with zero-based indexing):

$$(H_1/\lambda + H_2/\lambda + I_d)^{-1}(H_2/\lambda)\Delta(H_2/\lambda)(H_1/\lambda + H_2/\lambda + I_d)^{-1}\Sigma_1 = Q_{1,16}^{-1}, \tag{170}$$

where the linear pencil $Q$ is defined as follows:

$$\tag{171}$$

$$Q =$$

Using OVFPT, we deduce that, in the limit given by Equation 40, the following holds:

$$\mathbb{E}\bar{\mathrm{tr}}\,(H_1/\lambda + H_2/\lambda + I_d)^{-1}(H_2/\lambda)\Delta(H_2/\lambda)(H_1/\lambda + H_2/\lambda + I_d)^{-1}\Sigma_1 = G_{1,33}, \tag{172}$$

with:

$$
\begin{aligned}
&G_{1,33} \\
&= \lambda^{-1}\bar{\mathrm{tr}}\,p_2\Sigma_2\Delta(p_2\Sigma_2 G_{2,19}^2(\lambda - p_1 G_{6,27})\Sigma_1 - G_{2,30}(p_1\Sigma_1 G_{6,23} + \lambda I_d)^2) \\
&\quad \cdot (p_1\Sigma_1 G_{6,23} + p_2\Sigma_2 G_{2,19} + \lambda I_d)^{-2}.
\end{aligned}
\tag{173}
$$

By identifying identical entries of $\overline{Q}^{-1}$, we must have that $\eta_1 = \frac{G_{6,23}}{\lambda} = \frac{G_{7,24}}{\lambda} = \frac{G_{11,28}}{\lambda}, \eta_2 = \frac{G_{2,19}}{\lambda} = \frac{G_{3,20}}{\lambda} = \frac{G_{14,31}}{\lambda}$. For $G_{7,24}$ and $G_{3,20}$, we observe that:

$$G_{7,24} = -\frac{\lambda}{-\lambda + \phi G_{8,23}}, \quad G_{8,23} = -\lambda \bar{\mathrm{tr}} \, \Sigma_1 \left( p_1 \Sigma_1 G_{7,24} + p_2 \Sigma_2 G_{3,20} + \lambda I_d \right)^{-1} \tag{174}$$

$$\implies G_{7,24} = \frac{1}{1 + \phi \bar{\mathrm{tr}} \, \Sigma_1 \left( p_1 \Sigma_1 G_{7,24} + p_2 \Sigma_2 G_{3,20} + \lambda I_d \right)^{-1}}, \tag{175}$$

$$G_{3,20} = -\frac{\lambda}{-\lambda + \phi G_{4,19}}, \quad G_{4,19} = -\lambda \bar{\mathrm{tr}} \, \Sigma_2 \left( p_1 \Sigma_1 G_{7,24} + p_2 \Sigma_2 G_{3,20} + \lambda I_d \right)^{-1} \tag{176}$$

$$\implies G_{3,20} = \frac{1}{1 + \phi \bar{\mathrm{tr}} \, \Sigma_2 \left( p_1 \Sigma_1 G_{7,24} + p_2 \Sigma_2 G_{3,20} + \lambda I_d \right)^{-1}}. \tag{177}$$

By again identifying identical entries of $\overline{Q}^{-1}$, we further have that $v_1^{(1)} = -G_{6,27} = -G_{7,28}, v_2^{(1)} = -G_{2,30} = -G_{3,31}$. We observe that:

$$G_{7,28} = \phi \lambda^2 \eta_1^2 \frac{G_{8,27}}{\lambda}, \tag{178}$$

$$\frac{G_{8,27}}{\lambda} = \lambda^{-2} \bar{\mathrm{tr}} \, K^{-2} (p_1 \Sigma_1 G_{7,28} + p_2 \Sigma_2 G_{3,31} - \lambda \Sigma_1) \Sigma_1 \tag{179}$$

$$\implies v_1^{(1)} = \phi \eta_1^2 \bar{\mathrm{tr}} \, K^{-2} (v_1^{(s)} p_1 \Sigma_1 + v_2^{(s)} p_2 \Sigma_2 + \lambda \Sigma_1) \Sigma_1, \tag{180}$$

$$G_{3,31} = \phi \lambda^2 \eta_2^2 \frac{G_{4,30}}{\lambda}, \tag{181}$$

$$\frac{G_{4,30}}{\lambda} = \lambda^{-2} \bar{\mathrm{tr}} \, K^{-2} (p_1 \Sigma_1 G_{7,28} + p_2 \Sigma_2 G_{3,31} - \lambda \Sigma_1) \Sigma_2, \tag{182}$$

$$\implies v_2^{(1)} = \phi \eta_2^2 \bar{\mathrm{tr}} \, K^{-2} (v_1^{(s)} p_1 \Sigma_1 + v_2^{(s)} p_2 \Sigma_2 + \lambda \Sigma_1) \Sigma_2. \tag{183}$$

Putting all the pieces together:

$$B_1^{(1)}(\widehat{f}) = \bar{\mathrm{tr}} \, p_2 \Sigma_2 \Delta (p_2 \eta_2^2 \Sigma_2 (1 + p_1 u_1^{(s)}) \Sigma_1 + u_2^{(s)} (p_1 \eta_1 \Sigma_1 + I_d)^2) K^{-2}. \tag{184}$$

In conclusion:

$$B_s^{(1)}(\widehat{f}) = \bar{\mathrm{tr}} \, p_{s'} \Sigma_{s'} \Delta (p_{s'} \eta_{s'}^2 \Sigma_{s'} (1 + p_s u_s^{(s)}) \Sigma_s + u_{s'}^{(s)} (p_s \eta_s \Sigma_s + I_d)^2) K^{-2}. \tag{185}$$

Now, switching our focus to $B_2^{(2)}(\widehat{f})$, the matrix of interest has a linear pencil representation given by (with zero-based indexing):

$$(H_1/\lambda + H_2/\lambda + I_d)^{-1} \Delta (H_1/\lambda)(H_1/\lambda + H_2/\lambda + I_d)^{-1} \Sigma_2 = Q_{0,15}^{-1}, \tag{186}$$

where the linear pencil $Q$ is defined as follows:

$$Q = \left( \begin{array}{c} \text{(large linear pencil matrix)} \end{array} \right). \tag{187}$$

Like before, the following holds:

$$\mathbb{E}\bar{\mathrm{tr}}\,(H_1/\lambda + H_2/\lambda + I_d)^{-1}\Delta(H_1/\lambda)(H_1/\lambda + H_2/\lambda + I_d)^{-1}\Sigma_2 = G_{1,25}, \tag{188}$$

with:

$$\begin{aligned} G_{1,25} &= \bar{\mathrm{tr}}\,p_1\Sigma_1\Delta(\lambda\Sigma_2 G_{2,18} + \lambda G_{2,26}I_d - p_2\Sigma_2 G_{2,18}G_{6,29} + p_2\Sigma_2 G_{2,26}G_{6,22}) \\ &\quad \cdot (p_1\Sigma_1 G_{2,18} + p_2\Sigma_2 G_{6,22} + \lambda I_d)^{-2} \end{aligned} \tag{189}$$

By identifying identical entries of $\overline{Q}^{-1}$, we must have that $\eta_1 = \frac{G_{2,18}}{\lambda} = \frac{G_{3,19}}{\lambda} = \frac{G_{11,27}}{\lambda}, \eta_2 = \frac{G_{6,22}}{\lambda} = \frac{G_{7,23}}{\lambda} = \frac{G_{14,30}}{\lambda}$. For $G_{3,19}$ and $G_{7,23}$, we observe that:

$$G_{3,19} = -\frac{\lambda}{-\lambda + \phi G_{4,18}}, \quad G_{4,18} = -\lambda \bar{\mathrm{tr}}\,\Sigma_1\,(p_1\Sigma_1 G_{3,19} + p_2\Sigma_2 G_{7,23} + \lambda I_d)^{-1} \tag{190}$$

$$\implies G_{3,19} = \frac{1}{1 + \phi\bar{\mathrm{tr}}\,\Sigma_1\,(p_1\Sigma_1 G_{3,19} + p_2\Sigma_2 G_{7,23} + \lambda I_d)^{-1}}, \tag{191}$$

$$G_{7,23} = -\frac{\lambda}{-\lambda + \phi G_{8,22}}, \quad G_{8,22} = -\lambda \bar{\mathrm{tr}}\,\Sigma_2\,(p_1\Sigma_1 G_{3,19} + p_2\Sigma_2 G_{7,23} + \lambda I_d)^{-1} \tag{192}$$

$$\implies G_{7,23} = \frac{1}{1 + \phi\bar{\mathrm{tr}}\,\Sigma_2\,(p_1\Sigma_1 G_{3,19} + p_2\Sigma_2 G_{7,23} + \lambda I_d)^{-1}}. \tag{193}$$

By again identifying identical entries of $\overline{Q}^{-1}$, we further have that $v_1^{(2)} = -G_{2,26} = -G_{3,27}, v_2^{(2)} = -G_{6,29} = -G_{7,30}$. We observe that:

$$G_{3,27} = \phi\lambda^2\eta_1^2\frac{G_{4,26}}{\lambda}, \tag{194}$$

$$\frac{G_{4,26}}{\lambda} = \lambda^{-2}\bar{\mathrm{tr}}\,K^{-2}(p_1\Sigma_1 G_{3,27} + p_2\Sigma_2 G_{7,30} - \lambda\Sigma_2)\Sigma_1, \tag{195}$$

$$\implies v_1^{(2)} = \phi\eta_1^2\bar{\mathrm{tr}}\,K^{-2}(v_1^{(2)}p_1\Sigma_1 + v_2^{(2)}p_2\Sigma_2 + \lambda\Sigma_2)\Sigma_1, \tag{196}$$

$$G_{7,30} = \phi\lambda^2\eta_2^2\frac{G_{8,29}}{\lambda}, \tag{197}$$

$$\frac{G_{8,29}}{\lambda} = \lambda^{-2}\bar{\mathrm{tr}}\,K^{-2}(p_1\Sigma_1 G_{3,27} + p_2\Sigma_2 G_{7,30} - \lambda\Sigma_2)\Sigma_2, \tag{198}$$

$$\implies v_2^{(2)} = \phi\eta_2^2\bar{\mathrm{tr}}\,K^{-2}(v_1^{(2)}p_1\Sigma_1 + v_2^{(2)}p_2\Sigma_2 + \lambda\Sigma_2)\Sigma_2. \tag{199}$$

Putting all the pieces together:

$$B_2^{(1)}(\widehat{f}) = 0, \tag{200}$$

$$B_2^{(2)}(\widehat{f}) = \bar{\mathrm{tr}}\,p_1\Sigma_1\Delta\big(\eta_1\Sigma_2 - u_1^{(2)}I_d + p_2\Sigma_2(\eta_1 u_2^{(2)} - \eta_2 u_1^{(2)})\big)K^{-2}. \tag{201}$$

Finally, switching our focus to $B_1^{(3)}(\widehat{f})$, the matrix of interest has a linear pencil representation given by (with zero-based indexing):

$$(H_1/\lambda + H_2/\lambda + I_d)^{-1}\Theta(H_1/\lambda + H_2/\lambda + I_d)^{-1}\Sigma_1 = Q_{1,8}^{-1}, \tag{202}$$

where the linear pencil $Q$ is defined as follows:

$$(203)$$

$$Q =$$

The following holds:

$$(H_1/\lambda + H_2/\lambda + I_d)^{-1}\Theta(H_1/\lambda + H_2/\lambda + I_d)^{-1}\Sigma_1 = G_{1,23}, \qquad (204)$$

with $G_{1,23} = \lambda\bar{\mathrm{tr}}\,\Theta(-p_1\Sigma_1 G_{2,24} - p_2\Sigma_2 G_{5,27} + \lambda\Sigma_1)(p_1\Sigma_1 G_{2,17} + p_2\Sigma_2 G_{5,20} + \lambda I_d)^{-2}$.

By identifying identical entries of $\overline{Q}^{-1}$ and following similar steps as before, we must have the identification $\eta_1 = \frac{G_{2,17}}{\lambda}, \eta_2 = \frac{G_{5,20}}{\lambda}$, as well as $v_1^{(1)} = -G_{2,24}, v_2^{(1)} = -G_{5,27}$. Therefore, in conclusion:

$$B_s^{(3)}(\widehat{f}) = \bar{\mathrm{tr}}\,\Theta_s(p_1 u_1^{(s)}\Sigma_1 + p_2 u_2^{(s)}\Sigma_2 + \Sigma_s)K^{-2}. \qquad (205)$$

In the limit $p_s \to 1$ (i.e., $p_{s'} \to 0$), we observe that:

$$\phi \to \phi_s, \lambda \to \lambda_s, \tag{206}$$

$$B_s^{(1)}(\widehat{f}) \to 0, \tag{207}$$

$$B_s^{(2)}(\widehat{f}) \to 0, \tag{208}$$

$$B_s^{(3)}(\widehat{f}) = \bar{\text{tr}}\,\Theta_s(u_s^{(s)} + 1)\Sigma_s K^{-2}, \tag{209}$$

$$v_s^{(s)} = \phi_s \eta_s^2 \bar{\text{tr}}\,K^{-2}(v_s^{(s)} + \lambda_s)\Sigma_s^2 \tag{210}$$

$$= \phi_s(v_s^{(s)} + \lambda_s)\bar{\text{df}}_2^{(s)}(\kappa_s) \tag{211}$$

$$u_s^{(s)} = \frac{\phi_s \bar{\text{df}}_2^{(s)}(\kappa_s)}{1 - \phi_s \bar{\text{df}}_2^{(s)}(\kappa_s)}, \tag{212}$$

$$B_s^{(3)}(\widehat{f}) = \frac{\kappa_s^2 \bar{\text{tr}}\,\Theta_s \Sigma_s(\Sigma_s + \kappa_s I_d)^{-2}}{1 - \phi_s \bar{\text{df}}_2^{(s)}(\kappa_s)}, \tag{213}$$

$$B_s(\widehat{f}) \to B_s^{(3)}(\widehat{f}), \tag{214}$$

which matches up exactly with $B_s(\widehat{f}_s)$ as expected. $\qquad\square$

## G  PROOF OF THEOREM 3.1

*Proof.* The gradient of the loss $L$ is given by:

$$\nabla L(\eta) = \sum_s S^\top X_s^\top (X_s S\eta - Y_s)/n + \lambda\eta = \sum_s S^\top M_s S\eta - \sum_s S^\top X_s^\top Y_s/n + \lambda\eta$$

$$= H\eta - \sum_s S^\top X_s^\top Y_s/n,$$

where $H = S^\top M S + \lambda I_m \in \mathbb{R}^{m\times m}$, with $M = M_1 + M_2$ and $M_s = X_s^\top X_s/n$. Thus, setting $R = H^{-1}$, we may write:

$$\widehat{w} = S\widehat{\eta} = SRS^\top (X_1^\top Y_1 + X_2^\top Y_2)/n$$
$$= SRS^\top (M_1 w_1^* + M_2 w_2^*) + SRS^\top X_1^\top E_1/n + SRS^\top X_2^\top E_2/n.$$

We deduce the following bias-variance decomposition:

$$\mathbb{E}\|\widehat{w} - w_s^*\|_{\Sigma_s}^2 = B_s(\widehat{f}) + V_s(\widehat{f}), \text{ where}$$
$$V_s(\widehat{f}) = V_s^{(1)}(\widehat{f}) + V_s^{(2)}(\widehat{f}), \text{ with } V_s^{(j)}(\widehat{f}) = \sigma_j^2 \phi \mathbb{E}\bar{\mathrm{tr}}\, M_j SRS^\top \Sigma_s SRS^\top,$$
$$B_s(\widehat{f}) = \mathbb{E}\|SRS^\top (M_1 w_1^* + M_2 w_2^*) - w_s^*\|_{\Sigma_s}^2.$$

We can further decompose $B_s(\widehat{f})$, first considering the case $s = 1$. We define $\delta = w_2^* - w_1^*$.

$$\mathbb{E}\|SRS^\top (M_1 w_1^* + M_2 w_2^*) - w_1^*\|_{\Sigma_1}^2$$
$$= \mathbb{E}\|(SRS^\top (M_1 + M_2) - I_d)w_1^* + SRS^\top M_2\delta\|_{\Sigma_1}^2$$
$$= \mathbb{E}\|(SRS^\top M - I_d)w_1^*\|_{\Sigma_1}^2 + \mathbb{E}\|SRS^\top M_2\delta\|_{\Sigma_1}^2$$
$$= \mathbb{E}\bar{\mathrm{tr}}\,\Theta(MSRS^\top - I_d)\Sigma_1(SRS^\top M - I_d) + \mathbb{E}\bar{\mathrm{tr}}\,\Delta M_2 SRS^\top \Sigma_1 SRS^\top M_2$$
$$= \mathbb{E}\bar{\mathrm{tr}}\,\Theta\Sigma_1 + \mathbb{E}\bar{\mathrm{tr}}\,\Theta MSRS^\top \Sigma_1 SRS^\top M - 2\mathbb{E}\bar{\mathrm{tr}}\,\Theta\Sigma_1 SRS^\top M + \mathbb{E}\bar{\mathrm{tr}}\,\Delta M_2 SRS^\top \Sigma_1 SRS^\top M_2.$$

We can similarly decompose $B_2$:

$$\mathbb{E}\|SRS^\top (M_1 w_1^* + M_2 w_2^*) - w_2^*\|_{\Sigma_2}^2$$
$$= \mathbb{E}\|SRS^\top (M_1 w_1^* + M_2 w_2^*) - w_2^*\|_{\Sigma_2}^2$$
$$= \mathbb{E}\|(SRS^\top (M_1 + M_2) - I_d)w_2^* - SRS^\top M_1\delta\|_{\Sigma_2}^2$$
$$= \mathbb{E}\|(SRS^\top M - I_d)w_2^*\|_{\Sigma_2}^2 + \mathbb{E}\|SRS^\top M_1\delta\|_{\Sigma_2}^2 - 2\mathbb{E}\,\mathrm{tr}\,(w_2^*)^\top (MSRS^\top - I_d)\Sigma_2 SRS^\top M_1\delta$$
$$= \mathbb{E}\bar{\mathrm{tr}}\,\Theta_2(MSRS^\top - I_d)\Sigma_2(SRS^\top M - I_d) + \mathbb{E}\bar{\mathrm{tr}}\,\Delta M_1 SRS^\top \Sigma_2 SRS^\top M_1$$
$$- 2\mathbb{E}\bar{\mathrm{tr}}\,\Delta(MSRS^\top - I_d)\Sigma_2 SRS^\top M_1$$
$$= \mathbb{E}\bar{\mathrm{tr}}\,\Theta_2\Sigma_2 + \mathbb{E}\bar{\mathrm{tr}}\,\Theta_2 MSRS^\top \Sigma_2 SRS^\top M - 2\mathbb{E}\bar{\mathrm{tr}}\,\Theta_2\Sigma_2 SRS^\top M$$
$$+ \mathbb{E}\bar{\mathrm{tr}}\,\Delta M_1 SRS^\top \Sigma_2 SRS^\top M_1 - 2\mathbb{E}\bar{\mathrm{tr}}\,\Delta MSRS^\top \Sigma_2 SRS^\top M_1 + 2\mathbb{E}\bar{\mathrm{tr}}\,\Delta\Sigma_2 SRS^\top M_1.$$

Furthermore, we observe that:

$$\mathbb{E}\bar{\mathrm{tr}}\,AMSRS^\top BSRS^\top M \tag{215}$$
$$= \mathbb{E}\bar{\mathrm{tr}}\,AM_1 SRS^\top BSRS^\top M_1 + \mathbb{E}\bar{\mathrm{tr}}\,AM_2 SRS^\top BSRS^\top M_2 + 2\mathbb{E}\bar{\mathrm{tr}}\,AM_1 SRS^\top BSRS^\top M_2, \tag{216}$$
$$\mathbb{E}\bar{\mathrm{tr}}\,ASRS^\top M = \mathbb{E}\bar{\mathrm{tr}}\,ASRS^\top M_1 + \mathbb{E}\bar{\mathrm{tr}}\,ASRS^\top M_2. \tag{217}$$

Hence, we desire deterministic equivalents for the following expressions:

$$r_j^{(1)}(A) = A S\overline{R}S^\top \overline{M}_j, \tag{218}$$
$$r_j^{(2)}(A, B) = A\overline{M}_j S\overline{R}S^\top B S\overline{R}S^\top, \tag{219}$$
$$r_j^{(3)}(A, B) = A\overline{M}_j S\overline{R}S^\top B S\overline{R}S^\top \overline{M}_j, \tag{220}$$
$$r_j^{(4)}(A, B) = A\overline{M}_j S\overline{R}S^\top B S\overline{R}S^\top \overline{M}_{j'}, \tag{221}$$

where:

$$\overline{M}_j = \Sigma_j^{1/2} Z_j^\top Z_j \Sigma_j^{1/2}, \overline{R} = (S^\top \overline{M} S + I_m)^{-1}, \overline{M} = \overline{M}_1 + \overline{M}_2, \tag{222}$$

$$\overline{M}_j = M_j/\lambda, \overline{R} = \lambda R, \overline{M} = M/\lambda. \tag{223}$$

In summary:

$$V_s^{(j)}(\widehat{f}) = \sigma_j^2 \phi \lambda^{-1} \mathbb{E}\bar{\mathrm{tr}}\, r_j^{(2)}(I_d, \Sigma_s), \tag{224}$$

$$B_s(\widehat{f}) = \bar{\mathrm{tr}}\, \Theta_s \Sigma_s \tag{225}$$

$$+ \mathbb{E}\bar{\mathrm{tr}}\, r_1^{(3)}(\Theta_s, \Sigma_s) + \mathbb{E}\bar{\mathrm{tr}}\, r_2^{(3)}(\Theta_s, \Sigma_s) + 2\mathbb{E}\bar{\mathrm{tr}}\, r_1^{(4)}(\Theta_s, \Sigma_s) \tag{226}$$

$$- 2\mathbb{E}\bar{\mathrm{tr}}\, r_1^{(1)}(\Theta_s \Sigma_s) - 2\mathbb{E}\bar{\mathrm{tr}}\, r_2^{(1)}(\Theta_s \Sigma_s) \tag{227}$$

$$+ \mathbb{E}\bar{\mathrm{tr}}\, r_{s'}^{(3)}(\Delta, \Sigma_s) \tag{228}$$

$$- 2 \begin{cases} 0, & s = 1, \\ \mathbb{E}\bar{\mathrm{tr}}\, r_1^{(3)}(\Delta, \Sigma_2) + \mathbb{E}\bar{\mathrm{tr}}\, r_2^{(4)}(\Delta, \Sigma_2) - \mathbb{E}\bar{\mathrm{tr}}\, r_1^{(1)}(\Delta\Sigma_2), & s = 2 \end{cases}. \tag{229}$$

## G.1 COMPUTING $\mathbb{E}\bar{\mathrm{tr}}\, r_j^{(1)}$

WLOG, we focus on $r_1^{(1)}$. The matrix of interest has a linear pencil representation given by (with zero-based indexing):

$$r_1^{(1)} = Q_{1,10}^{-1}, \tag{230}$$

where the linear pencil $Q$ is defined as follows:

$$Q = \begin{pmatrix} I_d & 0 & -S & 0 & 0 & 0 & 0 & 0 & 0 & 0 & 0 \\ -A & I_d & 0 & 0 & 0 & 0 & 0 & 0 & 0 & 0 & 0 \\ 0 & 0 & I_m & S^\top & 0 & 0 & 0 & 0 & 0 & 0 & 0 \\ 0 & 0 & 0 & I_d & -\Sigma_1^{\frac{1}{2}} & 0 & 0 & -\Sigma_2^{\frac{1}{2}} & 0 & 0 & 0 \\ 0 & 0 & 0 & 0 & I_d & -\frac{1}{\sqrt{\lambda}}Z_1^\top & 0 & 0 & 0 & 0 & 0 \\ 0 & 0 & 0 & 0 & 0 & I_{n_1} & -\frac{1}{\sqrt{\lambda}}Z_1 & 0 & 0 & 0 & 0 \\ -\Sigma_1^{\frac{1}{2}} & 0 & 0 & 0 & 0 & 0 & I_d & 0 & 0 & 0 & \Sigma_1^{\frac{1}{2}} \\ 0 & 0 & 0 & 0 & 0 & 0 & 0 & I_d & -\frac{1}{\sqrt{\lambda}}Z_2^\top & 0 & 0 \\ 0 & 0 & 0 & 0 & 0 & 0 & 0 & I_{n_2} & -\frac{1}{\sqrt{\lambda}}Z_2 & 0 \\ -\Sigma_2^{\frac{1}{2}} & 0 & 0 & 0 & 0 & 0 & 0 & 0 & 0 & I_d & 0 \\ 0 & 0 & 0 & 0 & 0 & 0 & 0 & 0 & 0 & 0 & I_d \end{pmatrix}. \tag{231}$$

Using the tools of OVFPT, the following holds:

$$\mathbb{E}\bar{\mathrm{tr}}\, r_1^{(1)} = G_{1,21}, \tag{232}$$

with:

$$G_{1,21} = \bar{\mathrm{tr}}\, \gamma p_1 \Sigma_1 A G_{2,13} G_{5,16}(\gamma G_{2,13}(p_1 \Sigma_1 G_{5,16} + p_2 G_{8,19}) + \lambda I_d)^{-1}. \tag{233}$$

For $G_{5,16}$ and $G_{8,19}$, we observe that:

$$G_{5,16} = \frac{-\lambda}{-\lambda + \phi G_{6,15}}, \quad G_{6,15} = -\lambda \gamma G_{2,13} \bar{\mathrm{tr}}\, \Sigma_1(\gamma G_{2,13}(p_1 \Sigma_1 G_{5,16} + p_2 \Sigma_2 G_{8,19}) + \lambda I_d)^{-1}, \tag{234}$$

$$\implies G_{5,16} = \frac{1}{1 + \psi G_{2,13} \bar{\mathrm{tr}}\, \Sigma_1(\gamma G_{2,13}(p_1 \Sigma_1 G_{5,16} + p_2 \Sigma_2 G_{8,19}) + \lambda I_d)^{-1}}, \tag{235}$$

$$G_{8,19} = \frac{-\lambda}{-\lambda + \phi G_{9,18}}, \quad G_{9,18} = -\lambda \gamma G_{2,13} \bar{\mathrm{tr}}\, \Sigma_2(\gamma G_{2,13}(p_1 \Sigma_1 G_{5,16} + p_2 \Sigma_2 G_{8,19}) + \lambda I_d)^{-1}, \tag{236}$$

$$G_{8,19} = \frac{1}{1 + \psi G_{2,13} \bar{\mathrm{tr}}\, \Sigma_2(\gamma G_{2,13}(p_1 \Sigma_1 G_{5,16} + p_2 \Sigma_2 G_{8,19}) + \lambda I_d)^{-1}}. \tag{237}$$

We define $e_1 = G_{5,16}, e_2 = G_{8,19}$, with $e_1 \geq 0, e_2 \geq 0$. We further observe that:

$$G_{2,13} = \frac{1}{1 + G_{3,11}}, \tag{238}$$

$$G_{3,11} = \bar{\text{tr}}\,(p_1 \Sigma_1 G_{5,16} + p_2 \Sigma_2 G_{8,19})(\gamma G_{2,13}(p_1 \Sigma_1 G_{5,16} + p_2 \Sigma_2 G_{8,19}) + \lambda I_d)^{-1}. \tag{239}$$

We define $\tau = G_{2,13} \geq 0$. We further define $L = p_1 e_1 \Sigma_1 + p_2 e_2 \Sigma_2, K = \gamma \tau L + \lambda I_d$. Therefore, we have the following system of equations:

$$e_s = \frac{1}{1 + \psi \tau \bar{\text{tr}}\,\Sigma_s K^{-1}}, \tau = \frac{1}{1 + \bar{\text{tr}}\,L K^{-1}}. \tag{240}$$

In conclusion:

$$\mathbb{E}\bar{\text{tr}}\,r_j^{(1)} = p_j \gamma e_j \tau \bar{\text{tr}}\,A \Sigma_j K^{-1}. \tag{241}$$

## G.2 Computing $\mathbb{E}\bar{\text{tr}}\,r_j^{(2)}$

WLOG, we focus on $r_1^{(2)}$. The matrix of interest has a linear pencil representation given by (with zero-based indexing):

$$r_1^{(2)} = -Q_{1,13}^{-1}, \tag{242}$$

where the linear pencil $Q$ is defined as follows:

$$Q = \begin{pmatrix} & & & & & & & & & & & & & & & & & -A & I_d \\ & & & & & & & & & & & & & & & & & I_d & \\ & & & & & & & & & & & -\Sigma_1^{\frac{1}{2}} & & & & & I_d & & -\Sigma_1^{\frac{1}{2}} \\ & & & & & & & & & & & & & & \frac{1}{\sqrt{N}}Z_1^\top I_{n_1} & & & \\ & & & & & & & & & & & & \frac{1}{\sqrt{N}}Z_1 I_d & & & & \\ & & & & & & -\Sigma_2^{\frac{1}{2}} & & & & I_d & -\Sigma_1^{\frac{1}{2}} & & & & & \\ & & & & & & & & & I_m & -S & & & & & & \\ & & & & & & & & I_d & -S^\top & & & & & & & \\ & & & & & & & I_d & -\Sigma_2^{\frac{1}{2}} & & & & & & & & \\ & & & & & & \frac{1}{\sqrt{N}}Z_2^\top I_{n_2} & & & & & & & & & \\ & & & & & I_d & \frac{1}{\sqrt{N}}Z_2 & & & & & & & & & \\ -\Sigma_2^{\frac{1}{2}} & & -\Sigma_1^{\frac{1}{2}} & & & I_d & & B & & & & & & & & \\ & & & I_m & -S & & & & & & & & & & & \\ & & S^\top I_d & & & & & & & & & & & & & \\ & & I_d & -\Sigma_1^{\frac{1}{2}} & & & & & & & & & & & & \\ & \frac{1}{\sqrt{N}}I_{n_1}Z_1^\top & & & & & & & & & & & & & & \\ & I_d \frac{1}{\sqrt{N}}Z_1 & & & & & & & & & & & & & & \\ I_d & & & -\Sigma_2^{\frac{1}{2}} & & & & & & & & & & & & \\ \frac{1}{\sqrt{N}}I_{n_2}Z_2^\top & & & & & & & & & & & & & & & \\ I_d \frac{1}{\sqrt{N}}Z_2 & & & & & & & & & & & & & & & \end{pmatrix}.$$

(243)

The following holds:

$$\bar{\mathbb{E}\mathrm{tr}}\, r_1^{(2)} = -G_{1,33},$$

(244)

with:

$$G_{1,33} = -p_1 \bar{\mathrm{tr}}\, A\Sigma_1 P_1 P_2^{-1},$$ (245)

$$P_1 = \gamma\lambda B G_{3,23} G_{6,26} G_{12,32} - \gamma p_2 \Sigma_2 G_{3,23} G_{6,26} G_{9,38} G_{12,32}$$ (246)

$$+ \gamma p_2 \Sigma_2 G_{3,35} G_{6,26} G_{9,29} G_{12,32} + \lambda G_{3,23} G_{6,32} I_d + \lambda G_{3,15} G_{12,32} I_d,$$ (247)

$$P_2 = (\gamma G_{6,26}(p_1\Sigma_1 G_{3,23} + \gamma p_2 \Sigma_2 G_{9,29}) + \lambda I_d)$$ (248)

$$\cdot (\gamma G_{12,32}(p_1\Sigma_1 G_{15,35} + p_2\Sigma_2 G_{18,38}) + \lambda I_d).$$ (249)

Following similar steps as before and recognizing identifications, we arrive at that:

$$e_1 = G_{3,23} = G_{15,35},$$ (250)

$$e_2 = G_{9,29} = G_{18,38},$$ (251)

$$\tau = G_{6,26} = G_{12,32}.$$ (252)

We now focus on the remaining terms. We observe that:

$$G_{3,35} = \phi e_1^2 \frac{G_{4,14}}{\lambda} \tag{253}$$

$$\frac{G_{4,14}}{\lambda} = \gamma \bar{\text{tr}}\, \Sigma_1 (\gamma \tau^2 (p_1 \Sigma_1 G_{3,35} + p_2 \Sigma_2 G_{9,38} - \lambda B) - \lambda G_{6,32} I_d) K^{-2}, \tag{254}$$

$$G_{9,38} = \phi e_2^2 \frac{G_{10,37}}{\lambda}, \tag{255}$$

$$\frac{G_{10,37}}{\lambda} = \gamma \bar{\text{tr}}\, \Sigma_2 (\gamma \tau^2 (p_1 \Sigma_1 G_{3,35} + p_2 \Sigma_2 G_{9,38} - \lambda B) - \lambda G_{6,32} I_d) K^{-2}. \tag{256}$$

We define $u_1 = -\frac{G_{3,35}}{\lambda}, u_2 = -\frac{G_{9,38}}{\lambda}$, with $u_1 \le 0, u_2 \le 0$. We further define $D = p_1 u_1 \Sigma_1 + p_2 u_2 \Sigma_2 + B$. We now observe that:

$$G_{6,32} = -\frac{G_{7,31}}{(G_{7,25} + 1)(G_{13,31} + 1)} = -\tau^2 G_{7,31}, \tag{257}$$

$$G_{7,31} = -\bar{\text{tr}}\, (\gamma G_{6,32} L^2 + \lambda^2 D) K^{-2}. \tag{258}$$

Defining $\rho = G_{6,32}$, we must have the following system of equations:

$$u_s = \psi e_s^2 \bar{\text{tr}}\, \Sigma_s (\gamma \tau^2 D + \rho I_d) K^{-2}, \tag{259}$$

$$\rho = \tau^2 \bar{\text{tr}}\, (\gamma \rho L^2 + \lambda^2 D) K^{-2}. \tag{260}$$

In conclusion:

$$P_2 = K^2, \tag{261}$$

$$-P_1 = \lambda \gamma e_1 \tau^2 B + \lambda \gamma \tau^2 p_2 \Sigma_2 (e_1 u_2 - e_2 u_1) + \lambda e_1 \rho I_d - \lambda^2 u_1 \tau I_d, \tag{262}$$

$$\mathbb{E} \bar{\text{tr}}\, r_j^{(2)} = \lambda p_j \gamma \bar{\text{tr}}\, A \Sigma_j (\gamma e_j \tau^2 B + \gamma \tau^2 p_{j'} \Sigma_{j'} (e_j u_{j'} - e_{j'} u_j) + e_j \rho I_d - \lambda u_j \tau I_d) K^{-2}. \tag{263}$$

## G.3 Computing $\mathbb{E} \bar{\text{tr}}\, r_j^{(3)}$

WLOG, we focus on $r_1^{(3)}$. The matrix of interest has a linear pencil representation given by (with zero-based indexing):

$$r_1^{(3)} = Q_{1,20}^{-1}, \tag{264}$$

where the linear pencil $Q$ is defined as follows:

$$Q = \begin{pmatrix} & & & & & & & & & & & & & & & & & & \frac{I_d}{A} \\ & & & & & & & & & & & & & & & & & \frac{I_d}{} & \\ & & & & & & & & & & \frac{I_d}{\Sigma_1^{\frac{1}{2}}} & & & & & \frac{I_d}{} & & \frac{\Sigma_1^{\frac{1}{2}}}{} \\ & & & & & & & & & & & & & & \frac{I_{m_1}}{\sqrt{\lambda} Z_1^\top} & & & \\ & & & & & & & & & & & & \frac{I_d}{\sqrt{\lambda} Z_1} & & & & \\ & & & & & & \frac{I_d}{\Sigma_2^{\frac{1}{2}}} & & & & & \frac{I_d}{\Sigma_1^{\frac{1}{2}}} & & & & \\ & & & & & & & & & \frac{I_m}{S} & & & & & \\ & & & & & & & & \frac{I_d}{S^\top} & & & & & & \\ & & & & & & & \frac{I_d}{\Sigma_2^{\frac{1}{2}}} & & & & & & & & \\ & & & & & & & \frac{I_{n_2}}{\sqrt{\lambda} Z_2^\top} & & & & & & & & \\ & & & & & & \frac{I_d}{\sqrt{\lambda} Z_2} & & & & & & & & & \\ \frac{I_d}{\Sigma_2^{\frac{1}{2}}} & & \frac{I_d}{\Sigma_1^{\frac{1}{2}}} & & & & \frac{I_d}{} & & B & & & & & & \\ & & & & & \frac{I_m}{S} & & & & & & & & & \\ & & & & \frac{I_d}{S^\top} & & & & & & & & & & \\ & & & \frac{I_d}{\Sigma_1^{\frac{1}{2}}} & & & & & & & & & & & \\ & & \frac{I_{n_1}}{\sqrt{\lambda} Z_1^\top} & & & & & & & & & & & & \\ & & \frac{I_d}{\sqrt{\lambda} Z_1} & & & & & & & & & & & & \\ & \frac{I_d}{} & & & \frac{I_d}{\Sigma_2^{\frac{1}{2}}} & & & & & & & & & & \\ & \frac{I_{n_2}}{\sqrt{\lambda} Z_2^\top} & & & & & & & & & & & & & \\ \frac{I_d}{\sqrt{\lambda} Z_2} & & & & & & & & & & & & & & \\ \frac{I_d}{} & & \frac{I_d}{\Sigma_2^{\frac{1}{2}}} & & & & & & & & & & & & \end{pmatrix}$$

(265)

It holds that $\mathbb{E}\bar{\mathrm{tr}}\, r_1^{(3)} = G_{1,41}$. We immediately observe that:

$$e_1 = G_{3,24}, G_{15,36}, \tag{266}$$
$$e_2 = G_{9,30}, G_{18,39}, \tag{267}$$
$$\tau = G_{6,27}, G_{12,33}, \tag{268}$$
$$u_1 = -\frac{G_{3,36}}{\lambda}, \tag{269}$$
$$u_2 = -\frac{G_{9,39}}{\lambda}, \tag{270}$$
$$\rho = G_{6,33}. \tag{271}$$

In conclusion:

$$\mathbb{E}\bar{\mathrm{tr}}\, r_j^{(3)} = p_j \bar{\mathrm{tr}}\, A\Sigma_j(\gamma e_j^2 p_j \Sigma_j(\gamma\tau^2 u_{j'} p_{j'} \Sigma_{j'} + \gamma\tau^2 B + \rho I_d) + u_j(\gamma e_{j'}\tau p_{j'}\Sigma_{j'} + \lambda I_d)^2)K^{-2}. \tag{272}$$

## G.4 Computing $\mathbb{E}\bar{\mathrm{tr}}\, r_j^{(4)}$

WLOG, we focus on $r_1^{(4)}$. The matrix of interest has a linear pencil representation given by (with zero-based indexing):

$$r_1^{(4)} = Q_{1,20}^{-1}, \tag{273}$$

where the linear pencil $Q$ is defined as follows:

$$\tag{274}$$

$$Q = \underbrace{\begin{pmatrix} \cdots \end{pmatrix}}_{.}$$

It holds that $\mathbb{E}\bar{\mathrm{tr}}\, r_1^{(4)} = G_{1,41}$. We immediately observe that:

$$e_1 = G_{3,24}, G_{15,36}, \tag{275}$$

$$e_2 = G_{9,30}, G_{18,39}, \tag{276}$$

$$\tau = G_{6,27}, G_{12,33}, \tag{277}$$

$$u_1 = -\frac{G_{3,36}}{\lambda}, \tag{278}$$

$$u_2 = -\frac{G_{9,39}}{\lambda}, \tag{279}$$

$$\rho = G_{6,33}. \tag{280}$$

In conclusion:

$$\mathbb{E}\bar{\operatorname{tr}}\, r_j^{(4)} = p_j \gamma p_{j'} \bar{\operatorname{tr}}\, \Sigma_j \Sigma_{j'} A (\gamma \tau^2 (B e_j e_{j'} - p_j \Sigma_j e_j^2 u_{j'} - p_{j'} \Sigma_{j'} e_{j'}^2 u_j) \tag{281}$$

$$- \lambda \tau (e_j u_{j'} + e_{j'} u_j) I_d + e_j e_{j'} \rho I_d) K^{-2}. \tag{282}$$

$\square$

## H Theorem 3.2

**Definition H.1.** *Let $(e_1, e_2, \tau_1, \tau_2, u_1, u_2, \rho_1, \rho_2)$ is be unique positive solution to the following system of fixed-point equations:*

$$e_s = \frac{1}{1 + \psi_s \tau_s \bar{\mathrm{tr}} \, \Sigma_s (\gamma \tau_s e_s \Sigma_s + \lambda_s I_d)^{-1}}, \textit{ for } s \in \{1, 2\} \tag{283}$$

$$\tau_s = \frac{1}{1 + \bar{\mathrm{tr}} \, e_s \Sigma_s (\gamma \tau_s e_s \Sigma_s + \lambda_s I_d)^{-1}}, \textit{ for } s \in \{1, 2\} \tag{284}$$

$$u_s = \psi_s e_s^2 \bar{\mathrm{tr}} \, \Sigma_s (\gamma \tau_s^2 (u_s + 1) \Sigma_s + \rho_s I_d)(\gamma \tau_s e_s \Sigma_s + \lambda_s I_d)^{-2}, \textit{ for } s \in \{1, 2\} \tag{285}$$

$$\rho_s = \tau_s^2 \bar{\mathrm{tr}} \, (\gamma \rho_s (e_s \Sigma_s)^2 + \lambda_s^2 (u_s + 1) \Sigma_s)(\gamma \tau_s e_s \Sigma_s + \lambda_s I_d)^{-2}, \textit{ for } s \in \{1, 2\}. \tag{286}$$

*For deterministic $d \times d$ PSD matrices A and B, we define the following auxiliary quantities:*

$$h_j^{(1)}(A) := \gamma e_j \tau_j \bar{\mathrm{tr}} \, A \Sigma_j (\gamma \tau_j e_j \Sigma_j + \lambda_j I_d)^{-1}, \tag{287}$$

$$h_j^{(2)}(A) := \gamma \bar{\mathrm{tr}} \, A \Sigma_j (\gamma e_j \tau_j^2 \Sigma_j + e_j \rho_j I_d - \lambda_j u_j \tau_j I_d)(\gamma \tau_j e_j \Sigma_j + \lambda_j I_d)^{-2}, \tag{288}$$

$$h_j^{(3)}(A) := \bar{\mathrm{tr}} \, A \Sigma_j (\gamma e_j^2 \Sigma_j (\gamma \tau_j^2 \Sigma_j + \rho_j I_d) + \lambda_j^2 u_j I_d)(\gamma \tau_j e_j \Sigma_j + \lambda_j I_d)^{-2}. \tag{289}$$

Under Assumptions B.2 and 3.1, it holds that:

$$R_s(\widehat{f}_s) \simeq B_s(\widehat{f}_s) + V_s(\widehat{f}_s), \text{ with } V_s(\widehat{f}_s) = \lim_{p_s \to 1} V_s(\widehat{f}), \quad B_s(\widehat{f}_s) = \lim_{p_s \to 1} B_s(\widehat{f}). \tag{290}$$

More explicitly:

$$V_s(\widehat{f}_s) = \sigma_s^2 \phi_s h_s^{(2)}(I_d), \quad B_s(\widehat{f}_s) = \bar{\mathrm{tr}} \, \Theta_s \Sigma_s + h_s^{(3)}(\Theta_s) - 2 h_s^{(1)}(\Theta_s \Sigma_s). \tag{291}$$

*Proof.* Theorem 3.2 follows from Theorem 3.1 in the limit $p_s \to 1$ (i.e., $p_{s'} \to 0$). $\qquad\square$

The scalars $e_s, \tau_s, u_s, \rho_s$ can be intuitively interpreted in the setting where a separate model is learned for each group. For ridge regression with random projections and $\lambda \to 0^+$, we show in Equations 337 and 338 that $e_s, \tau_s$ are related to the normalized first-order degrees of freedom $I_{1,1}$ of the population covariance matrix $\Sigma_s$. $e_s$ captures the effect of the feature rate $\phi_s$ while $\tau$ captures the effect of the parameterization rate $\gamma$. Similarly, for classical ridge regression, we show in Equation 69 that $e_s$ is related to the normalized first-order degrees of freedom $\bar{\mathrm{df}}_1^{(s)}$. On the other hand, $u_s$ and $\rho_s$ can be understood as pseudo-variances. Indeed, for ridge regression with random projections and $\lambda \to 0^+$, Equations 341 and 349 show that $u_s, \rho_s, V_s$ are all related to the normalized second-order degrees of freedom $I_{2,2}$ of $\Sigma_s$.

# I  SOLVING FIXED-POINT EQUATIONS FOR THEOREM D.1

## I.1  PROPORTIONAL COVARIANCE MATRICES

When $\lambda \to 0^+$, it is not possible to analytically solve the fixed-point equations for the constants in Definition 3.1 for general $\Sigma_1, \Sigma_2$. As such, we consider a more tractable case where the covariance matrices are proportional, i.e., $\Sigma_1 = a_1\Sigma$ and $\Sigma_2 = a_2\Sigma$, for some $\Sigma \in \mathbb{R}^{d \times d}$.

We define $\theta = \frac{\lambda}{\gamma\tau(a_1 p_1 e_1 + a_2 p_2 e_2)}$ and $\eta = \bar{\mathrm{tr}}\,\Sigma(\Sigma + \theta I_d)^{-1}$. Then, we have that:

$$1/e_s = e_s' = 1 + \psi\tau\bar{\mathrm{tr}}\,\Sigma_s K^{-1} = 1 + \frac{\phi a_s \eta}{a_1 p_1 e_1 + a_2 p_2 e_2}, \tag{292}$$

$$1/\tau = \tau' = 1 + \bar{\mathrm{tr}}\,L K^{-1} = 1 + (\eta/\gamma)\tau' = \frac{1}{1 - \eta/\gamma}. \tag{293}$$

If $\theta_0 = 0$, then $\eta_0 = 1$. Therefore, $e_s' \to 1 + \frac{\phi a_s}{a_1 p_1 e_1 + a_2 p_2 e_2}$, which is a quadratic fixed-point equation. Accounting for the constraint that $e_s > 0$, the fixed-point equation requires that $\phi < 1$. Moreover, $\tau \to 1 - 1/\gamma$, which requires that $\gamma > 1$. We further observe that $\rho \to (\tau^2\bar{\mathrm{tr}}\,\gamma L^2 K^{-2})\rho$, which implies that $\rho \to 0$. We can then see that, for $c \in \{a_1, a_2\}$:

$$u_s \to \phi\gamma^2\tau^2 e_s^2 a_s(a_1 p_1 u_1 + a_2 p_2 u_2 + c)\bar{\mathrm{tr}}\,\Sigma^2 K^{-2} \tag{294}$$

$$= \frac{\phi e_s^2 a_s(a_1 p_1 u_1 + a_2 p_2 u_2 + c)}{(a_1 p_1 e_1 + a_2 p_2 e_2)^2}, \tag{295}$$

which is a linear fixed-point equation in $u_s$. In contrast, if $\theta_0 > 1$, we have $e_s' = 1 + \frac{\psi\tau a_s \eta\theta}{\lambda}$ and the equation:

$$\gamma\theta = \frac{\lambda}{(1 - \eta/\gamma)\left(\frac{a_1 p_1}{1 + \frac{\psi(1 - \eta/\gamma)a_1 \eta\theta}{\lambda}} + \frac{a_2 p_2}{1 + \frac{\psi(1 - \eta/\gamma)a_2 \eta\theta}{\lambda}}\right)}, \tag{296}$$

which is a quartic equation in $\eta$. This highlights the difficulties of rigorously isolating the effects of different components on bias amplification. We empirically investigate how different components (e.g., covariance structures, group sizes) affect bias amplification and minority-group bias in Sections 4 and 5, and extensively validate that our theory predicts these implications.

## I.2  THE GENERAL REGULARIZED CASE

We now consider the case where the covariance structure is the same for both groups, i.e., $\Sigma_1 = \Sigma_2 = \Sigma$. The calculations presented is this subsection are shared with Appendix H.2 of Dohmatob et al. (2024b), of which two authors of this work are also authors.

In this setting, it is clear that $e_1 = e_2 = e$ and $u_1 = u_2 = u$, where $(\tau, e, u, \rho)$ now satisfy:

$$1/e = 1 + \psi\tau\bar{\mathrm{tr}}\,\Sigma K^{-1}, \quad 1/\tau = 1 + \bar{\mathrm{tr}}\,K_0 K^{-1}, \text{ where } K_0 := e\Sigma, \ K := \gamma\tau K_0 + \lambda I_d, \tag{297}$$

$$u = \psi e^2\bar{\mathrm{tr}}\,\Sigma_1(\gamma\tau^2 L' + \rho I_d)K^{-2}, \ \rho = \tau^2\bar{\mathrm{tr}}\,(\gamma\rho K_0^2 + \lambda^2 L')K^{-2}, \ L' := (1 + u)\Sigma. \tag{298}$$

We first introduce some notation related to the degrees of freedom of $\Sigma$ before proceeding with our theoretical result.

**Definition I.1.** *Let* $\mathrm{df}_m(t) = \mathrm{tr}\,\Sigma^m(\Sigma + tI_d)^{-m}$ *for any positive integer* $m$. *Furthermore, define* $I_{a,b}(t) = \bar{\mathrm{tr}}\,\Sigma^a(\Sigma + tI_d)^{-b}$ *for any positive integers* $a, b$.

**Lemma I.1.** *The scalars* $u$ *and* $\rho' = \rho/(\gamma\tau^2)$ *solve the following pair of linear equations:*

$$u = \phi I_{2,2}(\theta)(1 + u) + \phi I_{1,2}(\theta)\rho',$$
$$\gamma\rho' = I_{2,2}(\theta)\rho' + \theta^2 I_{1,2}(\theta)(1 + u). \tag{299}$$

*Furthermore, the solutions can be explicitly represented as:*

$$u = \frac{\phi z}{\gamma - \phi z - I_{2,2}(\theta)}, \quad \rho' = \frac{\theta^2 I_{2,2}(\theta)}{\gamma - \phi z - I_{2,2}(\theta)}, \tag{300}$$

*where $z = I_{2,2}(\theta)(\gamma - I_{2,2}(\theta)) + \theta^2 I_{1,2}(\theta)^2$.*

*In particular, in the limit $\gamma \to \infty$, it holds that:*

$$\theta \simeq \kappa, \quad \rho' \to 0, \quad u \simeq \frac{\phi I_{2,2}(\kappa)}{1 - \phi I_{2,2}(\kappa)} \simeq \frac{\mathrm{df}_2(\kappa)/n}{1 - \mathrm{df}_2(\kappa)/n}, \tag{301}$$

*where $\kappa > 0$ is uniquely satisfies the fixed-point equation $\kappa - \lambda = \kappa \operatorname{tr} \Sigma(\Sigma + \kappa I_d)^{-1}/n$.*

*Proof.* The equations defining these scalars are:

$$u = \psi e^2 \bar{\operatorname{tr}} \, \Sigma(\gamma\tau^2 L' + \rho I_d) K^{-2}, \tag{302}$$
$$\rho = \tau^2 \bar{\operatorname{tr}} \, (\gamma\rho K_0^2 + \lambda^2 L') K^{-2}, \tag{303}$$

where $K_0 = e\Sigma$, $K = \gamma\tau K_0 + \lambda I_d$, and $L' := u\Sigma + B$. Further, since $B = \Sigma$, we have $L' = (1 + u)\Sigma$. Now, we can rewrite the previous equations like so

$$u = \psi e^2 \bar{\operatorname{tr}} \, \Sigma(\gamma\tau^2(1 + u)\Sigma + \rho I_d) K^{-2} = \phi\gamma^2\tau^2 e^2(1 + u)\bar{\operatorname{tr}} \, \Sigma^2 K^{-2} + \phi\gamma e^2 \rho \bar{\operatorname{tr}} \, \Sigma K^{-2},$$
$$\rho = \tau^2 \bar{\operatorname{tr}} \, (\gamma\rho e^2 \Sigma^2 + \lambda^2(1 + u)\Sigma) K^{-2} = \gamma\tau^2 e^2 \rho \bar{\operatorname{tr}} \, \Sigma^2 K^{-2} + \lambda^2\tau^2(1 + u)\bar{\operatorname{tr}} \, \Sigma K^{-2}.$$

This can be equivalently written as:

$$u = \phi(1 + u)\gamma^2\tau^2 e^2 \bar{\operatorname{tr}} \, \Sigma^2 K^{-2} + \phi\rho'\gamma^2\tau^2 e^2 \bar{\operatorname{tr}} \, \Sigma K^{-2}, \tag{304}$$
$$\gamma\rho' = \rho'\gamma^2\tau^2 e^2 \bar{\operatorname{tr}} \, \Sigma^2 K^{-2} + (1 + u)\lambda^2 \bar{\operatorname{tr}} \, \Sigma K^{-2}. \tag{305}$$

Now, observe that:

$$\tau^2 e^2 \bar{\operatorname{tr}} \, \Sigma^2 K^{-2} = \bar{\operatorname{tr}} \, \Sigma^2(\Sigma + \theta I_d)^{-2}/\gamma^2 = I_{2,2}(\theta)/\gamma^2, \tag{306}$$
$$\tau^2 e^2 \bar{\operatorname{tr}} \, \Sigma K^{-2} = \bar{\operatorname{tr}} \, \Sigma(\Sigma + \theta I_d)^{-2}/\gamma^2 = I_{1,2}(\theta)/\gamma^2, \tag{307}$$
$$\lambda^2 \bar{\operatorname{tr}} \, \Sigma K^{-2} = \theta^2 \bar{\operatorname{tr}} \, \Sigma(\Sigma + \theta I_d)^{-2} = \theta^2 I_{1,2}(\theta), \tag{308}$$
$$e^2 \bar{\operatorname{tr}} \, \Sigma K^{-2} = \bar{\operatorname{tr}} \, \Sigma(\Sigma + \theta I_d)^{-2}/(\gamma\tau)^2 = I_{1,2}(\theta)/(\gamma\tau)^2, \tag{309}$$
$$\tau^2 \bar{\operatorname{tr}} \, \Sigma K^{-2} = \bar{\operatorname{tr}} \, \Sigma(\Sigma + \theta I_d)^{-2}/(\gamma e)^2 = I_{1,2}(\theta)/(\gamma e)^2, \tag{310}$$

where we have used the definition $\theta = \lambda/(\gamma\tau e)$. Thus, $u$ and $\rho$ have limiting values which solve the system of linear equations:

$$u = \psi\gamma \cdot \gamma^{-2} I_{2,2}(\theta)(1 + u) + \psi\gamma \cdot \gamma^{-2} I_{1,2}\rho' = \phi I_{2,2}(\theta)(1 + u) + \phi I_{1,2}(\theta)\rho',$$
$$\gamma\rho' = I_{2,2}(\theta)\rho' + \theta^2 I_{1,2}(\theta)(1 + u) = I_{2,2}(\theta)\rho' + \theta^2 I_{1,2}(\theta)(1 + u),$$

where we have used the identity $\phi\gamma = \psi$. These correspond exactly to the equations given in the lemma. This proves the first part.

For the second part, indeed, $\tau = 1 - \eta_0/\gamma \to 1$ in the limit $\gamma \to \infty$, and so $\theta \simeq \lambda/(\gamma e)$ which verifies the equation:

$$\theta \simeq \lambda + \lambda\psi\bar{\operatorname{tr}} \, \Sigma(\gamma e\Sigma + \lambda)^{-1} = \lambda + \phi \cdot \frac{\lambda}{\gamma e}\bar{\operatorname{tr}} \, \Sigma(\Sigma + \frac{\lambda}{\gamma e}I_d)^{-1} \simeq \lambda + \theta \operatorname{tr} \Sigma(\Sigma + \theta I_d)^{-1}/n,$$

i.e., $\theta \simeq \lambda + \theta \, \mathrm{df}_1(\theta)/n$ and $\theta > 0$. By comparing with the equation $\kappa - \lambda = \kappa \, \mathrm{df}_1(\kappa)/n$ satisfied by $\kappa > 0$ in Definition D.2, we conclude $\theta \simeq \kappa$.

Now, Equation 299 becomes $\rho' = 0$, and $u = \phi I_{2,2}(\kappa)(1 + u)$, i.e.,

$$u = \frac{\phi I_{2,2}(\kappa)}{1 - \phi I_{2,2}(\kappa)} \simeq \frac{\mathrm{df}_2(\kappa)/n}{1 - \mathrm{df}_2(\kappa)/n},$$

as claimed. □

### I.3 UNREGULARIZED LIMIT

The calculations presented in this subsection are shared with Appendix H.2 of Dohmatob et al. (2024b), of which two authors of this work are also authors.

Define the following auxiliary quantities:

$$\theta := \frac{\lambda}{\gamma \tau e}, \quad \chi := \frac{\lambda}{\tau}, \quad \kappa := \frac{\lambda}{e}. \tag{311}$$

where $\tau$, $e$, $u$, and $\rho$ are as previously defined.

**Lemma I.2.** *In the limit $\lambda \to 0^+$, we have the following analytic formulae:*

$$\chi \to \chi_0 = (1 - \psi)_+ \cdot \gamma \theta_0, \tag{312}$$
$$\kappa \to \kappa_0 = (\psi - 1)_+ \cdot \theta_0/\phi, \tag{313}$$
$$\tau \to \tau_0 = 1 - \eta_0/\gamma, \tag{314}$$
$$e \to e_0 = 1 - \phi \eta_0, \tag{315}$$

*where $\theta_0$ is the unique positive solution of the fixed-point equation $\eta_0 = I_{1,1}(\theta_0)$.*

*Proof.* Observe that $K_0 = e\Sigma$ and $K = \gamma \tau K_0 + \lambda I_d = \gamma \tau e \cdot (\Sigma + \theta I_d)$. Defining $\eta := I_{1,1}(\theta)$, one can then rewrite the equations defining $e$ and $\tau$ as follows:

$$e' = \frac{\lambda}{e} = \lambda + \psi \tau \lambda \bar{\text{tr}} \Sigma K^{-1} = \lambda + \frac{\psi \tau \lambda}{\gamma \tau e} \bar{\text{tr}} \Sigma (\Sigma + \theta I_d)^{-1} = \lambda + \phi \eta e', \tag{316}$$

$$\tau' = \frac{\lambda}{\tau} = \lambda + \lambda \bar{\text{tr}} K_0 K^{-1} = \lambda + \frac{\lambda e}{\gamma \tau e} \bar{\text{tr}} \Sigma (\Sigma + \theta I_d)^{-1} = \lambda + (\eta/\gamma) \tau'. \tag{317}$$

We deduce that:

$$e' = \frac{\lambda}{1 - \phi \eta}, \quad \tau' = \frac{\lambda}{1 - \eta/\gamma}, \quad \tau' e' = \lambda \gamma \theta. \tag{318}$$

In particular, the above means that $\eta \le \min(\gamma, 1/\phi)$. The last part of equations Equation 318 can be rewritten as follows:

$$\frac{\lambda}{(1 - \phi \eta)(1 - \eta/\gamma)} = \gamma \theta, \text{ i.e., } \phi \eta^2 - (\phi \gamma + 1)\eta + \gamma - \frac{\lambda}{\theta} = 0. \tag{319}$$

This is a quadratic equation for $\eta$ as a function of $\lambda$ and $\theta$, with roots

$$\eta^{\pm} = \frac{\phi \gamma + 1 \pm \sqrt{(\phi \gamma + 1)^2 - 4(\phi \gamma - (\phi/\theta)\lambda)}}{2\phi} = \frac{\psi + 1 \pm \sqrt{(\psi + 1)^2 - 4(\psi - \phi/\theta')}}{2\phi}. \tag{320}$$

Now, for small $\lambda > 0$ and $\psi \ne 1$, we can do a Taylor expansion to get:

$$\eta^{\pm} \simeq \frac{\psi + 1 \pm |\psi - 1|}{2\phi} \pm \frac{1}{\theta|\psi - 1|}\lambda + O(\lambda^2).$$

More explicitly:

$$\eta^+ \simeq O(\lambda^2) \begin{cases} 1/\phi + \lambda/((1 - \psi)\theta), & \text{if } \psi < 1 \\ \gamma + \lambda/((\psi - 1)\theta), & \text{if } \psi > 1 \end{cases},$$

$$\eta^- \simeq O(\lambda^2) + \begin{cases} \gamma - \lambda/((1 - \psi)\theta), & \text{if } \psi < 1 \\ 1/\phi - \lambda/((\psi - 1)\theta), & \text{if } \psi > 1 \end{cases}.$$

Because $\eta \le \min(1, 1/\phi, \gamma)$, we must have the expansion:

$$\eta \simeq O(\lambda^2) + \begin{cases} \gamma - \lambda/((1 - \psi)\theta), & \text{if } \psi < 1 \\ 1/\phi + \lambda/((\psi - 1)\theta), & \text{if } \psi > 1 \end{cases},$$

$$= \eta_0 - \frac{1}{(1 - \psi)\theta_0}\lambda + O(\lambda^2), \tag{321}$$

provided $\theta_0 > 0$, i.e $\eta_0 \neq 1$. in this regime, we obtain:

$$\tau' = \frac{\lambda}{1 - \eta/\gamma} \simeq \begin{cases} \lambda/(1 - 1 + \lambda/((1 - \psi)\gamma\theta_0)) = (1 - \psi)\gamma\theta_0, & \text{if } \psi \leq 1 \\ \lambda/(1 - 1/\psi + o(1)) \to 0, & \text{if } \psi > 1 \end{cases},$$

$$e' = \frac{\lambda}{1 - \phi\eta} \simeq \begin{cases} \lambda/(1 - \psi + o(1)) \to 0, & \text{if } \psi \leq 1 \\ \lambda/(1 - 1 + \lambda\phi/((\psi - 1)\theta_0) \to (\psi - 1)\theta_0/\phi, & \text{if } \psi > 1 \end{cases},$$

$$\tau = 1 - \eta/\gamma \simeq 1 - \eta_0/\gamma = (1 - 1/\psi)_+,$$

$$e = 1 - \phi\eta \simeq 1 - \phi\eta_0 = (1 - \psi)_+.$$

On the other hand, if $\theta_0 = 0$ (which only happens if $\psi < 1$ and $\gamma > 1$, or $\psi \geq 1$ and $\phi \leq 1$), it is easy to see from Equation 318 that we must have $\tau' \to 0$, $e' \to 0$, $\tau \to 1 - 1/\gamma$, $e \to 1 - \phi \geq 0$. $\quad\square$

## J    COROLLARY J.1

As a special case of Theorem 3.1, we recover Corollary J.1, which aligns with Proposition 4 from (Bach, 2023). Theorem 3.1 is a non-trivial generalization of Proposition 4.

Corollary J.1 captures how the covariance matrix affects the test risk of a model through the normalized second and first-order degrees of freedom of $\Sigma_s$. Corollary J.1 also reveals that in the underparameterized regime ($\psi_s < 1$), the bias and variance of the test risk of the model strictly increase as a function of $\psi_s$ (rate of parameters to samples); the test risk of the model explodes (i.e., there is catastrophic overfitting (Bach, 2023)) when $\psi_s$ gets close to 1. In the overparameterized regime ($\psi_s > 1$), the bias and variance of the test risk decrease as $\psi_s$ increases.

**Corollary J.1.** *Under Assumptions B.2 and 3.1, it holds in the unregularized setting $\lambda_s \to 0^+$ that*

$$
B_s(\widehat{f}_s) = \begin{cases} \frac{\theta_0 \bar{\mathrm{tr}}\,\Theta_s \Sigma_s (\Sigma_s + \theta_0 I_d)^{-1}}{1 - \psi_s}, & \gamma, \psi_s < 1 \\ 0, & \psi_s < 1, \gamma \geq 1 \text{ or } 1 \leq \psi_s \leq \gamma\,, \\ \frac{\theta_0^2 \bar{\mathrm{tr}}\,\Theta_s \Sigma_s (\Sigma_s + \theta_0 I_d)^{-2}}{1 - \phi_s I_{2,2}(\theta_0)} + \frac{\theta_0 \bar{\mathrm{tr}}\,\Theta_s \Sigma_s (\Sigma_s + \theta_0 I_d)^{-1}}{\psi_s - 1}, & \psi_s \geq 1, \psi_s \geq \gamma \end{cases}
\tag{322}
$$

$$
V_s(\widehat{f}_s) = \begin{cases} \frac{\sigma_s^2 \psi_s}{1 - \psi_s}, & \gamma, \psi_s < 1 \\ \frac{\sigma_s^2 \phi_s}{1 - \phi_s}, & \psi_s < 1, \gamma \geq 1 \text{ or } 1 \leq \psi_s \leq \gamma\,, \\ \frac{\sigma_s^2 \phi_s I_{2,2}(\theta_0)}{1 - \phi_s I_{2,2}(\theta_0)} + \frac{\sigma_s^2}{\psi_s - 1}, & \psi_s \geq 1, \psi_s \geq \gamma \end{cases}
\tag{323}
$$

*where $I_{a,b}(t) = \bar{\mathrm{tr}}\,\Sigma^a (\Sigma + t I_d)^{-b}$ for any positive integers $a, b$; and $\theta_0$ is the unique solution to the following non-linear equation:*

$$
I_{1,1}(\theta_0) = \begin{cases} \gamma, & \gamma, \psi_s < 1 \\ 1, & \psi_s < 1, \gamma \geq 1 \text{ or } 1 \leq \psi_s \leq \gamma\,. \\ 1/\phi_s, & \psi_s \geq 1, \psi_s \geq \gamma \end{cases}
\tag{324}
$$

*Proof.* Define $e' = 1/e_s \geq 0$, $\tau' = 1/\tau_s \geq 0$, $\theta = \lambda_s \tau' e'/\gamma$, and $\eta = I_{1,1}(\theta) \in [0,1]$. One can then express $e'$ and $\tau'$ as:

$$
e' = 1 + \psi \tau_s \bar{\mathrm{tr}}\,\Sigma (\gamma \tau_s e_s \Sigma + \lambda_s I_d)^{-1} = 1 + \phi_s \eta e',
\tag{325}
$$

$$
\tau' = 1 + \bar{\mathrm{tr}}\,e_s \Sigma (\gamma \tau_s e_s \Sigma + \lambda_s I_d)^{-1} = 1 + (\eta/\gamma)\tau'.
\tag{326}
$$

We deduce that:

$$
e' = \frac{1}{1 - \phi_s \eta},
\tag{327}
$$

$$
\tau' = \frac{1}{1 - \eta/\gamma},
\tag{328}
$$

$$
\lambda \tau' e' = \gamma \theta.
\tag{329}
$$

We define the following limiting values:

$$
\lim_{\lambda_s \to 0^+} \theta \to \theta_0, \quad \lim_{\lambda_s \to 0^+} \eta \to \eta_0,
\tag{330}
$$

$$
\lim_{\lambda_s \to 0^+} e_s \to e_0, \quad \lim_{\lambda_s \to 0^+} \tau_s \to \tau_0,
\tag{331}
$$

$$
\lim_{\lambda_s \to 0^+} u_s \to u_0, \quad \lim_{\lambda_s \to 0^+} \rho_s \to \rho_0.
\tag{332}
$$

There are now two cases to consider.

### J.1    CASE 1: $\theta_0 = 0$

This implies $\eta_0 = 1$. Therefore, by simple computation, $e_0 = 1/e_0' = 1 - \phi_s \eta_0 = 1 - \phi_s$ and $\tau_0 = 1/\tau_0' = 1 - 1/\gamma$. This requires $\phi_s \leq 1$ and $\gamma \geq 1$.

## J.2 CASE 2: $\theta_0 > 0$

Equation 329 can be re-written as:

$$\frac{\lambda_s}{(1 - \phi_s \eta)(1 - \eta/\gamma)} = \gamma \theta, \text{ i.e., } \phi_s \eta^2 - (\psi_s + 1)\eta + \gamma - \frac{\lambda_s}{\theta} = 0. \tag{333}$$

We solve this quadratic equation for $\eta$, arriving at the solutions:

$$\eta^{\pm} = \frac{\psi_s + 1 \pm \sqrt{(\psi_s + 1)^2 - 4(\psi_s - (\phi_s/\theta)\lambda_s)}}{2\phi_s} = \frac{\psi_s + 1 \pm \sqrt{(\psi_s + 1)^2 - 4(\psi_s - (\phi_s/\theta)\lambda_s)}}{2\phi_s}. \tag{334}$$

Taking the limit of $\eta^{\pm}$ as $\lambda_s \to 0^+$ gives:

$$
\begin{aligned}
\eta^+ &\to \frac{\psi_s + 1 + |\psi_s - 1|}{2\phi_s} = \begin{cases} \psi_s/\phi_s = \gamma, & \text{if } \psi_s \geq 1, \\ 1/\phi_s, & \text{if } \psi_s < 1, \end{cases} \\
\eta^- &\to \frac{\psi_s + 1 - |\psi_s - 1|}{2\phi_s} = \begin{cases} 1/\phi_s, & \text{if } \psi_s \geq 1, \\ \psi_s/\phi_s = \gamma, & \text{if } \psi_s < 1. \end{cases}
\end{aligned} \tag{335}
$$

Recall that we have the following constraints:

- $e' \geq 0, \tau' \geq 0$.

- $\eta \in [0, 1]$.

We can show that $\eta_0 = 1/\phi_s$ is incompatible with $\psi_s < 1$. Indeed, otherwise we would have $\tau_0' = 1/(1 - \eta_0/\gamma) = 1/(1 - 1/\psi_s) < 0$. Similarly, if $\psi_s > 1$, we would have $e_0 = 1 - \phi_s \gamma = 1 - \psi_s < 0$. Therefore, $\eta_0 = \eta^-$. Furthermore, if $\psi_s, \gamma < 1$, it must be that $\theta_0 > 0$ and $\eta_0 = \gamma$. Instead, if $\psi_s < 1, \gamma \geq 1$, we must have that $\phi_s \leq 1$, and therefore, $\theta_0 = 0$ and $\eta_0 = 1$. Similarly, if $\psi_s \geq 1, \gamma \geq 1$, and $\phi_s \leq 1$ (i.e., $1 \leq \psi_s \leq \gamma$), we must have that $\theta_0 = 0$ and $\eta_0 = 1$. In all other cases where $\psi_s \geq 1$, it must be that $\eta_0 = 1/\phi_s$ (which additionally requires $\phi_s \geq 1$ or $\psi_s \geq \gamma$). Succinctly:

$$\eta_0 = \begin{cases} \gamma, & \gamma, \psi_s < 1 \\ 1, & \psi_s < 1, \gamma \geq 1 \text{ or } 1 \leq \psi_s \leq \gamma \\ 1/\phi_s, & \psi_s \geq 1, \psi_s \geq \gamma \end{cases}. \tag{336}$$

Plugging this into Equation 327 and Equation 328 gives:

$$e_0 = 1 - \phi_s \eta_0 = 1 - \phi_s I_{1,1}(\theta_0), \tag{337}$$

$$\tau_0 = 1 - \eta_0/\gamma = 1 - I_{1,1}(\theta_0)/\gamma. \tag{338}$$

We will now solve for $u_0$ and $\rho_0/\tau_0^2$. We can re-write $u_s$ and $\rho_s/\tau_s^2$ as:

$$\rho_s/\tau_s^2 = \gamma^{-1}(\rho_s/\tau_s^2)I_{2,2}(\theta) + \theta^2(u_s + 1)I_{1,2}(\theta), \tag{339}$$

$$\tau_s^2 u_s = \tau_s^2 \phi_s(u_s + 1)I_{2,2}(\theta) + \phi_s \gamma^{-1}\rho_s I_{1,2}(\theta). \tag{340}$$

Solving for $u_0$ and $\rho_0/\tau_0^2$ yields:

$$u_0 = \frac{\phi\zeta}{\gamma - \phi\zeta - I_{2,2}(\theta_0)}, \quad \rho_0/\tau_0^2 = \frac{\gamma \theta_0^2 I_{2,2}(\theta_0)}{\gamma - \phi\zeta - I_{1,2}(\theta_0)}, \tag{341}$$

$$\text{where } \zeta = I_{2,2}(\theta_0)(\gamma - I_{2,2}(\theta_0)) + \theta_0^2 I_{1,2}(\theta_0)^2. \tag{342}$$

We can then see for the variance term that:

$$V_s(\widehat{f}_s) = \sigma_s^2 \phi_s \gamma \bar{\text{tr}} \, \Sigma_s (\gamma e_s \tau_s^2 \Sigma_s + e_s \rho_s I_d - \lambda_s u_s \tau_s I_d)(\gamma \tau_s e_s)^{-2}(\Sigma_s + \theta I_d)^{-2} \tag{343}$$

$$= \sigma_s^2 \phi_s (1/e_s) \bar{\text{tr}} \, \Sigma_s^2 (\Sigma_s + \theta I_d)^{-2} + (\sigma_s^2 \phi_s / \gamma)(1/e_s)(\rho_s / \tau_s^2) \bar{\text{tr}} \, \Sigma_s (\Sigma_s + \theta I_d)^{-2} \tag{344}$$

$$- \sigma_s^2 \phi_s (u_s)(1/e_s) \theta \bar{\text{tr}} \, \Sigma_s (\Sigma_s + \theta I_d)^{-2} \tag{345}$$

$$= \sigma_s^2 \phi_s I_{2,2}(\theta)/e_s + \sigma_s^2 \phi_s (\rho_s / \tau_s^2) I_{1,2}(\theta)/(\gamma e_s) - \sigma_s^2 \phi_s u_s \theta I_{1,2}(\theta)/e_s \tag{346}$$

$$\to \frac{\sigma_s^2 \phi_s I_{2,2}(\theta_0) - \sigma_s^2 \phi_s u_0 \theta_0 I_{1,2}(\theta_0)}{1 - \phi_s I_{1,1}(\theta_0)} + \frac{\sigma_s^2 \phi_s \rho_0 / \tau_0^2}{\gamma(1 - \phi_s I_{1,1}(\theta_0))} \tag{347}$$

$$= -\frac{\sigma_s^2 \phi_s \xi}{\phi_s \xi + I_{2,2}(\theta_0) - \gamma}, \tag{348}$$

where $\xi = I_{1,1}^2(\theta_0) - 2I_{1,1}(\theta_0)I_{2,2}(\theta_0) + I_{2,2}(\theta_0)\gamma$ and we have used the fact that $I_{1,2}(\theta) = (I_{1,1}(\theta) - I_{2,2}(\theta))/\theta$. Plugging in $I_{1,1}(\theta_0) = \eta_0$, we have that:

$$V_s(\widehat{f}_s) \to \begin{cases} \frac{\sigma_s^2 \psi_s}{1 - \psi_s}, & \gamma, \psi_s < 1 \\ \frac{\sigma_s^2 \phi_s}{1 - \phi_s}, & \psi_s < 1, \gamma \geq 1 \text{ or } 1 \leq \psi_s \leq \gamma \,, \\ \frac{\sigma_s^2 \phi_s I_{2,2}(\theta_0)}{1 - \phi_s I_{2,2}(\theta_0)} + \frac{\sigma_s^2}{\psi_s - 1}, & \psi_s \geq 1, \psi_s \geq \gamma \end{cases} \tag{349}$$

where we have used that $I_{2,2}(\theta_0) = I_{2,2}(0) = 1$ in the second case.

Likewise, for the bias term, we obtain:

$$B_s(\widehat{f}_s) = \bar{\text{tr}} \, \Theta_s \Sigma_s + \bar{\text{tr}} \, \Theta_s \Sigma_s (\gamma e_s^2 \Sigma_s (\gamma \tau_s^2 \Sigma_s + \rho_s I_d) + \lambda_s^2 u_s I_d)(\gamma \tau_s e_s \Sigma_s + \lambda_s I_d)^{-2} \tag{350}$$

$$- 2\gamma e_s \tau_s \bar{\text{tr}} \, \Theta_s \Sigma_s^2 (\gamma \tau_s e_s \Sigma_s + \lambda_s I_d)^{-1} \tag{351}$$

$$\to \bar{\text{tr}} \, \Theta_s \Sigma_s (\Sigma_s^2 + 2\theta_0 \Sigma_s + \theta_0^2 I_d)(\Sigma_s + \theta_0 I_d)^{-2} \tag{352}$$

$$+ \bar{\text{tr}} \, \Theta_s \Sigma_s (\Sigma_s^2)(\Sigma_s + \theta_0 I_d)^{-2} \tag{353}$$

$$+ \bar{\text{tr}} \, \Theta_s \Sigma_s ((\rho_0 / \tau_0^2)\Sigma_s / \gamma)(\Sigma_s + \theta_0 I_d)^{-2} \tag{354}$$

$$+ \bar{\text{tr}} \, \Theta_s \Sigma_s (\theta_0^2 u_0 I_d)(\Sigma_s + \theta_0 I_d)^{-2} \tag{355}$$

$$+ \bar{\text{tr}} \, \Theta_s \Sigma_s (-2\Sigma_s^2 - 2\theta_0 \Sigma_s)(\Sigma_s + \theta_0 I_d)^{-2} \tag{356}$$

$$= \theta_0^2 (u_0 + 1)\bar{\text{tr}} \, \Theta_s \Sigma_s (\Sigma_s + \theta_0 I_d)^{-2} + (1/\gamma)(\rho_0 / \tau_0^2)\bar{\text{tr}} \, \Theta_s \Sigma_s^2 (\Sigma_s + \theta_0 I_d)^{-2}. \tag{357}$$

Again, plugging in $I_{1,1}(\theta_0) = \eta_0$, we have that:

$$B_s(\widehat{f}_s) \to \begin{cases} \frac{\theta_0 \bar{\text{tr}} \, \Theta_s \Sigma_s (\Sigma_s + \theta_0 I_d)^{-1}}{1 - \psi_s}, & \gamma, \psi_s < 1 \\ \frac{\theta_0^2 \bar{\text{tr}} \, \Theta_s \Sigma_s (\Sigma_s + \theta_0 I_d)^{-2}}{1 - \phi_s} = 0, & \psi_s < 1, \gamma \geq 1 \text{ or } 1 \leq \psi_s \leq \gamma \,, \\ \frac{\theta_0^2 \bar{\text{tr}} \, \Theta_s \Sigma_s (\Sigma_s + \theta_0 I_d)^{-2}}{1 - \phi_s I_{2,2}(\theta_0)} + \frac{\theta_0 \bar{\text{tr}} \, \Theta_s \Sigma_s (\Sigma_s + \theta_0 I_d)^{-1}}{\psi_s - 1}, & \psi_s \geq 1, \psi_s \geq \gamma \end{cases} \tag{358}$$

where we have used that $\bar{\text{tr}} \, \Theta_s \Sigma_s^2 (\Sigma_s + \theta_0 I_d)^{-2} = \bar{\text{tr}} \, \Theta_s \Sigma_s (\Sigma_s + \theta_0 I_d)^{-1} - \theta_0 \bar{\text{tr}} \, \Theta_s \Sigma_s (\Sigma_s + \theta_0 I_d)^{-2}$ and in the second case, $\theta_0 = 0$ and $I_{2,2}(\theta_0) = 1$. $\qquad \square$

## K    EXPERIMENTAL DETAILS

### K.1    SYNTHETIC EXPERIMENTS

Across all experiments on synthetic data, we choose $n = 400$. We further use 5 runs to estimate test risks (e.g., $\mathbb{E}R_s(\widehat{f}), \mathbb{E}R_s(\widehat{f}_s)$), and 5 runs to capture the variance of the estimators, for a total of 25 runs. We use 10,000 samples to estimate test risks.

Our experiments validate that bias amplification occurs even in low-dimensional regimes. In Sections 4 and 5, and Appendices L, O, and Q, we show that our theory predicts bias amplification for models trained on only $n = 400$ samples. The high-dimensional regime is commonly studied in ML theory and statistical physics (as we mention in Section 1.2), as it makes precise analysis more tractable.

**Setup for Section 4.1.**    We further choose $\lambda = 1 \times 10^{-6}$ to approximate the minimum-norm interpolator; we henceforth set $\lambda = \lambda_1 = \lambda_2$ for simplicity. We modulate $a_1, a_2, \sigma_1^2, \sigma_2^2$, as well as $\psi$ (rate of parameters to samples) and $\phi$ (rate of features to samples) to understand the effects of model size, number of features, and sample size on bias amplification. We consider diverse and dense values of these variables to obtain a clear picture of when and how models amplify bias.

**Setup for Section 5.**    The first $\pi d$ features represent common *core* features of groups 1 and 2 while the latter $(1 - \pi)d$ features capture unshared *extraneous* features for group 2 (e.g., spurious features). Intuitively, this setting can model: (1) learning from data from two groups where one group suffers from spurious features (Sagawa et al., 2020), or (2) learning from a mixture of raw data (i.e., with spurious features) and clean data (i.e., without spurious features) for a single population (Khani & Liang, 2021). We ask: Does our theory predict how the inclusion of different amounts of extraneous features affect the test risk of a minority group (compared to the majority group) when a single model is trained on data from both groups vs. a separate model is trained per group?

Although Sagawa et al. (2020) consider classification instead of regression, to mirror their experimental setting, we pick $p_1 = 0.9$ (i.e., group 1 is much larger than group 2) and $\Theta = I_d, \Delta = 0$ (i.e., $w_1^* = w_2^*$). We additionally choose $\lambda = 1 \times 10^{-6}$ and $\sigma_1^2 = \sigma_2^2 = 1$. We modulate $a_1, b_2$, as well as $\psi$ (rate of parameters to samples) and $\phi$ (rate of features to samples). Notably, this setting also captures learning problems with $o(d)$ overlapping core and extraneous features in our asymptotic scaling limit. An extremization of this setting is choosing $\Sigma_1 = a_1 I_{\pi d} \oplus 0 I_{(1-\pi)d}, \Sigma_2 = 0 I_{\pi d} \oplus b_2 I_{(1-\pi)d}$, where groups 1 and 2 have no overlapping features.

The experiments in Section 5 validate that our analysis does not rely on conditional dependence heterogeneity. That is, we empirically verify that our theory still holds and predicts bias amplification occurs even when $w_1^* = w_2^*$ (see Figure 4). In essence, the structure and eigenspectra of the covariance matrices of the two groups still contribute to bias amplification even when the ground-truth weights for the groups are the same. In our theory, we only allow the possibility of $w_1^* \neq w_2^*$ to be as general as possible. In practice, labeling rule heterogeneity may be leveraged, for example, to train a mixture of experts that is regularized to deamplify bias.

**Extraneous vs. Spurious Features.**    Our usage of extraneous features (i.e., features that are different across groups and correlated with labels) differs from classical definitions of spuriousness (i.e., non-causal correlations between features and labels) (Bell & Wang, 2024); indeed, the extraneous features are used to generate the labels of the minority group. For example, Khani & Liang (2021) model both the labels and spurious features in linear regression as being separately generated by the core features, such that the labels and spurious features are associated but not causally related. However, this setup is not encompassed by our modeling assumptions, as it entails that the ground-truth parameter and feature covariance matrices are not jointly diagonalizable. In contrast, Sagawa et al. (2020) study spurious correlations in classification. At a high level, Sagawa et al. (2020) create four subgroups of a population with different combinations of class labels $y \in \{-1, 1\}$ and group labels $a \in \{-1, 1\}$. The core and spurious features are then sampled from normal distributions parameterized by $y, a$ (respectively) and different variance levels. By setting the spurious features to have a significantly lower variance than the core features and making $y$ and $a$ highly associated (i.e., imbalanced groups), the authors coerce models to perform classification as a function of primarily the spurious features of the majority group, which does not generalize to the minority group. To capture this spirit, our setup uses imbalanced groups, and the data for the majority group provides no learning signal for the

extraneous features; this coerces models to perform regression as a function of primarily the core features, without learning appropriate parameters for the extraneous features, and thus generalize poorly to the minority group.

## K.2 COLORED MNIST EXPERIMENTS

**Train-test split.** Colored MNIST has a total of 60k instances. Each image is $28 \times 28 \times 3$ pixels. We use the prescribed 0.67-0.33 train-test split. We do not perform validation of hyperparameters, which we mostly adopt[3].

**Model architecture.** By default, our CNN architecture consists of: (1) a convolutional layer (3 in-channels, 20 out-channels, kernel size of 5, stride of 1); (2) a max pooling layer (kernel size of 2, stride of 2); (3) a second convolutional layer (20 in-channels, 50 out-channels, kernel size of 5, stride of 1); (4) a second max-pooling layer (kernel size of 2, stride of 2); (5) a fully-connected layer ($\mathbb{R}^{800} \to \mathbb{R}^{500}$); and (6) a second fully-connected layer ($\mathbb{R}^{500} \to \mathbb{R}^1$).

**Model training.** We train each model with a batch size of 250 for a single epoch with respect to groups (i.e., 80 training steps given there are two groups). We use a cross-entropy loss and the Adam optimizer with learning rate 0.01. We run all experiments on a single NVIDIA L40S. We report our results over 10 random seeds.

---

[3]https://colab.research.google.com/github/reiinakano/invariant-risk-minimization/blob/master/invariant_risk_minimization_colored_mnist.ipynb

## L    Bias Amplification Plots

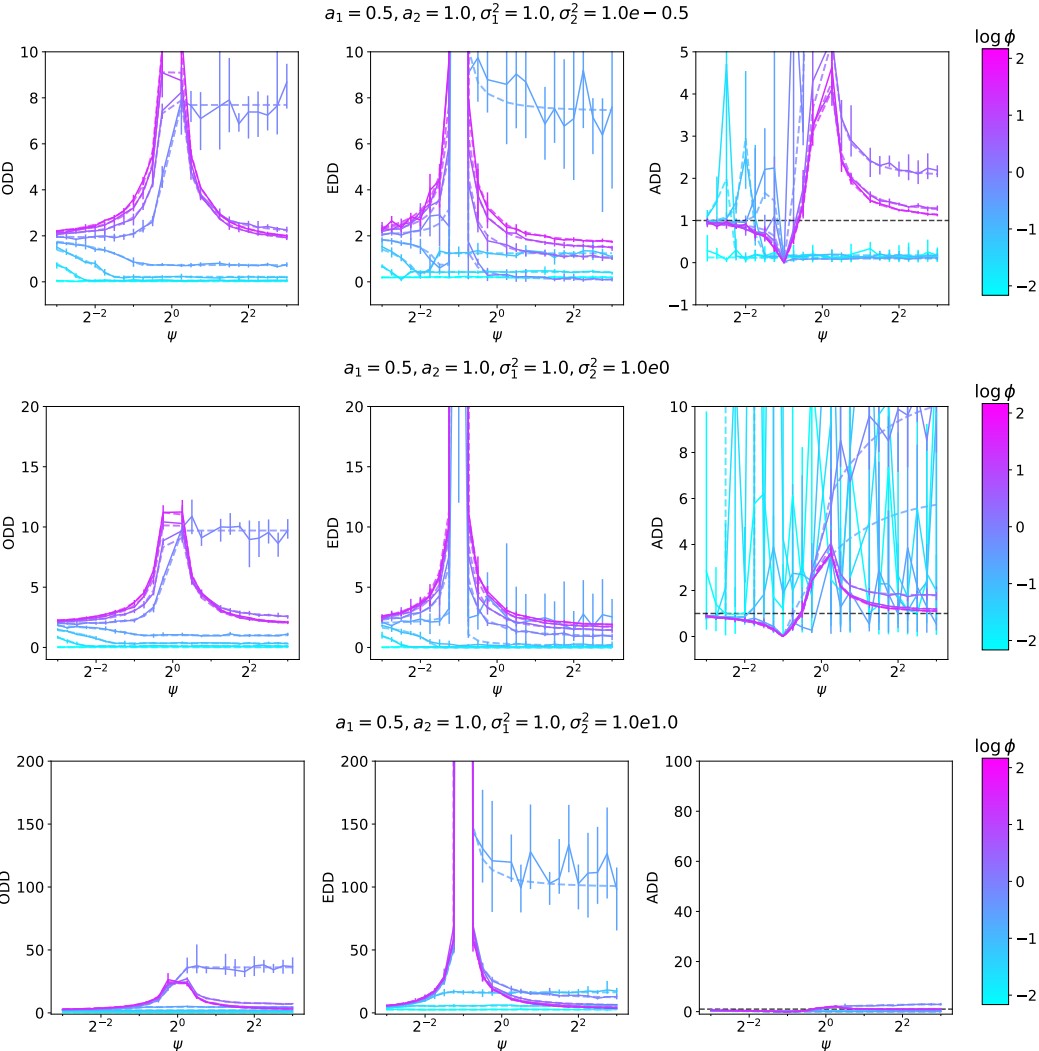

Figure 6: We empirically demonstrate that bias amplification occurs and validate our theory (Theorems 3.1 and 3.2) for $ODD$, $EDD$, and $ADD$ under the setup described in Section 4.1. The solid lines capture empirical values while the corresponding lower-opacity dashed lines represent what our theory predicts. We plot $ODD$ and $EDD$ on the same scale for easy comparison, and include a black dashed line at $ADD = 1$ to contrast bias amplification vs. deamplification. The error bars capture the range of the estimators over 25 random seeds.

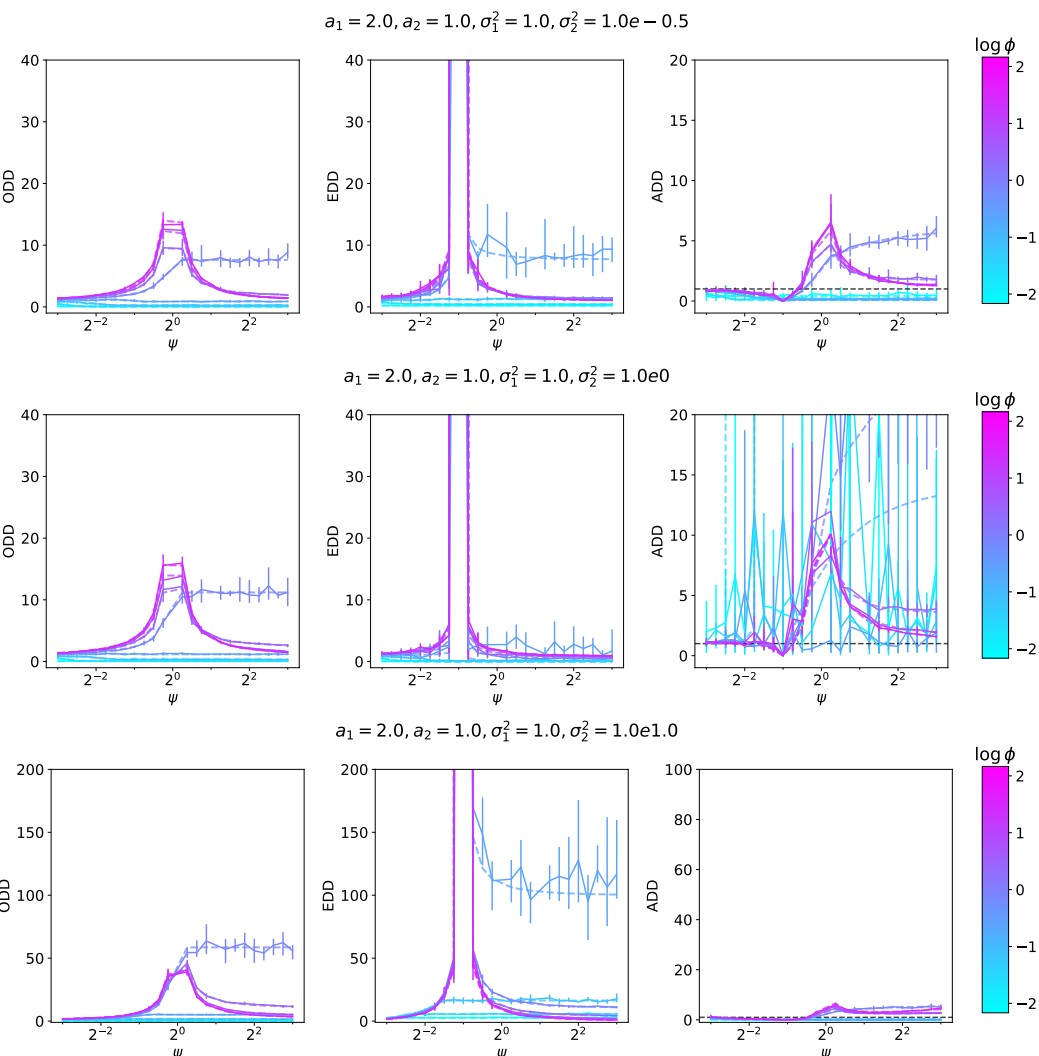

Figure 7: We empirically demonstrate that bias amplification occurs and validate our theory (Theorems 3.1 and 3.2) for $ODD$, $EDD$, and $ADD$ under the setup described in Section 4.1. The solid lines capture empirical values while the corresponding lower-opacity dashed lines represent what our theory predicts. We plot $ODD$ and $EDD$ on the same scale for easy comparison, and include a black dashed line at $ADD = 1$ to contrast bias amplification vs. deamplification. The error bars capture the range of the estimators over 25 random seeds.

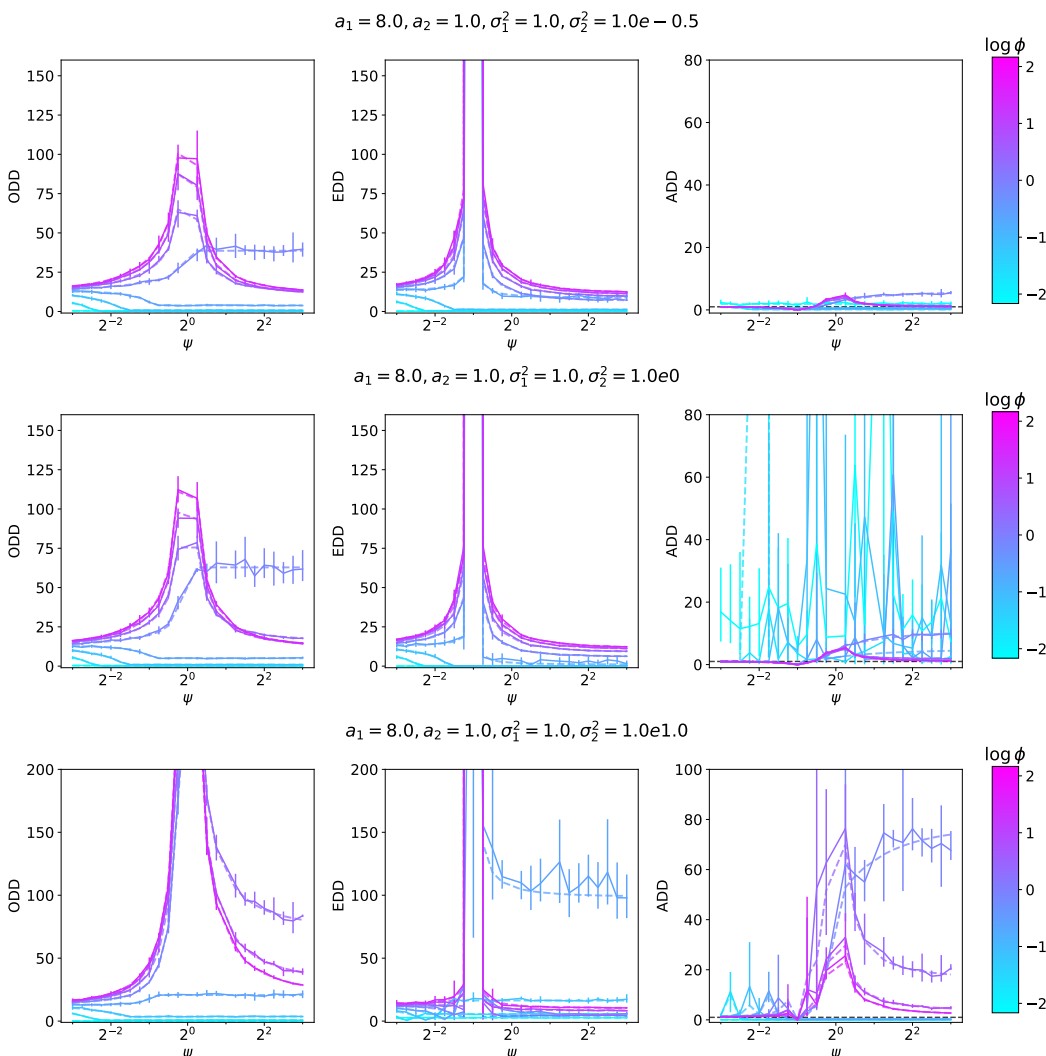

Figure 8: We empirically demonstrate that bias amplification occurs and validate our theory (Theorems 3.1 and 3.2) for $ODD$, $EDD$, and $ADD$ under the setup described in Section 4.1. The solid lines capture empirical values while the corresponding lower-opacity dashed lines represent what our theory predicts. We plot $ODD$ and $EDD$ on the same scale for easy comparison, and include a black dashed line at $ADD = 1$ to contrast bias amplification vs. deamplification. The error bars capture the range of the estimators over 25 random seeds.

## M  POWER-LAW COVARIANCE

To better understand how $\phi$ (rate of features to samples) and the label noise ratio $c$ affect bias amplification, we derive explicit phase transitions in the bias amplification profile of unregularized ridge regression with random projections in terms of these quantities. We consider the setting of power-law covariance, as it is analytically tractable and can be translated to the case of wide neural networks (Caponnetto & de Vito, 2007; Cui et al., 2022; Maloney et al., 2022), where the exponents can be empirically gauged. Let the eigenvalues $\lambda_k^{(s)}$ of $\Sigma_s$ have power-law decay, i.e., $\lambda_k^{(s)} = k^{-\beta_s}$, for all $k$ and some positive constants $\beta_1$ and $\beta_2$. WLOG, we will assume $\beta_1 > \beta_2$. Note that $\beta_s$ controls the effective dimension and ultimately the difficulty of fitting the noiseless part of the signal from group $s$. If $\beta_s$ is large, then all the information is concentrated in a few features, and so the learning problem is easier. We similarly assume that the eigenvalues $\mu_k$ of $\Delta$ have power-law decay $\mu_k = k^{-\alpha}$, for all $k$ and constant $\alpha > 0$. Finally, we consider balanced groups (i.e., $p_1 = p_2 = 1/2$). Under this setup, we have the following corollary.

**Corollary M.1.** *Suppose that in the single model setting, as $\lambda \to 0^+$, $(e_1, e_2, \tau, u_1, u_2, \rho)$ is the unique positive solution to the following fixed-point equations:*

$$1/\tau = 1 + 1/(\gamma\tau), \quad 1/e_s = 1 + \phi \bar{\mathrm{tr}}\, \Sigma_s L^{-1}, \text{ for } s \in \{1, 2\}, \tag{359}$$

$$\rho = 0, \quad u_s = \phi e_s^2 \bar{\mathrm{tr}}\, \Sigma_s D L^{-2}, \text{ for } s \in \{1, 2\}, \tag{360}$$

$$\text{where: } L = p_1 e_1 \Sigma_1 + p_2 e_2 \Sigma_2, \ D = p_1 u_1 \Sigma_1 + p_2 u_2 \Sigma_2 + B. \tag{361}$$

*Furthermore, suppose $\psi_s < 1, \gamma \geq 1$ or $1 \leq \psi_s \leq \gamma$. Under the assumptions of Theorem 3.1 and Assumption B.3, as $\lambda \to 0^+$, we have the following approximate analytical phase transitions in the bias amplification profile of ridge regression with random projections:*

$$\lim_{\substack{d,n_1,n_2\to\infty \\ \phi_{1,2}\to 2\phi}} ADD \to \frac{c}{|c-1|}, \quad \lim_{c\to 0^+} \lim_{\substack{d,n_1,n_2\to\infty \\ \phi_{1,2}\to 2\phi}} ADD \to 0, \tag{362}$$

$$\lim_{c\to\infty} \lim_{\substack{d,n_1,n_2\to\infty \\ \phi_{1,2}\to 2\phi}} ADD \to 1, \quad \lim_{c\to 1} \lim_{\substack{d,n_1,n_2\to\infty \\ \phi_{1,2}\to 2\phi}} ADD \to \infty. \tag{363}$$

We relegate the proof to Appendix N and empirically assess the validity of this result in Figure 9. The phase transitions reveal that bias amplification peaks near $c = 1$, bias deamplification peaks when $c \to 0^+$, and bias is roughly neither amplified or deamplified when $c \to \infty$. Furthermore, the right tail of the $ODD$ profile (which Corollary M.1 predicts to be proportional to $c$) is higher than the left tail (i.e., 0) for larger $c$. However, the left tail of the $EDD$ profile (which Corollary M.1 predicts to be proportional to $|c - 1|$) does not increase steeply as $c \to 0^+$. Interestingly, in the proof of Corollary M.1, we observe that the bias term depends on $\bar{\mathrm{tr}}\, \Delta\Sigma_s$; therefore, the setting $\forall k, \lambda_k^{(s)} \geq 1/\mu_k$ (e.g., common in learning from synthetic data (Dohmatob et al., 2024a)) can prevent the bias term from vanishing or even cause it to explode. This may explain why training models on synthetic data (i.e., data previously generated by the model) may amplify unfairness (Wyllie et al., 2024).

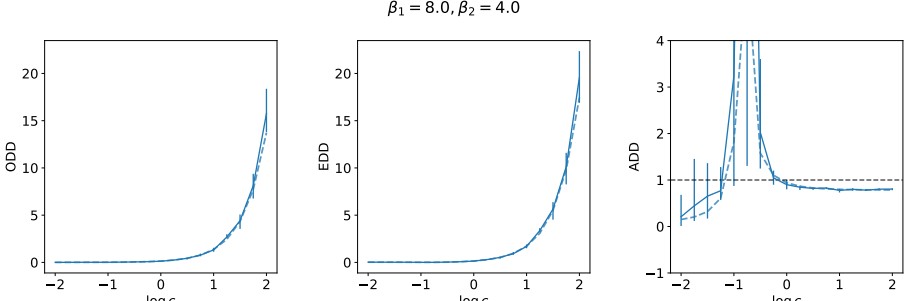

Figure 9: **Our theory predicts that bias amplification is larger for higher noise ratios than lower noise ratios.** We observe that Corollary M.1 generally predicts the $ADD$ profile with respect to the noise ratio $c$. The solid lines capture empirical values while the corresponding lower-opacity dashed lines represent what Theorem 3.1 predicts. We plot $ODD$ and $EDD$ on the same scale for easy comparison, and include a black dashed line at $ADD = 1$ to contrast bias amplification vs. deamplification. The error bars capture the range of the estimators over 25 random seeds. We consider the setup described in Appendix M with $\psi = 0.5$, $\phi = 0.2$, and $\lambda = 1 \times 10^{-6}$.

# N    PROOF OF COROLLARY M.1

*Proof.* We begin by computing the $ODD$ in the limit $\lambda \to 0^+$. We define $u_j^{(s)} = u_j$ for $B = \Sigma_s$. By Assumption B.3, we can re-express the constants in Definition 3.1 in terms of the limiting spectral densities of the covariance matrices:

$$e_1 = \frac{1}{1 + \phi \int_0^\infty \frac{1}{p_1 e_1 + p_2 e_2 r} d\nu(r)}, e_2 = \frac{1}{1 + \phi \int_0^\infty \frac{r}{p_1 e_1 + p_2 e_2 r} d\nu(r)}, \tag{364}$$

$$\tau = \frac{1}{1 + \frac{1}{\gamma \tau}} = 1 - 1/\gamma, \rho = 0, \tag{365}$$

$$u_1^{(1)} = \phi e_1^2 \int_0^\infty \frac{u_1^{(1)} p_1 + u_2^{(1)} p_2 r + 1}{(p_1 e_1 + p_2 e_2 r)^2} d\nu(r), u_2^{(1)} = \phi e_2^2 \int_0^\infty \frac{u_1^{(1)} p_1 r + u_2^{(1)} p_2 r^2 + r}{(p_1 e_1 + p_2 e_2 r)^2} d\nu(r), \tag{366}$$

$$u_1^{(2)} = \phi e_1^2 \int_0^\infty \frac{u_1^{(2)} p_1 + u_2^{(2)} p_2 r + r}{(p_1 e_1 + p_2 e_2 r)^2} d\nu(r), u_2^{(2)} = \phi e_2^2 \int_0^\infty \frac{u_1^{(2)} p_1 r + u_2^{(2)} p_2 r^2 + r^2}{(p_1 e_1 + p_2 e_2 r)^2} d\nu(r). \tag{367}$$

Since $\beta_1 > \beta_2$, $-\beta_2 - (-\beta_1) > 0$. As such, for $d \to \infty$, the ratios $r_k = \lambda_k^{(2)}/\lambda_k^{(1)}$ have the approximate limiting distribution $\nu = \delta_{r=\infty}$, i.e., a Dirac atom at infinity. Thus:

$$e_1 = 1, e_2 = 1 - \frac{\phi}{p_2} = 1 - \phi_2, \tau = 1 - 1/\gamma, \rho = 0, \tag{368}$$

$$u_1^{(1)} = 0, u_2^{(1)} = 0, u_1^{(2)} = 0, u_2^{(2)} = \frac{\phi}{p_2(p_2 - \phi)}. \tag{369}$$

Now, we can re-express the variance terms as:

$$V_1(\widehat{f}) = \phi \sigma_1^2 \int_0^\infty \frac{p_1}{(p_1 + p_2 e_2 r)^2} d\nu(r) + \phi \sigma_2^2 \int_0^\infty \frac{p_2 e_2 r}{(p_1 + p_2 e_2 r)^2} d\nu(r) = 0, \tag{370}$$

$$V_2(\widehat{f}) = \phi \sigma_1^2 \int_0^\infty \frac{p_1 r + p_1 p_2 u_2^{(2)} r}{(p_1 + p_2 e_2 r)^2} d\nu(r) + \phi \sigma_2^2 \int_0^\infty \frac{p_2 e_2 r^2}{(p_1 e_1 + p_2 e_2 r)^2} d\nu(r) = \frac{\sigma_2^2 \phi}{p_2 - \phi}. \tag{371}$$

Likewise, we can re-express the bias terms as:

$$B_1(\widehat{f}) = \int_0^\infty \int_0^\infty \int_0^\infty \frac{a\delta e_2^2 p_2^2 r^2}{(e_1 p_1 + e_2 p_2 r)^2} d\mu(r,a) d\pi(\delta) = \int_0^\infty \int_0^\infty a\delta \, d\mu(a) d\pi(\delta) = 0, \tag{372}$$

$$B_2(\widehat{f}) = 0. \tag{373}$$

In this calculation, we observe that the adversarial setting $\forall k, \lambda_k^{(1)} \geq 1/\mu_k$ can prevent the bias term from vanishing. Putting these pieces together and recalling that $p_2 = 1/2$:

$$ODD \to \left| V_1(\widehat{f}) - V_2(\widehat{f}) \right| = \frac{2\phi \sigma_1^2}{1 - 2\phi} c. \tag{374}$$

We now compute the $EDD$. We can once again re-express the constants in Definition H.1 in terms of the limiting spectral densities of the covariance matrices:

$$e_s = \frac{1}{1 + \phi_s/e_s} = 1 - \phi_s, \tau_s = 1 - 1/\gamma. \tag{375}$$

By Corollary J.1, because $\psi_s < 1, \gamma \geq 1$ or $1 \leq \psi_s \leq \gamma$, $B_s(\widehat{f}_s) = 0$ and $V_s(\widehat{f}_s) = \frac{\sigma_s^2 \phi_s}{1 - \phi_s}$.

Therefore, because $\phi = p_s \phi_s$:

$$EDD \to \left| V_1(\widehat{f}_1) - V_2(\widehat{f}_2) \right| = \frac{2\phi}{1 - 2\phi} \left| \sigma_1^2 - \sigma_2^2 \right| = \frac{2\phi \sigma_1^2}{1 - 2\phi} |c - 1|, \tag{376}$$

$$ADD \to \frac{c}{|c - 1|}. \tag{377}$$

$\square$

# O    BIAS AMPLIFICATION DURING TRAINING

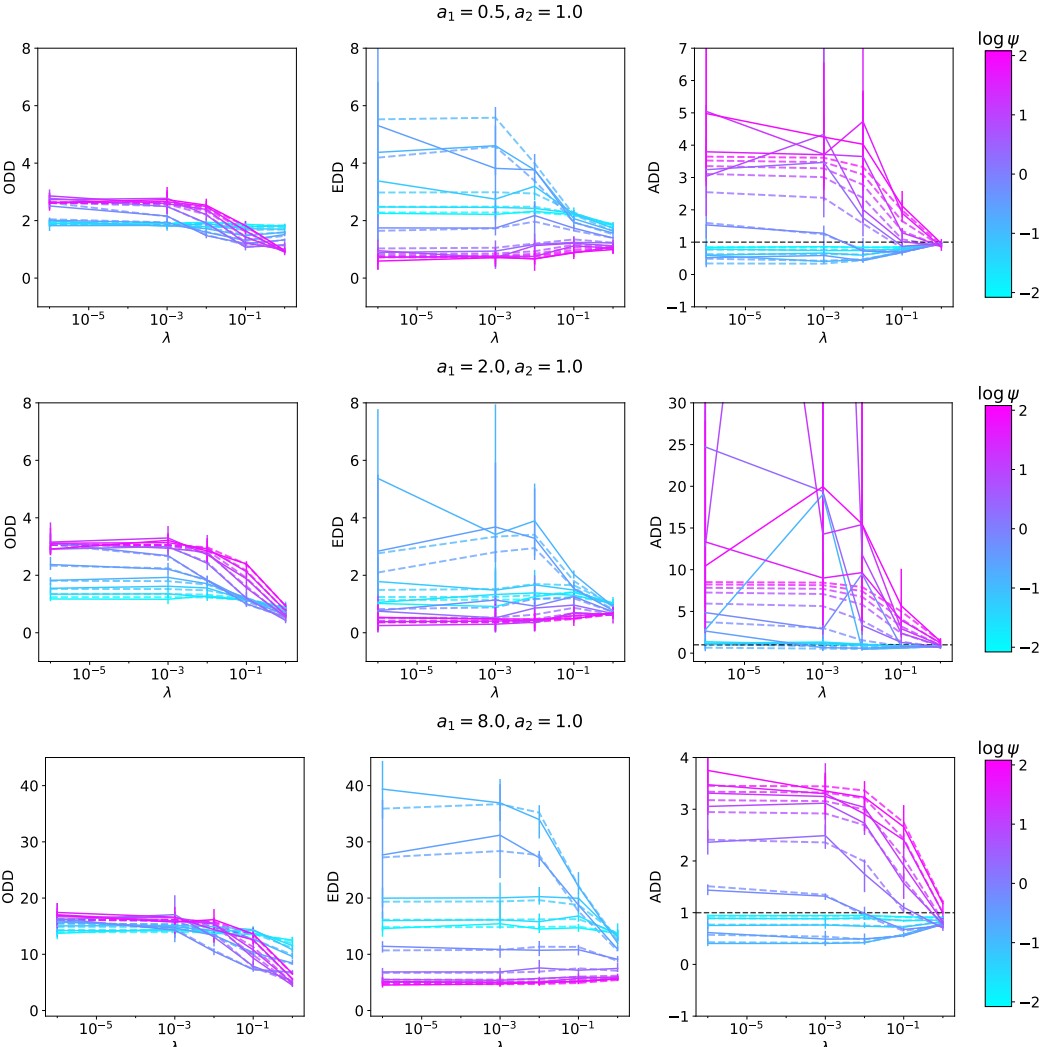

Figure 10: **Our theory reveals that there may be an optimal regularization penalty to deamplify bias.** We empirically demonstrate that bias amplification can be heavily affected by $\lambda$ and validate our theory (Theorems 3.1 and 3.2) for $ODD$, $EDD$, and $ADD$ under the setup described in Section 4.2. The solid lines capture empirical values while the corresponding lower-opacity dashed lines represent what our theory predicts. We include a black dashed line at $ADD = 1$ to contrast bias amplification vs. deamplification. The error bars capture the range of the estimators over 25 random seeds.

## P    COLORED MNIST PLOTS

We further assess the applicability of our conclusions about the effects of label noise (Figures 3, 11) and model size (Figure 12) on bias amplification for Colored MNIST. Please see Section 4.2 for a discussion of Figure 3. We observe in Figure 11 that as we increase the label noise ratio, the $EDD$ generally increases, while the $ODD$ remains relatively low, which is suggested by our theoretical reasoning in Section 4.2. Furthermore, in Figure 12, as the hidden dimension $m$ of the penultimate layer of the CNN increases, the $ODD$ appears to decrease and plateau, which is predicted by our theoretical results (see Section 4.1) in the Colored MNIST regime where $\phi < 1$. However, the $EDD$ does not appear to decrease; while this is plausibly predicted by our theory, it requires going beyond our simplistic assumption that $\Sigma_1$ roughly coincides with $\Sigma_2$ and studying the interplay between $\phi_s, \psi_s, \Sigma_s$ for each group $s$ (as suggested by Appendix L).

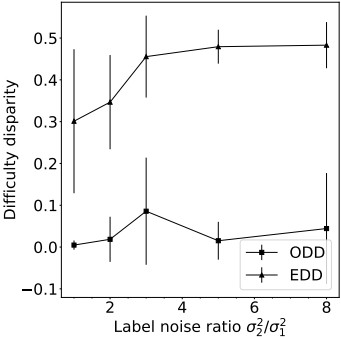

Figure 11: **Our theory predicts that more disparate label noise between groups deamplifies bias on Colored MNIST.** We plot the $ODD$ and $EDD$ of a CNN for different label noise ratios $c = \sigma_2^2/\sigma_1^2$ for Colored MNIST. As $c$ increases, the $EDD$ generally increases while the $ODD$ remains relatively low, which is predicted by our theory (see reasoning in Section 4.2). In our experiments, $\sigma_1^2 = 0.05$ stays fixed while $\sigma_2^2$ varies. For each value of $c$, the model is evaluated after $t = 80$ training steps and has a penultimate layer with dimension $m = 500$. The error bars capture the standard deviation computed over 10 random seeds.

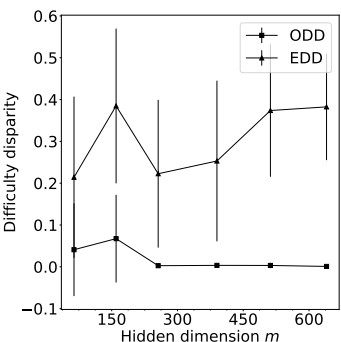

Figure 12: **Our theory predicts that a larger model size reduces bias on Colored MNIST in the single model setting.** We plot the $ODD$ and $EDD$ of a CNN for different model sizes $m$ (where $m$ is the dimension of the penultimate CNN layer) for Colored MNIST. As $m$ increases, the $ODD$ appears to decrease and plateau, which is in line with what our theory predicts in the regime where $\phi < 1$ (see analysis in Section 4.1). The $EDD$ does not tend towards 0. In our experiments, $\sigma_1^2 = \sigma_2^2 = 0.05$. For each value of $m$, the model is evaluated after $t = 80$ training steps. The error bars capture the standard deviation computed over 10 random seeds.

# Q    MINORITY-GROUP BIAS PLOTS

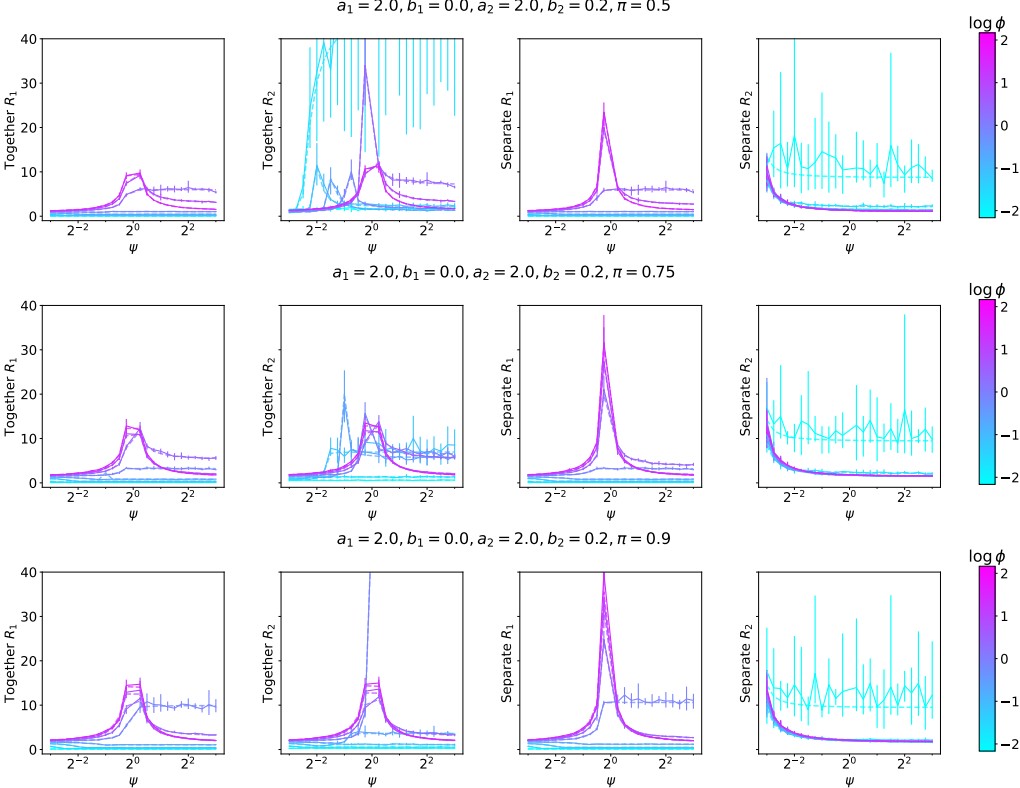

Figure 13: We empirically demonstrate that minority-group bias is affected by extraneous features. We validate our theory (Theorems 3.1 and 3.2) for together $R_1$, $R_2$ (i.e., single model learned for both groups) and separate $R_1$, $R_2$ (i.e., separate model learned per group) under the setup described in Section 4.2. The solid lines capture empirical values while the corresponding lower-opacity dashed lines represent what our theory predicts. We include a black dashed line at $ADD = 1$ to contrast bias amplification vs. deamplification. All y-axes are on the same scale for easy comparison. The error bars capture the range of the estimators over 25 random seeds.

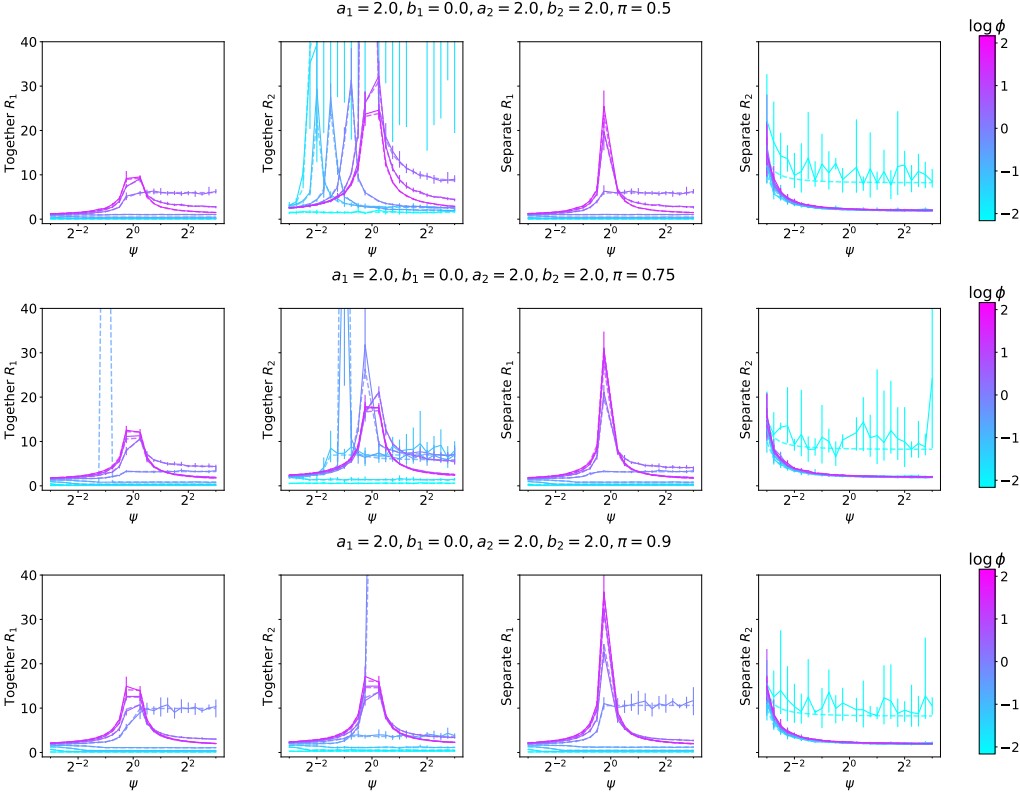

Figure 14: We empirically demonstrate that minority-group bias is affected by extraneous features. We validate our theory (Theorems 3.1 and 3.2) for together $R_1, R_2$ (i.e., single model learned for both groups) and separate $R_1, R_2$ (i.e., separate model learned per group) under the setup described in Section 4.2. The solid lines capture empirical values while the corresponding lower-opacity dashed lines represent what our theory predicts. We include a black dashed line at $ADD = 1$ to contrast bias amplification vs. deamplification. All y-axes are on the same scale for easy comparison. The error bars capture the range of the estimators over 25 random seeds.

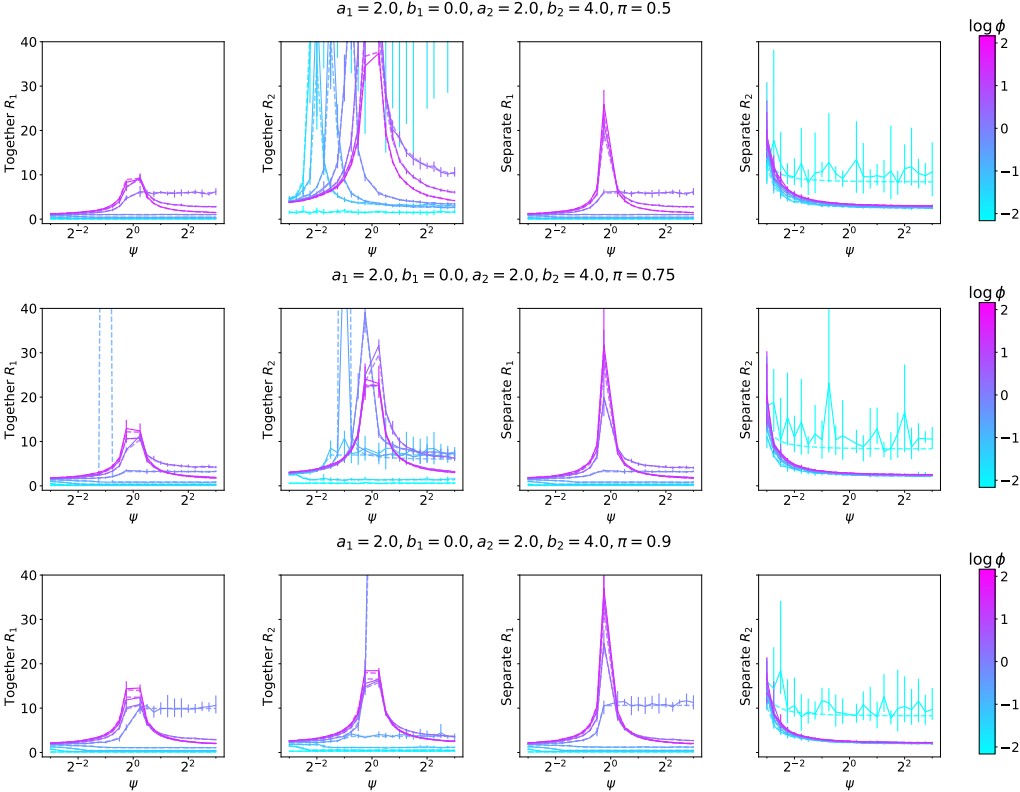

Figure 15: We empirically demonstrate that minority-group bias is affected by extraneous features. We validate our theory (Theorems 3.1 and 3.2) for together $R_1$, $R_2$ (i.e., single model learned for both groups) and separate $R_1$, $R_2$ (i.e., separate model learned per group) under the setup described in Section 4.2. The solid lines capture empirical values while the corresponding lower-opacity dashed lines represent what our theory predicts. We include a black dashed line at $ADD = 1$ to contrast bias amplification vs. deamplification. All y-axes are on the same scale for easy comparison. The error bars capture the range of the estimators over 25 random seeds.

# R    ACTIONABLE INSIGHTS FROM THEORY

**Searching for optimal hyperparameters.**    In practice, an optimal regularization penalty $\lambda$ or training time $t$ can be selected by searching for values that strike a desired balance between overall validation error (that is not too high) and bias amplification (that is not too high). As we would estimate the test error using the empirical validation error, we can estimate bias amplification using the validation set. Moreover, we would need to train: (1) the main model on a mixture of data from groups, and (2) auxiliary separate models on the data for each group.

However, it may be expensive to train auxiliary models for each candidate value of $\lambda$ and $t$. The search space can be reduced by using insights from our theory. For instance, with overparamaterization, as $\lambda$ decreases (or $t$ increases), bias amplification increases and plateaus, and with underparameterization, as $\lambda$ decreases (or $t$ increases), bias deamplification increases and plateaus. When the curves are monotone with respect to $\lambda$, the optimal $\lambda$ is either at the left tail of the curve (e.g., $\lambda = 0$) or the right tail (i.e., the largest $\lambda$ among the reasonable options). In contrast, Figure 10 shows that when $\psi$ is close to the interpolation threshold of 1, bias amplification is often not monotone with respect to $\lambda$.

**Informing evaluation and mitigation strategies.**    Our theory offers avenues to assess whether a ML model trained on certain real-world data is prone to bias amplification and mitigate this amplification, even though we may lack direct access to population parameters like $\Sigma$. We can estimate such parameters using samples (e.g., $\widehat{\Sigma} = X^\top X$). However, even if we are unable to robustly estimate these parameters, our theory still provides valuable insights. For example, we observe that the ratios of parameters for the groups is often what matters, e.g., label noise ratio $\sigma_2^2/\sigma_1^2$ (see Section 4.1), ratio of covariance eigenvalues (see Appendix M). Thus, practitioners can use our theory to get intuition about when *disparities* in the variability of labels and features across groups can amplify bias.

Moreover, our findings warn against the conventional wisdom that increased model overparameterization or data balancing can alleviate bias issues. In addition, our theory informs criteria for feature selection (e.g., discarding features with disparate variance across groups) and warns ML practitioners about the interplay between high vs. low feature-to-sample regimes and overparameterization in inducing bias amplification. Nevertheless, additional work is required to make rigorous connections between our theoretical findings and better strategies for evaluating and mitigating the bias of models.

