# OpenReview forum: "An Effective Theory of Bias Amplification"
_ICLR.cc/2025/Conference — ICLR 2025 Poster_

### Official Review · Reviewer_KjTp · 2024-10-27

**Soundness:** 3
**Presentation:** 3
**Contribution:** 3
**Rating:** 8
**Confidence:** 4

**Summary:**

The paper develops a rigorous statistical framework with theory to explain and analyze model bias, particularly focusing on one of the most popular statistical methods, ridge regression, both with and without random projections (which model neural networks in a simplified regime). Its key contributions include:

Unified Theory of Bias: The paper provides a unifying theory to understand how model design choices and data distribution properties lead to bias, including phenomena like bias amplification and minority-group error. This framework explains how bias can emerge and be amplified in different feature and parameter regimes.

Bias Amplification: It analyzes how training a single model on combined data from different groups can amplify bias more than training separate models on each group’s data, even without group imbalance or spurious correlations. The theory also suggests that early stopping and tuning regularization can reduce this amplification.

Minority Group Error: The paper investigates how overparameterization and extraneous features negatively affect test performance for minority groups. It explicitly connects model size to error amplification for these groups.
Empirical Validation: The theoretical predictions are extensively validated using synthetic and semi-synthetic datasets, and the results align with empirical findings reported in prior literature.

Overall, the paper offers novel statistical theory and new insights into model bias and proposes avenues for mitigating it through informed model design and training choices.

**Strengths:**

**Originality**: This paper has originality. There are many works on model bias, but most of them are empirical work. A unified statistical theory of model bias is underexplored. Compared to previous work on model bias theory, this paper focuses on disparities in test error across majority and minority groups with different data distributions when models are trained on data from both groups. This paper does use existing tools based on random matrix theory from previous work, but the theory is not a special case of the existing ones. Therefore, I think the paper has its originality.


**Quality**: This paper has good quality. The analysis using operator-valued free probability theory (OVFPT) and high-dimensional scaling limits look technically sound to me. Fixed-point equations and a detailed exploration of phase transitions (e.g., bias amplification at specific proportions of features to samples) are mathematically rigorous. Further, besides the theoretical analysis, this empirical section demonstrates how ridge regression models with random projections can amplify bias, even in balanced data without spurious correlations. This is an interesting finding. It offers a clear, empirical validation of theoretical predictions, demonstrating the impact of model size, label noise, and training dynamics on bias amplification.

**Clarity**: The structure is clear. From preliminaries in section 2 to a detailed model analysis in section 3, and empirical study in section 4, they all manifest a well-organized and thoughtful presentation.

**Significance**: This paper studies an important problem and provides a good contribution to the field. To be specific, this article tackles the critical issue of bias in machine learning, which can lead to unequal performance across demographic groups. It presents a new theoretical framework using operator-valued free probability theory to explain and predict bias amplification, unifying previous findings and offering a clearer understanding of bias mechanisms. The theory is validated in diverse empirical settings, showing practical relevance. Overall, this work makes a significant contribution by providing a structured approach to understanding and mitigating bias, with implications for fairer machine learning practices and future research directions.

**Weaknesses:**

There are no significant weaknesses in this paper. However, the following minor weaknesses can be improved.

1. The connection between the theoretical analysis in this paper and the prior work in OVFPT is not discussed clearly enough. This paper mentions that the theory cannot be derived as a special case in Adlame et al., Tripuraneni et al., Lee et al., but does not provide a more involved explanation on why this is the case. It would thus be great if a paragraph discussing this point can be added.

2. Section 4.2 nicely addresses how regularization and training time influence bias amplification, but it lacks a bit of practical guidance on applying these insights in real scenario. After discussing the effects of regularization and training dynamics on bias amplification, the authors could add a bit more remarks or comments. This would be an ideal place to provide concrete guidance on setting regularization parameters or selecting optimal training durations for different dataset sizes and feature-sample ratios. For instance, they could suggest regularization ranges or early stopping criteria when $\psi$ is close to or above 1.

**Questions:**

Please see the **Weaknesses** section.

---

> ### Author Response · Authors · 2024-11-18
> **Author Response**
>
> Thank you for your thoughtful feedback and questions!
>
> **To address the weaknesses/questions that you mentioned:**
>
> **(W1)** We will add the following context to the revised version of our manuscript:
>
> Our theoretical results cannot be derived as a special case of [1] because this work focuses on training and testing a random features model on data from the same, single Gaussian distribution. Furthermore, [2] and [3] focus on training a random features model on data from a single Gaussian distribution and testing the model on a different, single Gaussian. In contrast, we study the random features model in the setting of training on a mixture of Gaussian distributions and testing on each component. Because any Gaussian distribution ${\cal N}_1$ can be expressed as a mixture of Gaussians $p_1 {\cal N}_1 + p_2 {\cal N}_2$ where $p_1 = 1, p_2 = 0$, but a mixture is more expressive than a single Gaussian, our theoretical results cannot be a special case of [1, 2, 3].
>
> **(W2)** This is a great suggestion. We will add the following content to Section 4.2:
>
> When $\psi \approx 1$ (i.e., the number of parameters and samples are about matched), bias amplification is not monotonic with respect to regularization (or training time); this motivates finding an optimal $\lambda$ or training time to reduce bias amplification. In practice, an optimal $\lambda$ or training time can be selected by searching for values that strike a desired balance between overall validation error and bias amplification. The search space can be further reduced by using insights from our theory such as with overparamaterization, as regularization decreases (or training time increases), bias amplification increases and plateaus, and with underparameterization, as regularization decreases (or training time increases), bias amplification decreases and plateaus. It is further important for ML practitioners the consider the interplay between high vs. low feature-to-sample regimes and overparameterization in inducing bias amplification vs. deamplification, respectively, when selecting optimal hyperparameters (see Figure 2).
>
> **References**
>
> [1] Adlam, B., & Pennington, J. (2020, November). The neural tangent kernel in high dimensions: Triple descent and a multi-scale theory of generalization. In International Conference on Machine Learning (pp. 74-84). PMLR.
>
> [2] Lee, D., Moniri, B., Huang, X., Dobriban, E., & Hassani, H. (2023, July). Demystifying disagreement-on-the-line in high dimensions. In International Conference on Machine Learning (pp. 19053-19093). PMLR.
>
> [3] Tripuraneni, N., Adlam, B., & Pennington, J. (2021). Covariate shift in high-dimensional random feature regression. arXiv preprint arXiv:2111.08234.

---

> ### Author Response · Authors · 2024-11-21
>
> Thanks again for your review! Please feel free to let us know if there are any remaining comments not addressed. We appreciate any feedback, and we are happy to answer any further questions.

---

> > ### Comment · Reviewer_KjTp · 2024-11-23
> > **Response**
> >
> > I thank the authors for addressing my questions. Please ensure to add the additional discussion to the revision.
> > Overall, I think this is a good paper that contributes to the statistical theory for understanding model bias, a crucial step in the study of model generalization performance.
> >
> > I maintain my score at 8.
> >
> > Best regards,
> > Reviewer

---

> ### Author Response · Authors · 2024-11-24
>
> We have included the additional discussion in the revision (in green). We thank the reviewer again for their helpful feedback and positive reception of this work!

---

### Official Review · Reviewer_m7Lv · 2024-10-30

**Soundness:** 3
**Presentation:** 3
**Contribution:** 2
**Rating:** 6
**Confidence:** 4

**Summary:**

The authors introduce a simple modeling framework for understanding the phenomenon of bias amplification, i.e. the observation of abnormal disparate model performance on communities unequally represented in the dataset. The setup encompasses a regression with a linear model (both a single-layer network and a linear two-layer network are considered) on a dataset with multiple modalities and different associated rules. This framework allows them to capture some key aspects of bias amplification and to make qualitative predictions on the possible regimes where over-parametrization can help or hinder the model's unbiasedness. The analytical results are derived with the tools of "operator-valued free probability theory", and corroborated with some empirical observations on standard benchmarks.

**Strengths:**

I believe the topic analyzed in this paper is very relevant, and theoretical contributions in this setting are crucial to understanding how to mitigate the pressing issue of bias amplification. In this context, it is interesting to understand and discuss whether the usual recipe of overparametrization could be more harmful than expected.
The authors' analysis seems technically sound, although the methods required the introduction of some modeling assumptions that might be simplistic in some cases.

**Weaknesses:**

My main concern about this work is its extreme similarity, in goals, modeling setup, and some of the conclusions, with previous work ("Bias-inducing geometries: an exactly solvable data model with fairness implications" https://arxiv.org/abs/2205.15935), which is ignored in the present paper. Notice that this same reference is for example cited in ref. (Bell & Sagun, 2023) within this paper. While I can recognize some differences, reasonably motivating the present extension, there should be a clear discussion of the connections and differences.

Moreover, the authors do not discuss at length the limitations of some of their assumptions. For example, using a linear network as a proxy of a deep network, or using regularization intensity to approximate early stopping and to understand the training dynamics (without looking at the training dynamics). Although the assumptions might be "standard", the limitations should be discussed also in the present work.

Finally, some of the paragraphs on the results in figures are hardly readable because constantly refer to different values of the model parameters, often without providing a clear explanation of the "meaning" of the different regimes.

**Questions:**

MAJOR
1) As mentioned above, the authors should acknowledge and discuss the connection of their work to arxiv:2205.15935. Ignoring such similarity to a previous work is unacceptable for an ICLR paper.
2) It seems plausible that much of the observed bias phenomenology in this paper is a direct effect of the double-descent / interpolation peak phenomenology, which is known to shift with the degree of over-parametrization of the network and to cause a critical sensitivity to input noise (hindering the generalization). Could the authors discuss this connection more in detail, and clarify if a different cause is at play? Is the linear activation assumption a confounding effect regarding this?
3) The imputation of the effect of early stopping in the training dynamics from the effect of a properly scaled regularization (1/t) might lead to simplistic conclusions. Can the authors discuss, for example, the connection to this previous work arXiv:2405.18296? In general, I believe mentioning optimal training time, and training dynamics both in the abstract and in the paragraph titles without specifying that this extrapolation hinges on a very simplistic approximation is misleading.
4) Can the authors mention how, for example, their results can be used to "inform strategies to evaluate and mitigate unfairness in machine learning"?

MINORS
1) Not clear if in Fig. 1 the setting is with balanced groups, or if there is another figure for this.
2) The explanations e.g. in lines 311-317, 372-375, 400-403, 443-453, are hardly understandable if you are not extremely familiar with the model parametrization. Maybe a more effective way of presenting the results would be to state in words what setup is being considered, before looking at the parameter values.
3) I don't think it is fair practice to hide the error bars of Figure 3 in the supplementary. Moreover, the associated theoretical prediction seems very qualitative.


AFTER REVISION
Given the implemented changes, I will raise my score to a 6.

---

> ### Author Response · Authors · 2024-11-18
> **Author Response (Part 1)**
>
> Thank you for your detailed review and questions!
>
> **We first address each weakness mentioned below:**
>
> **(W1)** Thank you for bringing our attention to [1] and [2]. We were not previously aware of these papers; they were not ignored. We will include the following paragraphs in our revised manuscript, which clearly discusses the connections and differences of our work with [1] and [2]:
>
> **The main distinction between our work and [1] and [2] is we provide a precise theoretical characterization of how models amplify bias in different parameterization regimes.**
>
> In terms of similarities, [1] and [2] also precisely analyze the bias of models trained on a mixture of data from multiple groups in a high-dimensional setting. Like [1] and [2], we study linear models that are trained with regularization, and measure bias as the difference in test performance of a model between two groups. We further consider some similar factors that give rise to bias amplification, e.g., group imbalance, group variance, inter-group similarity, and dataset size. In addition, we share some theoretical conclusions, e.g., bias can occur even when the groups have the same ground-truth weights (Section 5) and are balanced (Section 4.1). We also discuss positive transfer in our experiment with CMNIST (Section 4.2).
>
> However, our treatment of bias amplification adds nuance to the discussion of positive transfer in [1], which claims that the $EDD$ generally tends to be higher than the $ODD$. Instead, we show that the bias amplification $ADD = ODD / EDD$ can vary greatly (going below 1 and above 1) as a function of the rate of parameters to samples $\psi$, even for a fixed $\phi$, or $\alpha$ as is used in [1] (see Figure 2 in our manuscript).
>
> This is because [1] and [2] only consider the setting where the number of samples $n$ and features $d$ proportionally scale to infinity, while we consider the setting $n, d \to \infty$ *and* the number of parameters $m \to \infty$. This enables us to expose new, richer insights into the impact of (over/under-)parameterization on bias amplification, such as in Figure 1, lines 368-397, and lines 510-520. For example, we find that “a larger number of features than samples can amplify bias under overparameterization, there may be an optimal regularization penalty or training time to avoid bias amplification, and there can be fundamental differences in test error between groups that do not vanish with increased parameterization” (lines 78-82), among various other findings in Sections 4 and 5.
>
> Additionally, [1] employs the replica method, which is non-rigorous, while we use operator-valued free probability theory (explained in Appendix A), which is entirely rigorous. Furthermore, while [1] discusses the paradigm of training separate models for each group, it theoretically focuses on a single model trained for both groups. In contrast, in our work, we theoretically treat both these paradigms (i.e., to isolate the contribution of the model itself to bias) and validate our theory extensively. Moreover, [1] and [2] study the application of linear classification to Gaussian data and ground-truth weights with isotropic covariance, while we focus on the application of regression with random projections (a simplified model of NNs) to Gaussian data and weights with more general covariance structure. This allows us to analyze additional factors of bias, such as group covariance structure and label noise.

---

> > ### Author Response · Authors · 2024-11-18
> > **Author Response (Part 2)**
> >
> > **(W2)** We will include the following elaboration on the limitations of our assumptions in the revised version of our manuscript:
> >
> > Ridge regression with random features has been posited as a reasonable approximation for NNs in the random features regime [3, 4]. For example, it has been argued that as the number of parameters $m \to \infty$ (as in our high-dimensional setting), gradient descent effectively learns a linear predictor over $m$ random features [3]. Furthermore, [4, 5, inter alia] are able to reproduce interesting phenomena like double descent using the random features model. Nevertheless, [3] has shown that in practice, “random features cannot be used to learn even a single ReLU neuron with standard Gaussian inputs,” which suggests that some mechanisms of bias amplification could be different in nonlinear networks. However, still using free probability theory and the Gaussian equivalence theorem [6], we can easily extend our theory to study bias amplification in NNs with nonlinearities in the NTK regime, as in [7], albeit with more complex equations (see lines 537-539 of our manuscript).
> >
> > Furthermore, the theoretical analysis of [8] reveals that the calibration $\lambda = 1 / t$ yields a ratio of gradient flow to ridge risk that is at most 1.6862, with no assumptions on the features $X$. Hence, while a good estimate, the calibration $\lambda = 1 / t$ may not in general yield a theoretically tight picture of how bias evolves with $t$. The use of discrete GD in practice rather than continuous-time gradient flows might yield further discrepancies. However, in the controlled settings considered by [8] and our work, this ratio empirically appears to be quite close to 1, and thus sufficient for extrapolating our results.
> >
> > **(W3)** Thanks for your feedback on readability. We tried to continually reiterate the definitions of important quantities like $\phi$ and $\psi$ throughout the paper (e.g., line 480). However, we will revise each of our results sections to more clearly convey what each regime means in plain terms (e.g., $\psi > 1$ is the overparameterized regime). We strive to make our manuscript as readable as possible.

---

> > > ### Author Response · Authors · 2024-11-18
> > > **Author Response (Part 3)**
> > >
> > > **To answer your questions:**
> > >
> > > **MAJOR**
> > >
> > > **(Q1)** Please see our response to (W1) above.
> > >
> > > **(Q2)** Indeed, some of the peaks and valleys in the bias amplification profile of models can be attributed to double descent / interpolation poles. However, double descent in high dimensions has been studied in the settings where data are drawn from single Gaussian distributions, not a mixture of Gaussians. This corresponds to the $EDD$ setting where a separate model is learned for each group. As expected, as we state in lines 312-313 and lines 372-373, we observe a double-descent peak in the $EDD$ at $\psi = 0.5$ (i.e., $\psi_1 = \psi_2 = 1$). We will make this connection to double descent clearer. Furthermore, our work extends the theoretical treatment of the double descent phenomenon to the setting of training a model on a mixture of Gaussians. By doing so, we find, e.g., a series of interpolation thresholds as $\psi$ increases, rather than just a single pole (lines 484-509).
> > >
> > > However, our theory of bias amplification can by no means be reduced exclusively to double descent. For example, we note other interpolation poles in lines 309-317 (e.g., at $\phi = \psi$). In addition, much of Sections 4, 5, and K are devoted to studying the tails or limiting behavior of bias amplification with respect to $\psi$ and $\phi$. For example, we study how increased parameterization affects minority-group bias (lines 368-397, 510-519) and derive the limiting values of $ADD$ when the covariance matrices have power-law decay (Appendix K).
> > >
> > > The linear activation assumption likely does not have a confounding effect, as interpolation poles have also been observed in nonlinear networks in the NTK regime [7].
> > >
> > > **(Q3)** Please see our response to (W2) above. We will make it clearer that our approximation $\lambda = 1/t$ is simplistic and may only be empirically tight in controlled settings.
> > >
> > > In relation to [2], [2] analytically characterizes the evolution of bias by exactly solving a set of ODEs in their setting. In doing so, [2] offers a rich characterization of how bias evolves during training, e.g., by identifying three phases and the crossing phenomenon. However, [2] does not consider the effect of (over/under-)parameterization on bias evolution. In contrast, our analysis, despite relying on the simplistic calibration $\lambda = 1/t$, reveals divergent behavior of how bias evolves depending on whether the model is under or over-parameterized (lines 411-424, Appendix J).
> > >
> > >  **(Q4)** Additional work is required to make rigorous connections between our theoretical findings and better strategies for evaluating/mitigating bias in models. However, our findings can provide intuition for better evaluations and mitigations. For example, our findings warn against the conventional wisdom that increased model overparameterization or data balancing can solve bias issues. In addition, our theory informs criteria for feature selection (e.g., discarding features with disparate variance across groups) and warns ML practitioners about the interplay between high vs. low feature-to-sample regimes and overparameterization in inducing bias amplification vs. deamplification, respectively (lines 369-397).
> > >
> > > Furthermore, in practice, an optimal $\lambda$ or training time can be selected by searching for values that strike a desired balance between overall validation error (that is not too high) and bias amplification (that is not too high). The search space can be reduced by using insights from our theory such as under overparamaterization, “as regularization decreases (or training time increases), bias amplification increases and plateaus,” and with underparameterization, “as regularization decreases (or training time increases), bias deamplification increases and plateaus” (lines 416-419).

---

> ### Author Response · Authors · 2024-11-18
> **Author Response (Part 4)**
>
> **MINOR**
>
> **(Q1)** Yes, Figure 1 is with balanced groups (as stated in line 341).
>
> **(Q2)** We will revise each of our results sections to regularly remind the reader of the setup and more clearly convey what each regime means in plain terms (e.g., $\psi > 1$ is the overparameterized regime). We described the setup upfront at the start of each subsection (e.g., lines 346-353, lines 408-410, lines 465-483).
>
> **(Q3)** We do not have any intention of “hiding” the error bars in Figure 3. We made an editorial decision to relegate all the versions of the figures with error bars to the appendix to make the figures more readable. Moreover, the version of Figure 3 with error bars does not convey a different message; our finding still holds, and we are happy to place the error-bar version in the main body. The theoretical predictions associated with Figure 3 are relatively qualitative because CMNIST is a classification problem while our theory is only precise for regression. However, under the assumptions and analogies that we make in lines 437-457, our derivations of the bias and variance terms $B$ and $V$ are correct, with $\approx$ simply capturing the slight discrepancy in $\Sigma_1$ and $\Sigma_2$ due to the colors of the digits.
>
> We hope that the explanations and clarifications provided above address the reviewer’s questions and comments.
>
> **References**
>
> [1] Mannelli, S. S., Gerace, F., Rostamzadeh, N., & Saglietti, L. (2022). Unfair geometries: exactly solvable data model with fairness implications. arXiv preprint arXiv:2205.15935.
>
> [2] Jain, A., Nobahari, R., Baratin, A., & Mannelli, S. S. (2024). Bias in Motion: Theoretical Insights into the Dynamics of Bias in SGD Training. arXiv preprint arXiv:2405.18296.
>
> [3] Yehudai, G., & Shamir, O. (2019). On the power and limitations of random features for understanding neural networks. Advances in neural information processing systems, 32.
>
> [4] Adlam, B., & Pennington, J. (2020). Understanding double descent requires a fine-grained bias-variance decomposition. Advances in neural information processing systems, 33, 11022-11032.
>
> [5] Bach, F. (2024). High-dimensional analysis of double descent for linear regression with random projections. SIAM Journal on Mathematics of Data Science, 6(1), 26-50.
>
> [6] Goldt, S., Loureiro, B., Reeves, G., Krzakala, F., Mézard, M., & Zdeborová, L. (2022, April). The gaussian equivalence of generative models for learning with shallow neural networks. In Mathematical and Scientific Machine Learning (pp. 426-471). PMLR.
>
> [7] Adlam, B., & Pennington, J. (2020, November). The neural tangent kernel in high dimensions: Triple descent and a multi-scale theory of generalization. In International Conference on Machine Learning (pp. 74-84). PMLR.
>
> [8] Ali, A., Kolter, J. Z., & Tibshirani, R. J. (2019, April). A continuous-time view of early stopping for least squares regression. In The 22nd international conference on artificial intelligence and statistics (pp. 1370-1378). PMLR.

---

> ### Author Response · Authors · 2024-11-21
>
> Thanks again for your review! Please feel free to let us know if there are any remaining comments not addressed. We appreciate any feedback, and we are happy to answer any further questions.

---

> > ### Comment · Reviewer_m7Lv · 2024-11-22
> > **Response**
> >
> > I would like to thank the authors for carefully answering my concerns and promising to change their manuscript accordingly.
> > If the promised changes are implemented in the revised version, in particular the additions mentioned in the Author Response, points W1, W2, Q2, and Q3, I will definitely raise my score from 3 to 6.

---

> > > ### Author Response · Authors · 2024-11-22
> > >
> > > Thanks again for your thoughtful and constructive review, and for recognizing our careful consideration of your concerns! We have revised the manuscript to implement the important additions mentioned in W1, W2, Q2, and Q3 in the main body of our paper; these revisions have been highlighted in magenta. We have additionally placed the version of Figure 3 with error bars in the main body. We thank the reviewer in advance for increasing their score!

---

### Official Review · Reviewer_HU3q · 2024-11-03

**Soundness:** 3
**Presentation:** 2
**Contribution:** 2
**Rating:** 6
**Confidence:** 3

**Summary:**

Machine learning models can capture and amplify biases present in data. This paper develops an analytical theory within the context of ridge regression to better understand and address these biases. The theory provides insights into phenomena like bias amplification and minority-group bias, revealing that optimal regularization penalties or training times may help mitigate bias. Their theoretical predictions are supported by empirical validation on diverse synthetic and semi-synthetic datasets.

**Strengths:**

This paper examines bias amplification in ridge regression with random projections. Theoretical findings not only predict known bias amplification effects but also reveal new amplification phenomena under isotropic covariance. These findings are further validated through empirical analysis on a complex semi-synthetic image dataset.

**Weaknesses:**

- It appears that the analysis of bias amplification heavily relies on an implicit assumption about the conditional dependence heterogeneity (e.g., $w_2\neq w_1$) between the two groups. What would happen if this assumption were relaxed or removed?

- The manuscript does not sufficiently justify the definition of “bias amplification.” I think more theoretical backing is needed for this definition.

- More theoretical justification is needed to demonstrate that ridge regression with random projections serves as an effective approximation for neural networks (NN). Additionally, in Line 172, the one-hidden-layer NN model seems lacks a nonlinear activation function?

**Questions:**

- I have mixed feelings about the analysis of bias amplification. On one hand, the results are somewhat intuitive if we assume two groups, one with $y = x$ and the other with $y = -x$. On the other hand, the theoretical results are based on isotropic covariance, which is a simplified scenario. What would happen if the covariance structure were non-isotropic, for instance, if the covariance between $X_i$ and $X_j$ fixes or exponentially decays?

- In real-world datasets, we often lack information about the detailed properties of each group such as $\Sigma_1$. How can we assess whether a machine learning model trained on such data is prone to bias amplification?

- Even if we identify bias amplification and attribute it to heterogeneous conditional dependencies, how can we design a practical criterion to select the optimal $\lambda$ or runtime? Additionally, could the heterogeneity itself be leveraged to develop a better machine learning model?

---

> ### Author Response · Authors · 2024-11-18
> **Author Response (Part 1)**
>
> Thank you for your detailed review and questions!
>
> **We first address each weakness mentioned below:**
>
> **(W1)** **Our analysis does not rely on conditional dependence heterogeneity.** To consider the setting where $w^*_1 = w^*_2$, we can simply set $\Delta = 0$ in our theoretical findings. In fact, we empirically verify that our theory still holds and predicts bias amplification occurs even when $w_1 = w_2$ in Section 5 (lines 477-479, Figure 4). In essence, the structure and eigenspectra of the covariance matrices of the two groups still contribute to bias amplification even when the ground-truth weights for the groups are the same. We only allow the possibility of $w^*_1 \neq w^*_2$ to be as general as possible in our theory.
>
> **(W2)** We draw our definition of bias amplification (Definition 2.1) from [1], which was peer-reviewed and published at FAccT 2023, in order to be consistent with prior literature. At a high level, our definition quantifies how many times worse model bias would be if a machine learning practitioner opted to train a single model on a mixture of data from two groups (i.e., the setting in which bias is observed in practice) vs. separate models for the data from each group (i.e., the setting which corresponds to the bias in the data alone, and thus the a priori amount of bias we would expect in the case of a single model). In sum, we seek to isolate the contribution of the *model* to bias when learning from data with different groups. This is consistent with the *conceptualization* of other bias amplification metrics in the literature (e.g., [2]).
>
> **(W3)** In Line 172, the one-hidden-layer NN model is defined correctly; we do not consider a nonlinear activation. Ridge regression with random features has been posited as a reasonable approximation for NNs in the random features regime [3, 4]. In particular, it has been argued that as the number of parameters $m \to \infty$ (as in our high-dimensional setting), gradient descent effectively learns a linear predictor over $m$ random features [3]. Furthermore, [4, 5, inter alia] are able to reproduce interesting NN phenomena like double descent using the random features model. We choose ridge regression with random projections, as it offers analytical tractability while exposing novel bias amplification phenomena related to model parameterization; such phenomena are not exposed by classical ridge regression. Furthermore, still using free probability theory and the Gaussian equivalence theorem [6], we can easily extend our theory to study bias amplification in NNs with nonlinearities in the NTK regime, as in [7] (see lines 537-539 of the manuscript), albeit with more complex equations.

---

> ### Author Response · Authors · 2024-11-18
> **Author Response (Part 2)**
>
> **To answer your questions:**
>
> **(Q1)** Much research has studied bias amplification empirically (as we detail in Section 1.2), with hypothesized factors such as conditional dependence heterogeneity. This might explain why bias amplification appears “intuitive” in the simple setting that $y = x$ and $y = -x$.
>
> However, our paper goes well beyond prior empirical work by contributing a general yet precise analytical theory of bias amplification that accommodates various model design choices (e.g., number of parameters, regularization penalty) and data distribution properties (e.g., number of features, covariance matrix structures, group imbalance, conditional dependence heterogeneity, label noises). That is, two major goals of the paper are to: (1) derive a precise profile for bias amplification in high dimensions, that goes beyond intuition, and (2) analyze the profile as a function of the aforementioned diverse model design and data distribution factors, to reproduce known bias amplification phenomena and expose new, interesting phenomena in various regimes. For example, we find that “a larger number of features than samples can amplify bias under overparameterization, there may be an optimal regularization penalty or training time to avoid bias amplification, and there can be fundamental differences in test error between groups that do not vanish with increased parameterization” (lines 78-82), among various other findings in Sections 4 and 5.
>
> We would like to clarify that **the theoretical results are not based on isotropic covariance.** The groups may have different covariance matrices (e.g., as in Section 5 (lines 465-471) and Appendix K), and the only assumption we make about the structure of the covariance matrices is that they are simultaneously diagonalizable (Assumption B.1). Our theory easily accommodates and correctly predicts bias amplification for covariance matrices with non-isotropic structure. For example, **we consider fixed covariance in Section 5 (line 468) and covariance that exponentially decays in Appendix K and Figure 10.**
>
> **(Q2)** While we may lack direct access to population parameters like $\Sigma_1$, we can estimate them using the sample covariance $\widehat{\Sigma}_1 = X^\top_1 X_1$. Even if we are unable to robustly estimate the population parameters, our theory still provides valuable insights. Our paper demonstrates that the ratios of parameters for the groups is often what matters, e.g., label noise ratio $\sigma_2^2 / \sigma_1^2$ (lines 361-367), ratio of covariance eigenvalues (Appendix L). Thus, practitioners can use our theory to get intuition about when *disparities* in the variability of labels and features across groups can severely amplify bias.
>
> **(Q3)** In practice, an optimal $\lambda$ or training time can be selected by searching for values that strike a desired balance between overall validation error (that is not too high) and bias amplification (that is not too high). The search space can be reduced by using insights from our theory such as with overparamaterization, “as regularization decreases (or training time increases), bias amplification increases and plateaus,” and with underparameterization, “as regularization decreases (or training time increases), bias de-amplification increases and plateaus” (lines 416-419). The heterogeneity definitely can be leveraged, for example, by training a mixture of experts that is regularized to deamplify bias.
>
> We hope that the explanations and clarifications provided above address the reviewer’s questions and comments.

---

> > ### Author Response · Authors · 2024-11-18
> > **Author Response (Part 3)**
> >
> > **References**
> >
> > [1] Bell S. J. and Sagun L. 2023. Simplicity Bias Leads to Amplified Performance Disparities. In Proceedings of the 2023 ACM Conference on Fairness, Accountability, and Transparency (FAccT '23). Association for Computing Machinery, New York, NY, USA, 355–369. https://doi.org/10.1145/3593013.3594003
> >
> > [2] Wang, T., Zhao, J., Yatskar, M., Chang, K. W., & Ordonez, V. (2019). Balanced datasets are not enough: Estimating and mitigating gender bias in deep image representations. In Proceedings of the IEEE/CVF international conference on computer vision (pp. 5310-5319).
> >
> > [3] Yehudai, G., & Shamir, O. (2019). On the power and limitations of random features for understanding neural networks. Advances in neural information processing systems, 32.
> >
> > [4] Adlam, B., & Pennington, J. (2020). Understanding double descent requires a fine-grained bias-variance decomposition. Advances in neural information processing systems, 33, 11022-11032.
> >
> > [5] Bach, F. (2024). High-dimensional analysis of double descent for linear regression with random projections. SIAM Journal on Mathematics of Data Science, 6(1), 26-50.
> >
> > [6] Goldt, S., Loureiro, B., Reeves, G., Krzakala, F., Mézard, M., & Zdeborová, L. (2022, April). The gaussian equivalence of generative models for learning with shallow neural networks. In Mathematical and Scientific Machine Learning (pp. 426-471). PMLR.
> >
> > [7] Adlam, B., & Pennington, J. (2020, November). The neural tangent kernel in high dimensions: Triple descent and a multi-scale theory of generalization. In International Conference on Machine Learning (pp. 74-84). PMLR.

---

> ### Author Response · Authors · 2024-11-21
>
> Thanks again for your review! Please feel free to let us know if there are any remaining comments not addressed. We appreciate any feedback, and we are happy to answer any further questions.

---

> ### Author Response · Authors · 2024-11-22
>
> We would like to advise you that a new revision of the manuscript has been uploaded, based on feedback from Reviewer m7Lv, where some line and section numbers have changed. We ask that you reference the **first revision** that we made in regards to any line or section numbers that we cited in our rebuttal. Thanks for your understanding!

---

> ### Comment · Reviewer_HU3q · 2024-11-23
>
> Thank you for your clarification on the theoretical foundation and results presented in this paper. I am happy to increase my score from 5 to 6.
>
> In the revision, I recommend including a discussion on the practical challenge of determining whether a machine learning model trained on data is prone to bias amplification (as your response to (Q2) acknowledges the population parameters cannot be accurately estimated).
>
> Additionally, I am still unclear about your response to (Q3). How can one effectively balance overall validation error and bias amplification  in practice when the latter is actually unknown?

---

> > ### Author Response · Authors · 2024-11-24
> >
> > We are glad that our response helped provide clarification! Additionally, thank you for your recommendation and question.
> >
> > We have uploaded a revision that includes a discussion (in cyan) on the practical challenge of determining whether an ML model is prone to bias amplification due to the difficulty of feature covariance matrix and label noise estimation for groups. We agree that this is important to be aware of when translating our theoretical findings into practice.
> >
> > Regarding our response to (Q3), as we would estimate the overall test error using the empirical validation error, we can estimate bias amplification using the validation set. In particular, we would need to train: (1) the main model on a mixture of data from groups, and (2) auxiliary separate models on the data for each group. Then, if we use $\widehat{R}_i (\widehat{f})$ to denote the empirical validation error of the main model on group $i$ and $\widehat{R}_i (\widehat{f}_i)$ to denote the empirical validation error of the auxiliary model for group $i$ on group $i$, we can compute an empirical bias amplification estimate as: $\widehat{ADD} = \frac{|\widehat{R}_2 (\widehat{f}) - \widehat{R}_1 (\widehat{f})|}{|\widehat{R}_2 (\widehat{f}_2) - \widehat{R}_1 (\widehat{f}_1)|}$.
> >
> > If we seek an optimal $\lambda$ or training time, it may be expensive to train the auxiliary models for each candidate value. However, as we stated in our previous response, this is where, for example, the search space of values can be reduced by using insights from our theory, such as the $ADD$ trends as a function of $\lambda$/$t$ that our theory rigorously and correctly predicts for overparameterized and underparameterized models.
> >
> > Please let us know if you have additional comments or questions.

---

### Official Review · Reviewer_X48V · 2024-11-05

**Soundness:** 3
**Presentation:** 3
**Contribution:** 3
**Rating:** 8
**Confidence:** 2

**Summary:**

This paper considers the theoretical foundation of the bias amplification phenomenon. On a Gaussian mixture data generating distribution and ridge regression model, the authors derive the exact form of the bias and variance formula under a high dimensional limit. Using the theory, the authors show the relationship between bias amplification and the model size, label noise, number of features, regularization, and training dynamics.

**Strengths:**

The proposed theory is solid with the exact formulation of the limiting risk, and it provides a rigorous foundation on the bias amplification phenomenon and its relationship with various components in machine learning models. The discovery is novel and the contribution is important to deepen our understanding of the bias amplification in ML.

**Weaknesses:**

1. Although the authors present the exact formulation of the risk in the main text, it is complicated to understand the implications of those formulas. It would be helpful to include more discussion to explain each term to better understand the results.

2. The paper's main contribution is to examine the bias amplification phenomenon using the formula. However, a formal statement about how different components affect the bias amplification is lacking. I would suggest the authors write them in formal theorems.

3. It is unclear how these theoretical findings relate to real-world deep learning models, I would suggest the authors verify the conclusion about the label noise and model size on MNIST and CNN as well.

**Questions:**

1. Is the bias amplification phenomena even in a low dimensional model, such as classical linear regression, or it is a special phenomenon on a high dimensional limit?

2. In Figure 1, the EDD plot has an extreme phase transition in the lower left corner, can the authors provide more discussion about this behavior? Moreover, the extreme case seems to mismatch with the ADD plot on the right-hand side.

3. According to Figure 11, most of the metrics are monotone with respect to lambda, how do you conclude the optimal lambda in this case?

---

> ### Author Response · Authors · 2024-11-18
> **Author Response (Part 1)**
>
> Thank you for your thoughtful feedback!
>
> **We first address each weakness mentioned below:**
>
> **(W1)** We will provide clearer intuition for each part of our formulae in the revised version of our manuscript. At a high level, in Definition 3.2, each of the terms $h_1, \ldots, h_4$ capture the limiting values of different sources of covariance between the sample covariance matrices $\overline{M}_1, \overline{M}_2$, the resolvent matrix $\overline{R}$, and the random projections matrix $S$; these sources of covariance are written explicitly in Equations 219-222 (in Appendix E). These terms naturally arise from expanding the solution to the ridge regression model with random projections, as Appendix E demonstrates.
>
> The fundamental constants $e_s, \tau_s, u_s, \rho_s$ (in Definition F.1) can be intuitively interpreted in the setting where a separate model is learned for each group and $\lambda \to 0^+$. In this setting, for ridge regression with random projections, we show in Equations 337-338 (Appendix H) that $e_s, \tau_s$ are linearly related to the first-order degrees of freedom $I_{1, 1}$ of the population covariance matrix $\Sigma_s$. $e_s$ captures the effect of the feature rate $\phi_s$ while $\tau$ captures the effect of parameterization $\gamma$. Similarly, for classical ridge regression, we show in Equations 68-70 (Appendix C) that  $e_s$ is linearly related to the degrees of freedom $\overline{\text{df}}_1$ of $\Sigma_s$.
>
> On the other hand, $u_s$ and $\rho_s$ can be understood as pseudo-variances. Indeed, Equations 341, 349, and 52 show that $u_s, \rho_s$ is proportional to $V_s$ for ridge regression with and without random projections. That is, both quantities depend on the second-order degrees of freedom  $I_{2, 2}$, or $\overline{\text{df}}_2$. In essence, our theory expands the fundamental constants to the more general setting where a single model is trained on a mixture of data from different groups and $\lambda > 0$ (Definition 3.2).
>
> **(W2)** We do make some formal statements about bias amplification in the paper. For example, in Appendix H, we directly express the bias and variance of an unregularized model trained on just group $s$ (equivalently, an unregularized model trained on data from both groups when $\Sigma_1 = \Sigma_2$) in terms of the second and first-order degrees of freedom of the population covariance matrix $I_{2, 2}$ and $I_{1, 1}$. Thus, Corollary H.1 captures how the covariance matrix affects the test risk of the model. Corollary H.1 also reveals that in the underparameterized regime ($\psi < 1$), the variance of the model’s test risk strictly increases as a function of $\psi$ (i.e., the rate of model parameters to samples); the test risk of the model skyrockets (i.e., there is catastrophic overfitting) when $\psi$ gets close to 1. Corollary H.1 additionally demonstrates that the bias and variance of the test risk decrease as $\psi$ increases.
>
> Furthermore, in Appendix K, to understand how the structure of the covariance matrix affects bias amplification, we derive the amplification profile of ridge regression with random projections with respect to the ratio $c$ of label noises. We do so in the setting where the eigenspectra of the covariance matrices have power-law decay. We show that bias amplification peaks near $c = 1$, bias reduction peaks when $c \to 0^+$, and amplification does not occur when $c \to \infty$ (lines 2832-2833). Furthermore, in lines 437-457, we mathematically reason about how a large label noise ratio $c$ greatly increases $EDD$ while $ODD$ remains relatively low.
>
> We discuss in Appendix G the difficulties of rigorously isolating the effects of different components on bias amplification. For example, even when the population covariance matrices of the two groups are proportional, the fundamental constants are quartic in the degrees of freedom of the covariance matrix. This is why we empirically investigate how different components (e.g., different covariance structures, group sizes) affect bias amplification and minority-group error in Sections 4 and 5, and extensively validate that our theory predicts these implications.

---

> ### Author Response · Authors · 2024-11-18
> **Author Response (Part 2)**
>
> **(W3)** **Per your suggestion, we run additional experiments on MNIST and CNN and report the results in the updated version of our paper (Appendix O).** We verify our conclusions about label noise and model size on MNIST and CNN. We see in Figure 15 that as we increase the label noise ratio, the $EDD$ generally increases, while the $ODD$ remains relatively low, as claimed in lines 437-457. Furthermore, in Figure 16, as the hidden dimension of the penultimate layer of the CNN increases, the $ODD$ tends towards 0 while the $EDD$ does not, which is in line with what our theory predicts in Figure 2 (in the regime where $\phi < 1$).
>
> Furthermore, ridge regression with random features has been posited as a reasonable approximation for NNs in the random features regime. For example, [1, 3, inter alia] are able to reproduce interesting NN phenomena like double descent using the random features model.
>
> **To answer your questions:**
>
> **(Q1)** Yes, the bias amplification phenomenon occurs in low-dimensional regimes. In Sections 4, 5, J, M, N, we show that our theory predicts bias amplification in experiments with models trained on only $n = 400$ samples. The high-dimensional regime is commonly studied in ML theory and statistical physics (as we mention in Section 1.2), as it makes precise analysis more tractable.
>
> **(Q2)** The phase transition in the lower left corner of the $EDD$ plot occurs at $\phi = \psi$ (i.e., where the rate of parameters to features $\gamma = 1$) (lines 311-312). This is a known interpolation pole in the literature (Appendix S7.2 of [1], [2]). There is not a mismatch with the $ADD$ plot; zooming into the $ADD$ plot reveals the interpolation threshold in light blue. This is due to the granularity of the color thresholding. We are happy to clarify this in the revised version of our manuscript.
>
> **(Q3)** When the curves are monotone with respect to $\lambda$, the optimal $\lambda$ is either at the left tail of the curve (e.g., $\lambda = 0$) or the right tail (i.e., the largest $\lambda$ among the reasonable options). A practitioner may consider the optimality of $\lambda$ with respect to a tradeoff between overall validation error and bias amplification. In contrast, Figure 11 shows that when $\psi$ is close to the interpolation threshold of 1, the bias amplification curve is often not monotone with respect to $\lambda$.
>
> **References**
>
> [1] Adlam, B., & Pennington, J. (2020). Understanding double descent requires a fine-grained bias-variance decomposition. Advances in neural information processing systems, 33, 11022-11032.
>
> [2] D’Ascoli, S., Refinetti, M., Biroli, G. &amp; Krzakala, F.. (2020). Double Trouble in Double Descent: Bias and Variance(s) in the Lazy Regime. Proceedings of the 37th International Conference on Machine Learning, in Proceedings of Machine Learning Research 119:2280-2290 Available from https://proceedings.mlr.press/v119/d-ascoli20a.html.
>
> [3] Bach, F. (2024). High-dimensional analysis of double descent for linear regression with random projections. SIAM Journal on Mathematics of Data Science, 6(1), 26-50.

---

> ### Author Response · Authors · 2024-11-21
>
> Thanks again for your review! Please feel free to let us know if there are any remaining comments not addressed. We appreciate any feedback, and we are happy to answer any further questions.

---

> ### Author Response · Authors · 2024-11-22
>
> We would like to advise you that a new revision of the manuscript has been uploaded, based on feedback from Reviewer m7Lv, where some line and section numbers have changed. We ask that you reference the **first revision** that we made in regards to any line or section numbers that we cited in our rebuttal. **In particular, the plots for our new experimental results on MNIST and CNN are in Appendix O in the first revision and in Appendix P in the latest revision.**
>
> Thanks for your understanding!

---

> > ### Author Response · Authors · 2024-11-26
> >
> > We sincerely thank the reviewer for their time and effort in evaluating our work. As the revision deadline approaches, we would greatly appreciate any feedback on our response, especially if there are remaining concerns that have not yet been addressed.

---

> > > ### Comment · Reviewer_X48V · 2024-11-28
> > >
> > > Thank you for your detailed response! I appreciate the authors' efforts to conduct new experiments, and my concerns have been properly addressed. I would like to raise my score to 8.

---

> > > > ### Author Response · Authors · 2024-11-29
> > > >
> > > > We are glad that your concerns have been addressed and are grateful for the according increase in score!

---

### Author Response · Authors · 2024-11-18
**Response to All Reviewers**

We thank all the reviewers for their thoughtful and helpful feedback! We are pleased that:

- **Reviewer X48V** finds that our theory “provides a rigorous foundation on the bias amplification phenomenon” and that the “discovery is novel and the contribution is important.”
- **Reviewer HU3q** finds that our theoretical findings “not only predict known bias amplification effects but also reveal new amplification phenomena.”
- **Reviewer m7Lv** finds that “the topic analyzed in this paper is very relevant” and “our analysis seems technically sound.”
- **Reviewer KjTp** finds that “the paper offers novel statistical theory and new insights into model bias.”

We address the weaknesses and questions raised by each reviewer in individual responses below. Please do not hesitate to let us know if you have additional comments or questions.

---

### Comment · Area_Chair_sqg8 · 2024-11-19
**Discussion Phase**

Dear Reviewers,

Please review the authors' replies and consider the feedback from other reviewers. If your concerns remain unresolved, feel free to ask for further clarifications. We have until November 26th for discussion.

Thank you for your efforts.

Best regards,
Area Chair

---

> ### Comment · Area_Chair_sqg8 · 2024-11-23
>
> Thank you Reviewers HU3q and m7Lv for engaging in the discussion.
>
> Please, Reviewers X48V and KjTp read the authors' replies and feedback from other reviewers. If any concerns remain, request clarifications. This is your last chance to engage.
>
> Thank you for your efforts.
>
> Best regards,
> Area Chair

---

### Meta-Review · Area_Chair_sqg8 · 2024-12-20

**Metareview:**

### Summary

The paper proposes a theoretical framework for analyzing bias amplification in simple neural networks.
Despite the framework’s simplifications, it reproduces several empirical observations from previous studies and predicts new ones.
The results are further validated with numerical simulations.

### Strengths

Despite its simplicity, the paper reconciles various findings in the literature and explores areas previous models couldn’t, particularly overparameterization.
Furthermore, the analysis relies on different techniques than previous theoretical papers, providing an additional technical contribution.
The results and predictions are validated with extensive numerical results.

### Weakness

The paper is highly technical and requires the reader to first familiarize themselves with the formalism before fully appreciating the results. Additionally, the theoretical framework is limited as it relies on a linear model. While random projections are used, the framework is likely insufficient to capture certain complex features that emerge from non-linearities, potentially limiting its applicability in real-world settings.

### Reason for acceptance

The problem addressed by this paper is very relevant and there are very limited theoretical frameworks that allow us to understand these phenomena. This paper extends the range of phenomena that theory could address and it therefore represents a valuable contribution for the conference.

**Additional Comments On Reviewer Discussion:**

During the discussion period, the reviewers raised several important points:
* **Limited comparison with existing literature:** In particular, some of the results presented in the paper were already discussed in previous publications that were not cited in the original submission. In response, the authors clarified the technical differences between their work and existing literature, explaining what their model uniquely contributes. The updated version of the paper dedicates a substantial section to this issue, alongside an expanded discussion of related works.
* **Clarity concerns:** Reviewers noted issues with both the formal presentation of the theorems/definitions and the clarity in explaining the results. The authors addressed these concerns by improving the clarity of their theoretical exposition and enhancing the explanations throughout the revised paper, making the contributions easier to follow.
* **Connections with real data and applications:** There were initial concerns about the applicability of the framework to real-world data. The authors resolved these by pointing reviewers to additional results provided in the appendix, which offer further insights into how their theoretical findings translate to practical scenarios.

---

### Decision · Program_Chairs · 2025-01-22

Accept (Poster)